# Inhibition of mitochondrial complex I reverses *NOTCH1*-driven metabolic reprogramming in T-cell acute lymphoblastic leukemia

Natalia Baran[1], Alessia Lodi[2], Yogesh Dhungana[3], Shelley Herbrich[1], Meghan Collins[2], Shannon Sweeney[2], Renu Pandey[2], Anna Skwarska[1], Shraddha Patel[1], Mathieu Tremblay[4], Vinitha Mary Kuruvilla[1], Antonio Cavazos[1], Mecit Kaplan[5], Marc O. Warmoes[6], Diogo Troggian Veiga[7], Ken Furudate[1,8], Shanti Rojas-Sutterin[4], Andre Haman[4], Yves Gareau[4], Anne Marinier[4], Helen Ma[1], Karine Harutyunyan[1], May Daher[5], Luciana Melo Garcia[5], Gheath Al-Atrash[5], Sujan Piya[1], Vivian Ruvolo[1], Wentao Yang[9], Sriram Saravanan Shanmugavelandy[10], Ningping Feng[11], Jason Gay[11], Di Du[6], Jun J. Yang[9], Fieke W. Hoff[1], Marcin Kaminski[12], Katarzyna Tomczak[13], R. Eric Davis[14], Daniel Herranz[15], Adolfo Ferrando[16], Elias J. Jabbour[1], M. Emilia Di Francesco[17], David T. Teachey[18], Terzah M. Horton[19], Steven Kornblau[1], Katayoun Rezvani[5], Guy Sauvageau[4], Mihai Gagea[20], Michael Andreeff[1], Koichi Takahashi[1], Joseph R. Marszalek[11], Philip L. Lorenzi[6], Jiyang Yu[21], Stefano Tiziani[2], Trang Hoang[4,22] & Marina Konopleva[1✉]

T-cell acute lymphoblastic leukemia (T-ALL) is commonly driven by activating mutations in *NOTCH1* that facilitate glutamine oxidation. Here we identify oxidative phosphorylation (OxPhos) as a critical pathway for leukemia cell survival and demonstrate a direct relationship between *NOTCH1*, elevated OxPhos gene expression, and acquired chemoresistance in pre-leukemic and leukemic models. Disrupting OxPhos with IACS-010759, an inhibitor of mitochondrial complex I, causes potent growth inhibition through induction of metabolic shut-down and redox imbalance in *NOTCH1*-mutated and less so in *NOTCH1*-wt T-ALL cells. Mechanistically, inhibition of OxPhos induces a metabolic reprogramming into glutaminolysis. We show that pharmacological blockade of OxPhos combined with inducible knock-down of glutaminase, the key glutamine enzyme, confers synthetic lethality in mice harboring *NOTCH1*-mutated T-ALL. We leverage on this synthetic lethal interaction to demonstrate that IACS-010759 in combination with chemotherapy containing L-asparaginase, an enzyme that uncovers the glutamine dependency of leukemic cells, causes reduced glutaminolysis and profound tumor reduction in pre-clinical models of human T-ALL. In summary, this metabolic dependency of T-ALL on OxPhos provides a rational therapeutic target.

---

A full list of author affiliations appears at the end of the paper.

T-cell acute lymphoblastic leukemia (T-ALL) is an aggressive type of leukemia, requiring intensive chemotherapy regimens[1,2]. The backbone of standard of care chemotherapy for T-ALL includes vincristine, dexamethasone, and L-asparaginase (VXL)[3,4]. Despite improved cure rates in pediatric T-ALL, relapsed disease occurs in 40% of adult primary T-ALL patients and carries a poor prognosis[1,2]. At the molecular level, T-ALL is associated with abnormalities in oncogenic transcription factors[5,6], with the most frequent activating mutations in *NOTCH1* that lead to upregulation of the pro-survival NOTCH1 signaling[7–9] and mutations in *FBXW7* gene, which is similar to mutations in NOTCH1 PEST domain, results in increased ICN1 protein stability, often co-occurring with mutations in *NOTCH1*[10–12]. NOTCH1 activation supports differentiation of hematopoietic progenitors toward thymocytes and activates multiple anabolic pathways required for cell growth, proliferation, and survival of both healthy and leukemic T-cells[6,7,13]. Thus, targeting NOTCH1 and its downstream effectors is an attractive therapeutic approach in T-ALL[14–16]. However, the development of NOTCH1 inhibitors remains challenging, due to acquired resistance and poor tolerability, so targeting metabolic pathways downstream of NOTCH1 may offer an alternative therapeutic approach.

Leukemia cells depend on oxidative phosphorylation (OxPhos) to meet their energetic and biosynthetic demands[17–27]. Phenformin, a mitochondrial complex I/OxPhos inhibitor (OxPhos-i), reduced disease burden in preclinical T-ALL models and improved overall survival (OS)[25]. IACS-010759, a more potent and specific small-molecule complex I inhibitor, has been investigated in clinical trials of acute myeloid leukemia (AML) and advanced cancers (NCT03291938, NCT02882321)[28]. Glutaminolysis is another metabolic feature of T-ALL and AML that is crucial for anaplerosis, biosynthetic processes, and cellular redox maintenance[29,30]. Based on promising preclinical evidence of antileukemic activity, the glutaminase inhibitor (GLS-i) CB-839 is being investigated in ongoing clinical trials[31]. Also, part of the efficacy of L-asparaginase against T-ALL relies on its GLS-i activity[32].

We report here our use of bioinformatics, genetic, and pharmacological approaches to investigate the metabolic features of T-ALL. We identify OxPhos as a targetable dependence required for mitochondrial energy production, whose disruption eliminates disease using preleukemic and leukemic cell lines, patient samples, and patient-derived xenograft (PDX) models. We also establish a mechanistic rationale for combining OxPhos-i and GLS-i, by showing synergistic efficacy of IACS-010759 and L-asparaginase-based chemotherapy against chemoresistant *NOTCH1*-mutated T-ALL.

## Results

### Activity of OxPhos is linked to Notch1 status in murine preleukemic and human models of T-ALL

Bioinformatic analysis of various datasets from T-ALL[12,33], normal thymocytes[34], and murine T- ALL cell lines[35] indicated that NOTCH1-bound target genes (NBTs) were enriched for MYC targets, genes of the OxPhos pathway, the tricarboxylic acid (TCA) cycle, and the mitochondrial electron transport chain (ETC) (Fig. 1a)[35,36], possibly linking NOTCH1 signaling to MYC and OxPhos. During normal thymocyte differentiation[34], expression of NBTs increases from the ETP stage onward, peaking at the DN3/DN4 stage, and remains elevated up to the immature single positive (ISP) stage (Fig. 1b). Beginning at the DN3 stage, there is a high expression of NBTs that are involved in OxPhos, which sharply drop at the DP stage (Fig. 1c), together with the global shutdown of canonical NOTCH1 target genes. Moreover, following the introduction of the constitutively active NOTCH1 oncogene (*Notch1/IC*) into DP cells[37], gene set enrichment analysis (GSEA) showed upregulation of NBTs exclusively within the OxPhos pathway, at both the preleukemic (*p* < 0.0001) and leukemic (*p* < 0.0001) stages (Supplementary Fig. 1a, b).

Having established a possible link between NOTCH1 and OxPhos in murine preleukemic T-cells, we next examined whether *NOTCH1* activation affects putative OxPhos activity in human T-ALL, using RNA-seq data from two T-ALL clinical trials[12,33] and a data-driven network inference algorithm (NetBID). As compared to samples without *NOTCH1* mutation, there was a higher activity of OxPhos signature genes in *NOTCH1*-mutant T-ALL (*p* = 0.012) along with several pathways associated with mitochondrial machinery (*p* = 0.01; Fig. 1d–f, Supplementary Fig. 1c–e) and a lower activity of apoptosis pathways (*p* = 0.0073; Fig. 1d, g; Supplementary Fig. 1c, f).

Furthermore, the comparison of dependence scores from DepMap unbiased genome-scale CRISPR-Cas9 screening data[38] between three T-cell lines (PF382, SUPT1, HSB2) to other hematologic cell lines (*N* = 73) or other cancer cell lines (*N* = 686) showed significantly higher number of OxPhos genes in T-ALL. In addition, OxPhos genes were significantly over-represented among T-ALL dependent genes, as compared to those for hematologic malignancies or other cancer cell lines (*p* = 0.03 *and* *p* = 0.048, respectively; Supplementary Fig. 1g) suggesting the importance of OxPhos genes for T-ALL survival[38]. Taken together, bioinformatic analysis provides evidence that oncogenic activation of *NOTCH1* is associated with OxPhos, and suggested that treatment with OxPhos-i could reduce the viability of murine and human T-ALL, potentially depending on *NOTCH1* status.

### OxPhos downstream of NOTCH1 is essential for preleukemic and leukemic stem cell function

Our bioinformatic analysis suggested that treatment with OxPhos-i could reduce the viability of murine and human T-ALL, potentially depending on *NOTCH1* status.

In the murine *SCL-LMO1* transgenic model, progression to acute leukemia is driven by the acquisition of *Notch1* gain-of-function mutations, starting at 7–8 weeks of age (Fig. 2a; Supplementary Fig. 2a), reproducing a major feature of the human disease. Indeed, introducing the *Notch1/IC* oncogene that drives elevated and activated NOTCH1 in thymocytes (Fig. 2b) is sufficient to transform *SCL-LMO1*-induced pre-LSCs into hypercompetitive leukemia-propagating cells and to trigger aggressive T-ALL without latency[36,39]. To directly address the question whether NOTCH1 activation affects the response to OxPhos-i, we harvested pre-LSCs at 1, 3, and 5 weeks of age, i.e. at least one month before the appearance of *Notch1* mutations (Supplementary Fig. 2a). The functional importance of NOTCH1 activation was directly addressed by comparing cells expressing or not the constitutively active *Notch1/IC* allele. When maintained on stromal cells presenting the NOTCH1 ligand DL4 (MS5-DL4), pre-LSCs from five genetic models: *LMO1*tg, *SCL*tg*LMO1*tg, *Notch1*tg, *LMO1*tg*Notch1*tg, and *SCL*tg*LMO1*tg*Notch1*tg, showed comparable growth inhibition upon OxPhos-i with IACS-010759, and comparable IC50 in dose–response curves (Fig. 2c, d; Supplementary Fig. 2b). Of note, the inhibitory effect of IACS-010759 on pre-LSCs was observed when cells started to proliferate in culture on day 2 and 3, but not on day 1 (Fig. 2e). In the absence of DL4 stimulation, *LMO1*- or *SCL-LMO1*-induced pre-LSCs survived for 2 days without proliferation and were insensitive to IACS-010759 (Fig. 2d, f). Under these conditions, the *Notch1* transgene was sufficient to confer responsiveness to IACS-010759, even though cell proliferation was minimal.

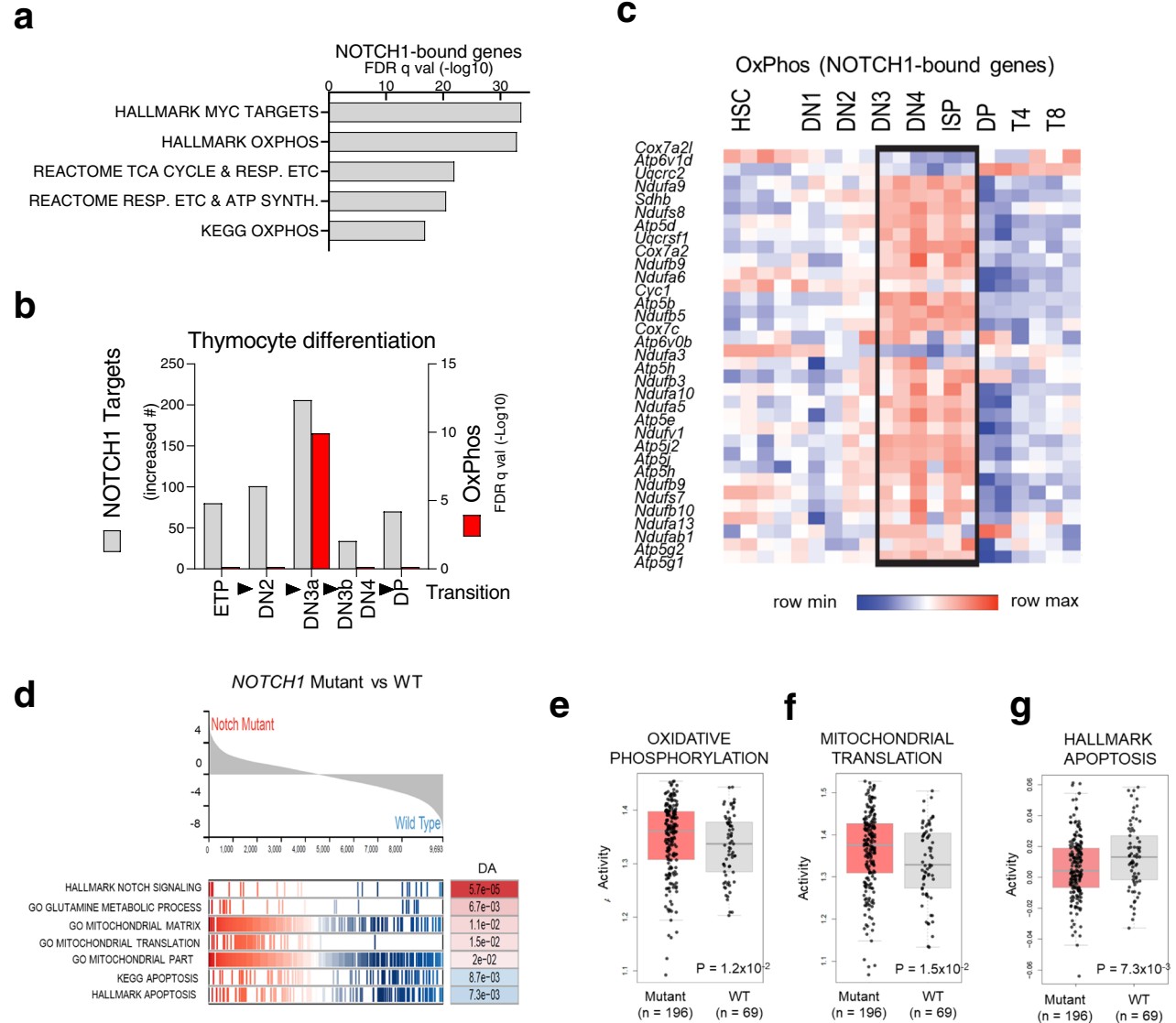

**Fig. 1 Oxidative phosphorylation downstream of NOTCH1 is essential for preleukemic stem cells function. a** Gene set enrichment analysis of NOTCH1-bound genes in Molecular Signatures Database. The list of genes in which NOTCH1 binding was observed within 2 kb of transcription start sites in murine T-ALL cell lines was extracted as described in Materials and Methods. False discovery rates (FDRs) are shown for overlaps computed with Hallmark gene sets or canonical pathways (Reactome and Kyoto Encyclopedia of Genes and Genomes). TCA: tricarboxylic acid; ETC, electron transport chain. **b** Dynamics of NOTCH1 target gene expression during thymocyte differentiation. The numbers of NOTCH1 target genes that increased >1.3-fold at each transition (gray bars, left axis). Gene expression data are from the Immunological Genome Project. GSEA was conducted on the gene sets whose expression increased at each transitional stage. FDR values overlap with the OxPhos gene set (Hallmark) were computed as above (green bars, right axis). ETP, early T-lineage precursor; DN3a, (CD4−CD8−CD25+CD44−CD28−), DN3b, (CD4−CD8−CD25+CD44−CD28+). **c** Heatmap of expression of NOTCH1-bound genes within the OxPhos pathway. Ultralow input (ULI) RNA-seq data were obtained from IMMGEN. ETP, early thymocyte progenitor (Lin^lowCD25−CD44+Kit+), DN2, double negative 2 (Lin^lowCD25+CD44+Kit+), DN3 (CD4−CD8−CD25+CD44−), DN4 (CD4−CD8−CD25−CD44−CD28+), ISP, immature single positive (CD4−CD8+CD24hiTCR^lo), DP (CD4+CD8+TCR^loCD24hi), T4 (CD4+CD8− − TCR^hi), T8 (CD4−CD8+TCR^hi). **d** GSEA of the TARGET gene set (Therapeutically Applicable Research to Generate Effective Treatments) (N = 265), indicated enrichment of genes related to glutamine metabolism, mitochondrial metabolism as well as translation and downregulation of apoptosis-related genes in patients with *NOTCH1*-mutations (each vertical bar in x-axis is gene rank in the pathway list and y-axis represents running enrichment score). **e** The OxPhos gene signature in T-ALL patients from TARGET cohort. **f** Enrichment of mitochondrial translation genes in T-ALL patients from TARGET cohort. **g** Enrichment of apoptosis-related genes in T-ALL patients from the TARGET cohort; for (**e**), (**f**) and (**g**): two-sided *t*-test; The center line represents the median and whiskers represents maximum (Q3 + 1.5*IQR) and minimum value (Q1+1.5*IQR).

Growth inhibition was not due to concurrent apoptosis in pre-LSCs nor decreased viability of the supporting stromal cells (Supplementary Fig. 2c–f), indicating that OxPhox-i specifically acts on pre-LSCs. Although cells expressing the *Notch1/IC* allele were inhibited by IACS-010759, the curves did not reach the same levels of maximum inhibition observed in DL4-stimulated

cultures (Fig. 2d, Supplementary Fig. 2b). We then compared drug sensitivities by computing the areas under the curves (AUC). As shown in Fig. 2g, drug sensitivity was highest in DL4-stimulated cultures and all five genotypes cluster together. In the absence of DL4 exposure, pre-LSCs separated into two clusters: *LMO1*- and *SCL-LMO1*-induced pre-LSCs cluster with control

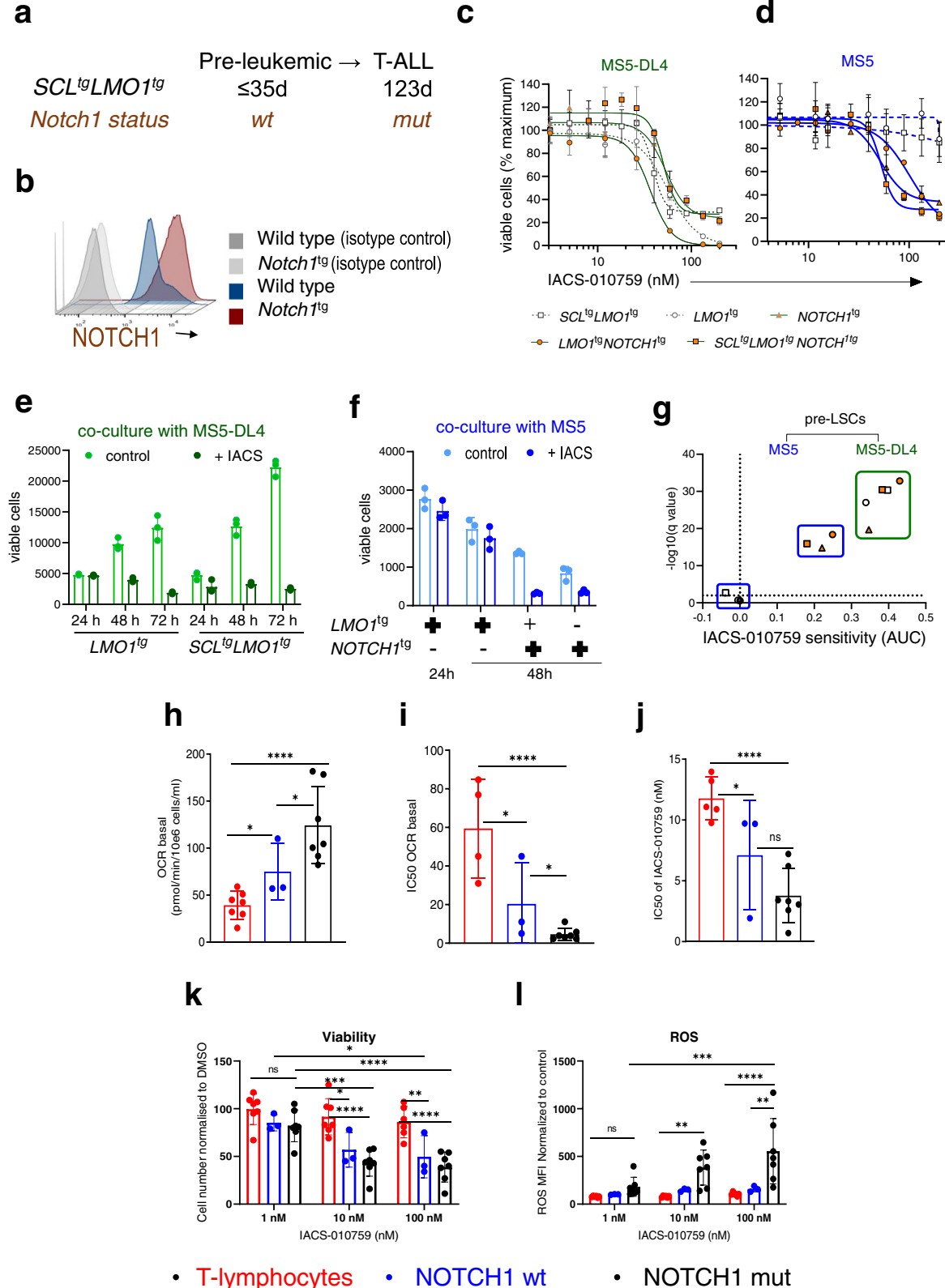

cultures exposed to DMSO whereas all three *Notch1/IC* genotypes cluster together, away from DMSO controls. Nonetheless, drug sensitivity remained lower compared to DL4-stimulated cells. Together with NOTCH1-occupancy of genes of the OxPhos pathway[35] (Fig. 1a), the response to OxPhos-i observed here suggests two levels of drug sensitivity: one directly regulated via NOTCH1 binding to OxPhos genes, and the other one associated

with DL4-induced cell proliferation. Together, the genetic models and the MS5-DL4 co-culture model establish a direct link between NOTCH1 activation and OxPhos-i sensitivity.

Next, we asked whether *NOTCH1* status impacts mitochondrial respiration in human models of T-ALL. Oxygen consumption rate (OCR) measurements in T-lymphocytes and T-ALL cell lines separated by *NOTCH1* mutation status showed an increased

**Fig. 2 NOTCH1 activation affects the response to OxPhos-inhibition of primary pre-LSCs. a** *Notch1* gene mutation in SCL$^{tg}$LMO1$^{tg}$ preleukemic and leukemic thymocytes. **b** Overexpression of NOTCH1 protein by flow cytometry. Thymocytes from wt- and *NOTCH1*$^{tg}$ mice were stained with anti-NOTCH1 antibody or an isotype control. **c** dose–response of primary pre-LSCs to IACS-010759, co-cultured on MS5-DL4 stromal cells followed by 48 h drug treatment (mean ± SD, $n = 3$ independent experiments in triplicates). **d** Dose–response of pre-LSCs to IACS-010759 treatment, co-cultured on MS5 stromal cells followed by 48 h drug treatment (mean ± SD, $n = 3$.independent experiments in triplicates). **e** Effects of IACS-010759 treatment (133 nM or DMSO) on viability of pre-LSCs cultured on MS5-DL4 (mean ± SD, $n = 3$ independent experiments). **f** Effects of IACS-010759 treatment (133 nM or DMSO) on viability of pre-LSCs cultured on MS5 (mean ± SD, $n = 3$ independent experiments). **g** Pre-LSC sensitivity to IACS-010759 for the indicated genotypes. The Area under the curves (AUC) were computed from dose–response data illustrated in (**c**). Shown is the AUC difference obtained between DMSO (black ring) and drug-treated cells ($p < 0.00001$, multiple unpaired *t* tests, $n = 3$ mice per genotype, in triplicates); with (green) and without DL4 (blue), in pre-LSCs with or without NOTCH1 oncogene in the absence of DL4. **h** Basal oxygen consumption rate (OCR) in T-lymphocytes ($n = 7$), *NOTCH1*-wt ($n = 3$) and -mutant cell lines ($n = 7$) by Mito Stress Test assay (mean ± SD, $n = 3$ independent measurements for each line, 4 replicates); one-way ANOVA; *$p < 0.05$; ***$p < 0.001$. **i** IC$_{50}$ for basal OCR inhibition (Supplementary Figs. 3c and 4) for T-lymphocytes ($n = 4$) and *NOTCH1*-wt ($n = 3$) and -mutated T-ALL cell lines ($n = 7$) (mean ± SD, n = 3 independent experiments for each line, 4 replicates); one-way ANOVA; *$p < 0.05$; ***$p < 0.001$. **j** IC$_{50}$ for viability inhibition for T-lymphocytes ($n = 5$), *NOTCH1*-wt ($n = 3$) and -mutant cell lines ($n = 7$) treated with IACS-010759 (0–123 nM, 96 h); (mean ± SD, $n = 3$ independent experiments); one-way ANOVA; ns-no significance, **$p < 0.005$. **k** Viable cell number in T-lymphocytes ($n = 7$), *NOTCH1*-wt ($n = 3$) and -mutant ($n = 7$) T-ALL cell lines, treated with IACS-010759 (1, 10, 100 nM, 96 h), (mean ± SD, $n = 3$ independent experiments per line, 3 replicates); two-way ANOVA: ns-no significance, *$p < 0.05$; **$p < 0.005$; ***$p < 0.001$; and ****$p < 0.0001$. **l** ROS (MFI) treated with IACS-010759 (1, 10, 100 nM; 96 h) in T-lymphocytes ($n = 6$), T-ALL cell lines with wt- ($n = 3$) and mutant *NOTCH1* ($n = 7$); (mean ± SD, $n = 3$ independent experiments per line, 3 replicates); two-way ANOVA: ns-no significance, *$p < 0.05$; **$p < 0.005$; ***$p < 0.001$; and ****$p < 0.0001$.

OCR in a subset of cells with *NOTCH1* wild type (wt) ($p = 0.03$; Fig. 2h) and in all cell lines harboring *NOTCH1* mutation ($p = 0.0004$; Fig. 2h) compared with OCR of healthy T-lymphocytes. Albeit the *NOTCH*-mutant cells on average showed higher basal and maximal OCR (Supplementary Fig. 3a), these differences did not reach statistical significance. All T-ALL cell lines showed high expression of mitochondrial complexes I–V, with no clear correlation between mitochondrial protein assembly level and *NOTCH1* status (Supplementary Fig. 3b).

As expected, IACS-010759 inhibited OxPhos across T-ALL cell lines ($n = 11$), reducing basal and maximal OCR in both *NOTCH1*-wt and -mutant cell lines ($p = 0.05$ and $p = 0.005$, respectively, Supplementary Figs. 3c–d, 4), with the latter showing greater sensitivity to OCR inhibition when compared to T-lymphocytes ($p = 0.0006$, Fig. 2i, Supplementary Fig. 3e). Consistent with this, IACS-010759 produced striking dose-dependent inhibition of ATP production in T-ALL lines (Supplementary Fig. 3f), with the lowest IC$_{50}$ in *NOTCH1*-mutant lines versus T-lymphocytes ($p = 0.0006$, Fig. 2j), while only moderate ATP and viability reduction of less than 20% at the highest dose was seen in healthy T-lymphocytes (Supplementary Fig. 3f–h). The response to IACS-010759 observed in matured T-lymphocytes was not increased by stimulation with DLL4. Neither ATP production, cell number, nor apoptosis rate were significantly changed upon combined DLL4 stimulation and IACS-010759 treatment (Supplementary Fig. 3i–k). Reduced ATP production upon OxPhos-i was paralleled by decreased viability in T-ALL cell lines, PDX models and primary T-ALL samples (Supplementary Fig. 3g, h). At the clinically achievable concentration of 10 nM, IACS-010759 reduced viability substantially in both *NOTCH1*-wt ($p = 0.01$) and *NOTCH1*-mutated cells ($p < 0.0001$, Fig. 2k), and minimally affected the viability of healthy T-lymphocytes (Fig. 2k). Similar effects were seen at 100 nM dose of IACS-010759, producing significant reduction in both *NOTCH1*-wt ($p = 0.0093$) and mutants ($p < 0.0001$, Fig. 2k). These results correlated with basal OCR changes (Supplementary Fig. 5a, $R^2 = 0.4572$, $p = 0.0079$). Of note, IACS-010759 increased significantly the level of reactive oxygen species (ROS), where *NOTCH1*-mutation contributed to 8-fold higher and *NOTCH1*-wt 2-fold higher ROS increase when compared to T-lymphocytes ($p = 0.0012$), leading to higher ROS accumulation at 10 nM and 100 nM doses ($p = 0.05$ and 0.0036) and indicating a distinct propensity to undergo redox imbalance upon OxPhos-i (Fig. 2l). ROS increase correlated with viability reduction ($R^2 = 0.559$,

$p = 0.0021$), indicating that T-ALL cells have ineffective mechanisms of coping with oxidative stress (Supplementary Fig. 5b). DLL4 incubation did not produce significant changes in respiration of matured T-lymphocytes (Supplementary Fig. 5c±f), suggesting that DLL4 stimulation of NOTCH1 plays a role mainly at the early stages of lymphocyte development.

OxPhos-i in *NOTCH1*-mutated cells also produced time-dependent changes in total levels and phosphorylation status of multiple metabolism-relevant proteins (Supplementary Fig. 5g), including activation of mTOR and glycolytic enzymes, activation of LKB1 and AMPK in response to ATP depletion, and upregulation of DNA damage (pH2AX) in response to energetic stress and ROS accumulation. GSEA of OxPhos-i-induced transcriptional changes revealed downregulation of mitochondrial gene sets involving transcriptional processes, the ETC, and cytochrome c release ($p = 0.046$; Supplementary Fig. 5h).

In summary, our data provide direct evidence that ligand-dependent activation of NOTCH1 signaling by DLL4 or by the *NOTCH1* oncogene upregulates the OxPhos pathway and confers responsiveness of primary pre-LSCs and human T-ALL cells to OxPhos-i.

**OxPhos-I induces distinct metabolic reprogramming in *NOTCH1*-mutated T-ALL toward reductive metabolism of glutamine.** To investigate further the role of *NOTCH1* in regulating metabolism in T-ALL cells, we analyzed the impact on cellular processes of supplementation or withdrawal of essential nutrients glutamine (Gln), glucose (Glc), and pyruvate (Pyr). Supplementation with Gln alone, but not with Glc, contributed to 50% of ATP generation by *NOTCH1*-mutant cell lines ($p = 0.0064$; Fig. 3a, Supplementary Fig. 6a). The addition of Pyr with Gln further improved ATP generation to 80% ($p = 0.001$) and increased basal and maximal OCR to the level observed in complete medium (Gln + Glc+Pyr; $p = 0.001$; Fig. 3b, Supplementary Fig. 6b), but had no impact on extracellular acidification rates (ECAR).

The presence of Gln and Pyr without Glc in *NOTCH1*-mutant cells contributed to a distinct aerobic metabolic phenotype with predominant OxPhos activity, as indicated by the basal OCR/ECAR ratio, while the addition of Glc clustered the metabolic profile toward an energetic/glycolytic phenotype (Fig. 3c). In contrast, *NOTCH1*-wt cells showed their most efficient ATP production and growth under Glc supplementation ($p = 0.001$; Supplementary Fig. 6c). These observations revealed divergent

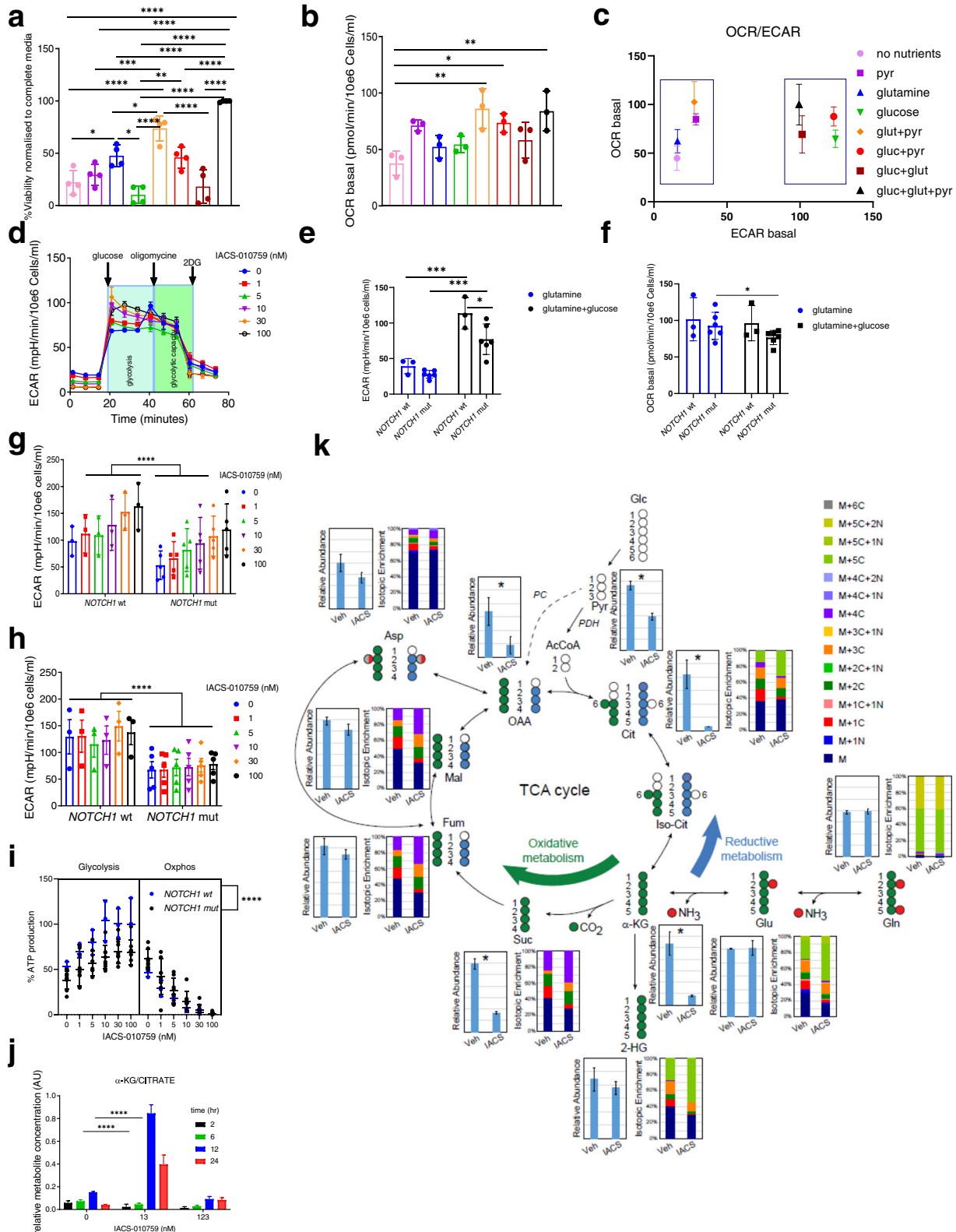

nutrient utilization, with *NOTCH1*-wt cells relying strongly on Glc, while *NOTCH1*-mutants were dependent on Gln to activate OxPhos and generate ATP.

To investigate whether T-ALL cells switch to glycolysis to cope with OxPhos-i, we measured the effects of IACS-010759 on basal and maximal ECAR, in the presence of Gln, before and after Glc supplementation (Fig. 3d; Supplementary Fig. 7a, b). With Gln only, both *NOTCH1*-wt and -mutant relied only on OxPhos (high OCR, low ECAR) (Fig. 3e, f), and IACS-010759 inhibited Gln-driven OCR in a dose-dependent manner (Supplementary Fig. 7). Acute Glc addition produced an immediate shift toward maximal glycolytic activity ($p = 0.005$, Fig. 3e, f), however this switch toward glycolysis was significantly higher for *NOTCH1*-wt than in mutant, thereby establishing a new metabolic equilibrium

**Fig. 3 Mitochondrial complex I inhibition induces distinct metabolic reprogramming followed by reductive metabolism of glutamine in *NOTCH1*-mutated cell lines, uncovering a potential metabolic mechanism of acquired resistance under OxPhos blockade. a** ATP levels for *NOTCH1*-mutated T-ALL ($n = 4$) treated 96 h in media with: no nutrients, pyruvate (Pyr), glutamine (Gln), glucose (Glc), Gln+Pyr, Glc+Pyr, Glc+Gln, Glc+Pyr+Gln, (mean ± SD, $n = 4$ independent experiments, 3 replicates); one-way ANOVA: \*$p < 0.05$; \*\*$p < 0.005$; \*\*\*$p < 0.001$; \*\*\*\*$p < 0.0001$. **b** Basal OCR of 4 *NOTCH1*-mutated T-ALL cell lines treated 24 h as in (**a**), (mean ± SD, $n = 3$ independent experiments, 4 replicates); one-way ANOVA: \*$p < 0.05$; \*\*$p < 0.005$; \*\*\*$p < 0.001$; \*\*\*\*$p < 0.0001$. **c** Metabolic phenotype (basal OCR/basal ECAR ratio) as in (**a**); (mean ± SD, $n = 3$ independent experiments, 4 replicates). **d** Extracellular acidification rate (ECAR) in *NOTCH1*-mutated T-ALL cell line PF-382 after IACS-010759 treatment (0–123 nM, 4 h); Glycolysis Stress Test (mean ± SD, $n = 3$ independent experiments, 4 replicates). **e** Basal ECAR of *NOTCH1*-wt ($n = 3$) and -mutated ($n = 6$) cell lines treated 24 h in media with Gln only (blue) and after acute injection of Glc (black); (Glycolysis Stress Test), (mean ± SD, $n = 3$ independent experiments, 4 replicates); two-way ANOVA: \*\*\*$p = 0.0002$. **f** Basal OCR in *NOTCH1*-wt ($n = 3$) and -mutated ($n = 6$) cell lines treated as in (**e**); (mean ± SD, $n = 3$ independent experiments, 4 replicates); two-way ANOVA: \*$p = 0.0250$. **g** Basal ECAR of *NOTCH1*-mutated ($n = 5$) and -wt ($n = 3$) cell lines treated with IACS-010759 (0–123 nM, 4 h); (mean ± SD, $n = 3$ independent experiments, 4 replicates); two-tailed $t$ test: \*\*\*\*$p < 0.0001$. **h** Glycolytic capacity in *NOTCH1*-mutated ($n = 5$) and -wt ($n = 3$) cell lines treated with IACS-010759 (0–123 nM, 4 h), Glycolysis Stress Test; (mean ± SD, $n = 3$ independent experiments, 4 replicates); one-way ANOVA: \*\*\*\*$p < 0.0001$. **i** ATP production (%) for *NOTCH1*-mutant ($n = 8$) and -wt ($n = 3$) cell lines treated with IACS-010759 (0–123 nM, 4 h), ATP Rate Assay; (mean ± SD, $n = 3$ independent experiments, 4 replicates); one-way ANOVA: \*\*\*\*$p < 0.0001$. **j** Levels of citrate and α-ketoglutarate and citrate/α-ketoglutarate ratio over time in *NOTCH1*-mutated cell line PF382; (mean ± SD, $n = 1$, 4 replicates); two-way ANOVA: \*\*\*\*$p < 0.0001$. **k** Stable isotope-resolved metabolomics ultra-high-pressure liquid chromatography MS analysis (SIRM) of *NOTCH1*-mutant PF-382 cell line cultured with 13C5,15N2-glutamine, treated with DMSO or IACS-010759 (10 nM, 12 h); (mean ± SD, $n = 1$, 4 replicates). two-tailed $t$-test; \*$p < 0.05$; \*\*$p < 0.005$; \*\*\*$p < 0.001$; and \*\*\*\*$p < 0.0001$.

(high OCR, high ECAR) (Fig. 3e, f). This was accompanied by significant decrease of OCR observed upon Glc injection, indicating switch toward glycolysis with reduced OxPhos utilization in *NOTCH1*-mutants and thus more tightly regulated nutrient utilization. Furthermore, independently of OxPhos activity, *NOTCH1*-wt cell lines displayed higher glycolysis ($p = 0.0001$, Fig. 3g) and glycolytic capacity, showing nearly 3-fold higher ECAR than *NOTCH1*-mutants at both baseline and upon OxPhos-i ($p = 0.0001$, Fig. 3h).

The simultaneous real-time measurement of ATP (Seahorse) generated by glycolysis and OxPhos, in the presence of both Gln and Glc, showed a comparable inhibition of OxPhos-derived ATP by IACS-010759 in all cell lines. However, glycolysis-derived ATP generation was lower in *NOTCH1* mutants ($p < 0.0001$; Fig. 3i; Supplementary Fig. 8a, b), confirming the distinct role of *NOTCH1*-status in the maintenance of metabolic equilibrium upon OxPhos-i.

IACS-010759, like other Complex I inhibitors, has been shown to impair mitochondria by blocking the TCA cycle (thereby reducing concentrations of its metabolites: citrate, α-ketoglutarate [α-KG], succinate, fumarate, and malate) and by inhibiting ATP production and decreasing aspartate levels (thereby depleting energy and blocking nucleotide biosynthesis)[17,19,21,25,40]. To define the metabolic effects of OxPhos-i in *NOTCH1*-mutated T-ALL cells, we performed an untargeted metabolomics analysis that showed accumulation of nucleoside monophosphate (NMP) ($p = 0.04$) and redox imbalance, as indicated by diminished ratios (reductive to oxidized) of glutathione and coenzyme Q10 (CoQ10) (Supplementary Fig. 9a). Also, OxPhos-i at 10 nM decreased levels of citrate, succinate, L-glutamine, and L-aspartate, slightly increased fumarate and malate, and transiently increased α-KG ($p < 0.0001$; Supplementary Fig. 9b, c). Since the elevated α-KG/citrate ratio indicates utilization of reductive OxPhos[41,42], we next analyzed the dynamics of α-KG/citrate ratio changes upon OxPhos-i. Indeed, a spike in the α-KG/citrate ratio was observed after 12 h ($p < 0.0001$; Fig. 3j; Supplementary Fig. 9c), indicating a switch to reductive OxPhos metabolism upon OxPhos blockade.

To further support reductive oxidative phosphorylation as a mechanism of counteracting OxPhos blockade, we next performed stable isotope-resolved metabolomics (SIRM) to follow the fate of incorporation of isotopically-enriched $^{13}C_5,^{15}N_2$-glutamine ($^{13}C_5,^{15}N_2$-Gln) or $^{13}C_6$-glucose ($^{13}C_6$-Glc) in *NOTCH1*-mutant (Fig. 3k; Supplementary Fig. 10) and *NOTCH1*-wt cell lines (Supplementary Fig. 11a, b). As expected,

OxPhos-i decreased levels of TCA intermediates in *NOTCH1*-mutant cells (Fig. 3k). We also observed simultaneous incorporation of Glc and its utilization in glycolysis, indicated by a moderately increased rate of lactate enrichment (Supplementary Fig. 10, 11a). Furthermore, the increase of $^{13}C_5^{15}N_2$-Gln isotopomer fractions (M+5 citrate, M+4 malate, and M+4 fumarate) indicated an enrichment of Gln incorporation (Fig. 3k), supporting a compensatory Gln influx into the TCA cycle upon OxPhos blockade.

Taken together, these results demonstrate an increase in Gln oxidative phosphorylation in *NOTCH1*-mutated cell lines and selective boosting to utilize Gln in the reductive oxidative phosphorylation pathway upon OxPhos-i, offering a possible mechanism of metabolic escape in *NOTCH1*-mutated T-ALL.

**Blocking the glutamine pathway is lethal to *NOTCH1*-mutant T-ALL cells when combined with complex I inhibition.** Because reductive metabolism of Gln was increased by OxPhos-i, we examined further the importance of Gln metabolism in T-ALL. GSEA of TARGET T-ALL RNA-seq datasets showed that genes involved in Gln metabolism are expressed more highly by *NOTCH1*-mutant tumors ($p = 0.0067$; Fig. 4a). These observations prompted the hypothesis that the dual inhibition of both OxPhos and Gln metabolism would facilitate the efficacy of OxPhos-i against T-ALL. We tested this hypothesis by determining the sensitivity to IACS-010759 of cell lines and primary T-ALL samples, under varying conditions of selective supplementation of essential substrates (Gln, Glc, and Pyr). *NOTCH1*-mutant cell lines showed higher sensitivity to IACS-010759 in the presence of Gln (blue curve) and glutamine+pyruvate (orange curve), with average IC$_{50}$s of 0.677 nM and 0.552 nM ($p < 0.05$ and $p < 0.0005$ respectively, Fig. 4b-c; Supplementary Fig. 12a–c).

These results indicate that Gln and Pyr, in the absence of Glc, support OxPhos and sensitize *NOTCH1*-mutated T-ALL cells to OxPhos-i. In contrast, Gln+Pyr supplementation reduced the sensitivity to OxPhos-i of *NOTCH1*-wt cells (Supplementary Fig. 12a, b). In *NOTCH1*-mutated T-ALL, the growth-inhibitory effect of OxPhos-i in Gln/Pyr-supplemented medium was reduced by the addition of Glc (black curve) (Fig. 4b, c; Supplementary Fig. 12a–c). Exposure to Glc, alone or in the presence of Gln, switched *NOTCH1*-mutated cells from OxPhos towards glycolysis (brown curve), flattening the dose–response curve upon OxPhos-i, indicating reduced sensitivity to IACS-010759; however, this increased the sensitivity of *NOTCH1*-wt cells

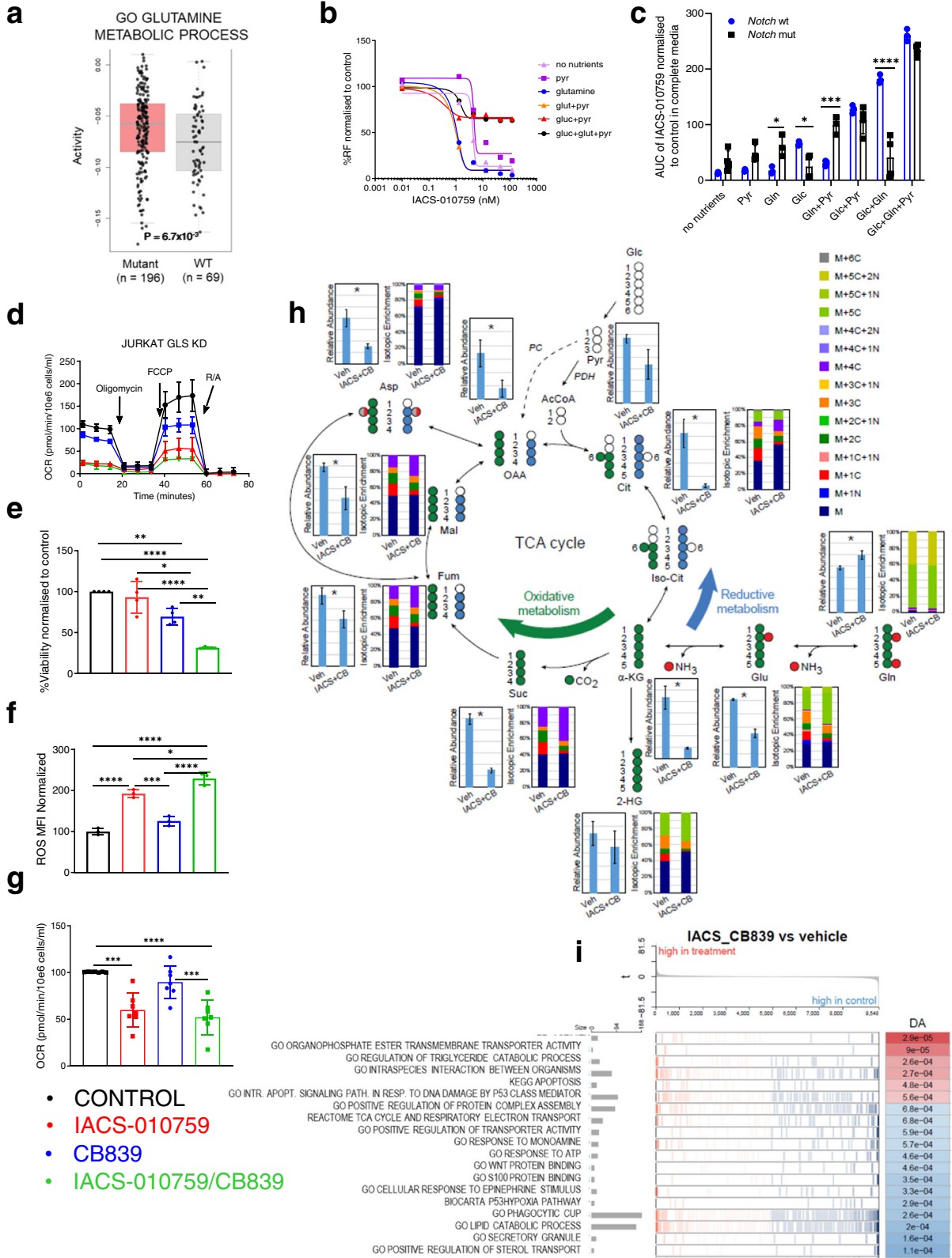

to OxPhos-i ($p < 0.0001$; Fig. 4b, c; Supplementary Fig. 12a–c). These results suggest that the maintenance of OxPhos function depends on Gln in *NOTCH1*-mutated cells and that Gln is crucial for the sensitivity to OxPhos-i ($p = 0.0001$, Fig. 4c; Supplementary Fig. 12a, b).

We next reassessed the impact of OxPhos-i on T-ALL cells in the context of inhibiting GLS, a key enzyme in Gln utilization.

Compared to treatment with IACS-010759 alone, *GLS* knock-down combined with OxPhos-i inhibited cell growth, reduced cell viability (Supplementary Fig. 13a), reduced basal and maximal OCR (Fig. 4d; Supplementary Fig. 13b), and resulted in inhibition of the c-MYC, Akt/mTOR, and AMPK pathways, initiation of the DNA damage response pathway (γ-H2AX), and activation of the apoptosis pathway (cleaved PARP and cleaved

**Fig. 4 Blockade of glutamine pathway through glutaminase knockdown and glutaminase inhibition in combination with complex I inhibition induces metabolic shutdown in T-ALL cells in vitro. a** Enrichment of glutamine metabolism components in T-ALL TARGET cohort; (Gene Ontology analysis); two-sided $t$ test; The center line represents the median and whiskers represents maximum (Q3+1.5*IQR) and minimum value (Q1+1.5*IQR). **b** Viability of *NOTCH1*-mutated T-ALL cell lines (96 h, 0–100 nM of IACS01759) upon following conditions: no nutrients, pyruvate (Pyr), glutamine (Gln), glutamine +pyruvate (Glut+Pyr), glucose+pyruvate (Glc+Pyr), glucose+glutamine+pyruvate (Glc+Glut+Pyr), normalized to DMSO. **c** AUC upon IACS-010759 treatment (by CTG) calculated from (**b**), normalized to DMSO from Gln+Glc+Pyr, for *NOTCH1*-wt (blue) ($n = 3$) and *NOTCH1*-mutated (black) T-ALL cell lines ($n = 4$) (mean ± SD, $n = 3$ independent experiments, $n = 3$ replicates/condition). Two-way ANOVA; *$p = 0.02$; ***$p = 0.0002$; ****$p < 0.0001$; **d** Representative OCR response in *NOTCH1*-mutated T-ALL cell lines JURKAT subjected to doxycycline-induced knockdown of GLS (blue) and treatment with IACS-010759 (10 nM) for 4 h (red), or the combination (green) (mean ± SD, $n = 3$ biological replicates, $n = 4$ replicates/condition). **e** Cell viability in *NOTCH1*-mutated T-ALL patient samples ($n = 4$) treated with CB-839 (1 µM) and IACS-01759 (10 nM) after 96 h, by FL (mean ± SD, $n = 4$ independent patient samples, $n = 3$ replicates/condition); one-way ANOVA; *$p = 0.039$; **$p = 0.0017$; ****$p < 0.0001$. **f** Reactive oxygen species (ROS) -MFI of in *NOTCH1*-mutant T-ALL cell lines ($n = 3$) treated 96 h with CB-839 (1 µM) and IACS-010759 (10 nM). (mean ± SD, $n = 3$) by FL; one-way ANOVA; *$p = 0.0202$; ***$p = 0.0005$; ****$p < 0.0001$. **g** Basal OCR in *NOTCH1*-mutated T-ALL cell lines after treatment with vehicle, IACS-010759 (10 nM, 4 h), CB-839 (1 µM, 12 h), or IACS-010759/CB-839 combination (mean ± SD, $n = 7$ cell lines, $n = 3$ independent experiments, $n = 4$ replicates per condition); one-way ANOVA: ***$p = 0.0003$; ****$p < 0.0001$. **h** SIRM ultra-HPLC MS of *NOTCH1*-mutated T-ALL cell line PF-382 after enrichment with $^{13}C_5^{15}N_2$-glutamine and treatment with the combination of 10 nM IACS-010759 and 1 µM CB-839 or vehicle for 12 h. **i** Gene expression analysis in *NOTCH1*-mutant T-ALL after treatment with the combination of 10 nM IACS-010759 and 1 µM CB-839 compared to vehicle; (x-gene rank in the pathway list; y-running enrichment score).

caspase-3) (Supplementary Fig. 13c). Furthermore, treatment with the GLS-i CB-839 in combination with OxPhos-i led to a synergistic drop in cell viability (Supplementary Fig. 14a), reflected by an AUC decrease ($p = 0.001$; Supplementary Fig. 14b) and by delta BLISS index values (Supplementary Fig. 14c). The efficacy of dual inhibition was confirmed across cell lines (Supplementary Fig. 14d, e) and primary *NOTCH1*-mutated T-ALL blasts ($p < 0.0001$, Fig. 4e), by decreased cell number and increased apoptosis rate for the GLS-i/OxPhos-i combination. Supplementation with Pyr and Gln failed to reverse the inhibitory effects of dual inhibition (Supplementary Fig. 14f). Given that Gln contributes to the redox homeostasis, we tested the impact of the dual blockade on ROS induction and showed that GLS-i plus OxPhos-i elevated ROS greater than OxPhos-i alone ($p < 0.0001$; Fig. 4f). Finally, the GLS-i/OxPhos-i combination additively reduced basal and maximal OCR, but only in *NOTCH1*-mutant cells ($p < 0.0001$; Fig. 4g; Supplementary Figs. 15a, b, 16a, b). Of note, dual GLS-i/OxPhos-i blockade only moderately reduced the cellular respiration (24 h) and ATP levels (72 h) of healthy bone marrow cells (Supplementary Fig. 16c, d) and did not significantly affect colony formation (Supplementary Fig. 16e, f).

The evidence of successful blockade of glutaminolysis by GLS-i upon OxPhos-i was confirmed by SIRM in both *NOTCH1*-mutated (Fig. 4h; Supplementary Figs. 17, 19a) and *NOTCH1*-wt cell lines (Supplementary Fig. 18a, b, 19b). GLS-i/ OxPhos-i severely reduced the levels of TCA cycle intermediates citrate, α-KG, and succinate ($p < 0.0001$), and decreased the levels of fumarate ($p = 0.012$), malate ($p = 0.0003$), aspartate ($p < 0.0001$), and glutamate ($p < 0.0001$), accompanied by relative accumulation of Gln (Fig. 4h, Supplementary Figs. 17, 18a, b, 19a, b). These results demonstrate a very slow TCA turnover indicative of cellular exhaustion (Fig. 4h; Supplementary Fig. 17). Similar effects were observed for *NOTCH1*-wt cells, without significant accumulation of Gln in the labeled fraction (Supplementary Fig. 18a, b). This potent metabolic shutdown was reflected by GSEA, indicating the downregulation of genes involved in the TCA cycle, ETC, ATP response (Fig. 4i; Supplementary Fig. 20a, b), and mitochondrial activity (Supplementary Fig. 20a), accompanied by an upregulation of apoptosis activation in response to DNA damage and catabolic processes (Fig. 4i; Supplementary Fig. 20a, b). Given that redox balance is controlled by glutathione concentration[43,44], we tested whether N-acetylcysteine (NAC) supplementation would rescue the effects of OxPhos-i alone or combined with GLS-i on

viability, apoptosis, and ROS production. Surprisingly, co-treatment with NAC at 0.5 or 2.0 mM[45] increased the cytotoxicity of IACS-010759 or IACS-010759/CB-839 combination in T-ALL cells, decreasing cell number, inducing apoptosis, and triggering a stronger accumulation of ROS. These data suggest that NAC supplementation is unable to quench the ROS accumulation and glutathione depletion upon OxPhos-i/GLS-i treatment, making those changes irreversible. On the contrary, NAC completely rescued the growth-inhibitory effects of CB-839 alone (Supplementary Fig. 21a–c).

Together, these data demonstrate that dual OxPhos and glutamine pathway blockade induced profound metabolic shutdown in T-ALL cells with *NOTCH1* mutations, in turn resulting in apoptosis.

**Blockade of glutamine metabolism with complex I inhibition improves depth of tumor burden reduction in vivo.** We evaluated dual OxPhos/GLS inhibition in vivo, using a *Notch1*-mutated *GLS*^fl/fl murine T-ALL model[29] in C57BL/6 mice (Fig. 5a). Following established leukemia engraftment in peripheral blood at day 7 (Supplementary Fig. 22a, b), mice received vehicle or IACS-010759 therapy, alone or with concomitant tamoxifen administration to induce *GLS* knockout selectively in leukemic cells. After only 5 days of dual pharmacologic/genetic blockade, disease burden was reduced in both peripheral blood (PB) (Fig. 5b) and bone marrow (BM) ($p = 0.0001$; Fig. 5c and f), with substantial clearance of leukemia cells in the spleen ($p = 0.013$; Fig. 5d), reduction of splenomegaly (Fig. 5e) and clearance of leukemia cells in the liver (Fig. 5f). Consistent with in vitro observations, metabolomic analysis of PB from leukemic mice subjected to dual intervention, revealed an accumulation of NMP, with decreased ATP and TCA intermediates, indicative of profound metabolic reprogramming that suppressed all critical amino acid and nucleotide biosynthesis pathways (Ingenuity Pathway Analysis, IPA) (Fig. 5g, Supplementary Fig. 22a, b). The long-term evaluation found that leukemia progression was significantly delayed in mice treated with either IACS-01075 or subjected to *GLS* knockout ($p = 0.0001$; Fig. 5h). This antileukemic effect was significantly enhanced under dual inhibition ($p < 0.0001$; Fig. 5h), with no circulating leukemia cells in peripheral blood up to day 220 (Fig. 5i). Taken together, our results demonstrate that blocking *GLS* in combination with complex I inhibition provides a powerful approach to abrogate leukemia progression in vivo.

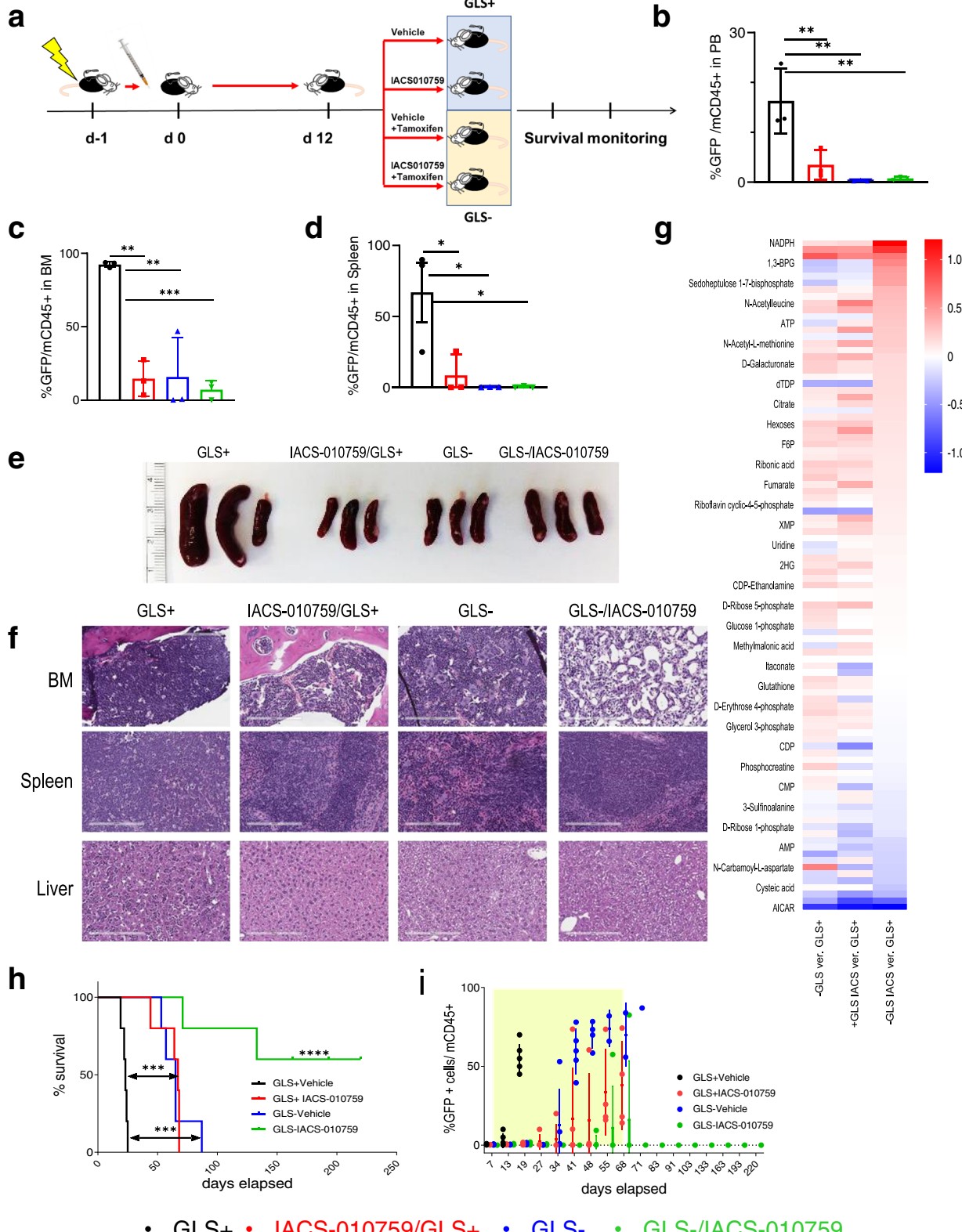

**OxPhos-I added to Vxl chemotherapy induces prominent metabolic changes in vivo and improves overall survival in T-ALL PDX models.** Since L-asparaginase, a component of the VXL standard-of-care chemotherapy, contains GLS-inhibiting activity, we next focused on metabolic consequences induced by VXL-chemotherapy in T-ALL. First, we determined the effect in vitro on OxPhos activity of each VXL agent used alone or combined with OxPhos-i. Dexamethasone and vincristine increased OCR significantly ($p = 0.0001$; Supplementary Fig. 23a) and led to moderate reductions of cell viability (Supplementary Fig. 23b, c). L-asparaginase similarly induced OCR ($p = 0.0001$; Supplementary Fig. 23a) but decreased viability potently as monotherapy (Supplementary Fig. 23d). When combined with OxPhos-i, all drugs synergized to reduce viability, with the most

**Fig. 5 Complex I inhibition in combination with glutaminase deletion impedes *Notch1*-mutant T-ALL leukemia development, improves tumor burden reduction, and extends overall survival in a murine model. a** Schematic of study design utilizing murine Notch1-mutated GLS fl/fl. **b** GFP$^+$mCD45$^+$ leukemia cells (%) in PB from mice bearing murine *Notch1*-mutated GLS fl/fl T-ALL cells after 5 days of treatment as shown in (**a**) (mean ± SD, $n = 3$ individual mice per treatment arm); one-way ANOVA; **$p = 0.0025$. **c** GFP$^+$mCD45$^+$ leukemia cells (%) in BM from mice as in (**b**) (mean ± SD, $n = 3$ individual mice per treatment arm); one-way ANOVA, **$p = 0.001$; and ***$p = 0.0005$. **d** GFP$^+$mCD45$^+$ leukemia cells (%) in spleens from mice as in (**b**) (mean ± SD, $n = 3$ individual mice per treatment arm); one-way ANOVA, *$p = 0.013$. **e** Spleens harvested from mice bearing murine *Notch1*-mutated GLS fl/fl T-ALL cells following 5 days of treatment with vehicle; IACS-010759; tamoxifen; or combination of IACS-010759 and tamoxifen ($n = 3$). **f** H&E staining of BM, spleen and liver from one representative mouse from each group, selected based on complementary FL results from (**c**) and (**d**). Scale bars represent 100 μm; from the same experiment as FL data. **g** Heatmap of MS analysis of metabolites in PB of mice bearing murine *Notch1*-mutated GLS fl/fl T-ALL cells after 5 days of treatment with vehicle; IACS-010759; tamoxifen; or IACS-010759 and tamoxifen; (mean log of fold-change ratio over the level of metabolites measured in mice treated with vehicle); (mean ± SD, $n = 4$ individual mice/group). **h** Kaplan–Meier survival curves of mice transplanted with murine *Notch1*-mutated GLS fl/fl T-ALL. Mice ($n = 5$ per treatment arm) were treated with vehicle; tamoxifen (1 mg/mouse over 5 days); IACS-010759 (5 days per week 5 mg/kg Mo–Fr) or concomitantly with IACS-010759 and tamoxifen; treatment was initiated after detection of GFP$^+$ cells in PB by FL at day 7; log-rank test, ***$p < 0.005$, ****$p < 0.0001$. **i** Weekly tumor burden monitoring by detection of GFP$^+$mCD45$^+$ leukemia cells in PB in mice as indicated in (**h**); the timeframe of treatment is labeled by the yellow windows on the graph; (mean ± SD, $n = 5$ individual mice per treatment arm).

potent effect observed in the VXL combination (Supplementary Fig. 23e). Further, in comparison to OxPhos-i or VXL alone, the VXL/OxPhos-i combination potently reduced both basal and maximal OCR only in *NOTCH1* mutants ($p < 0.0001$; Supplementary Figs. 24a, b; 25a, b). In contrast, the impact of basal and maximal OCR and ECAR in healthy bone marrow cells was moderate after pre-treatment with OxPhos-i/VXL for 24 h (Supplementary Fig. 25c), did not significantly impact ATP production after 72 h (Supplementary Fig. 25d) and did not affect the colony-forming capacity (Supplementary Fig. 25e).

VXL/OxPhos-i, like GLS-i/OxPhos-i, led to metabolic shutdown as measured by SIRM (Supplementary Figs. 26a, b, 27a, b), with expected changes in TCA metabolites citrate, α-KG, succinate ($p < 0.0001$) and moderate changes in malate (Supplementary Fig. 28a, b). The ʟ-Asparagine driven glutamine-depleting effects were observed in media obtained after incubating cells with VXL and VXL/OxPhos-i, with a reduction of glutamine level down to 70% of control for *NOTCH1*-mutated cells ($p = 0.0004$, Supplementary Fig. 26c) and *NOTCH1*-wt cells ($p = 0.02$, Supplementary Fig. 27c), respectively. Further, complete depletion of asparagine ($p < 0.0001$), accumulation of glutamate ($p < 0.0001$) and aspartate ($p < 0.0001$) were observed in media with VXL treatment (Supplementary Figs. 26c, 27c). This translated into intracellular depletion of Asparagine ($p < 0.0001$) and decrease in Aspartate level in VXL/OxPhos-i treated cells ($p = 0.0003$; Supplementary Fig. 28a, b).

Similar to results obtained for CB-839, we observed rescue after treatment of JURKAT and PF382 cells with NAC of 2 mM only in VXL treatment; in conditions containing OxPhos-i, NAC treatment led to deeper reduction of viable cell number ($p < 0.0001$), increased apoptosis rate ($p < 0.0001$) and enhanced ROS accumulation ($p < 0.0001$) (Supplementary Fig. 29a–c).

Next, we investigated the effect of VXL/OxPhos-i on the metabolic and transcriptomic profiles of T-ALL PDXs with advanced disease. The pharmacodynamic profile of mice bearing 70-80% of leukemia cells in PB was determined after 12 h of treatment with IACS-010759, VXL, or VXL/IACS-010759. Metabolome analysis revealed profound downregulation of pathways related to OxPhos in cells derived from PB, spleen, and BM after OxPhos-i (Fig. 6a; Supplementary Figs. 29d, 30, 31), facilitated further by VXL. Leukemia cells from PB, BM, and spleen displayed a striking reduction in citric acid, isocitric acid, α-KG, succinate, fumarate, and malate, as well as lesser decreases in aspartate and Gln after VXL/OxPhos-i combination therapy (Fig. 6a; Supplementary Figs. 29d, 30, 31). These results indicate effective blockade of glutaminolysis and induction of redox imbalance, evidenced by low levels of glutathione and homocysteine. VXL/OxPhos-i reduced nucleotide triphosphates (NTP),

leading to blockade of DNA and RNA synthesis (Fig. 6a; Supplementary Figs. 29d, 30, 31). Pathway analysis by IPA confirmed that VXL/OxPhos-i upregulated AMPK and down-regulated the sirtuin pathway, CoQ10 biosynthesis, and nucleotide biosynthesis (Supplementary Fig. 29e), as reflected further by OCR reduction ($p < 0.0001$; Supplementary Fig. 32a) and down-regulated pathways showed by GSEA (Supplementary Fig. 32b).

The impact of these metabolic changes was further validated by single-cell CyTOF analysis, which confirmed the reduction in PB leukemia cells by only 12 h of treatment in vivo. Fractions of T-ALL cells expressing CD45, CD7, and CD3 were reduced and showed reduced expression of Ki67 and other intracellular markers such as NOTCH1, p-p38, c-MYC, p-ERK, and p-H2AX (Fig. 6b, c). GSEA of global gene expression changes occurring in vivo after VXL/OxPhos-i showed a global shutdown of all mitochondria-related processes, indicating that this treatment negatively impacted a wide spectrum of metabolic and cellular processes in leukemia cells (Fig. 6d; Supplementary Fig. 32b).

Finally, we determined the short-term (5 days) and long-term effects of VXL/OxPhos-i on disease progression and survival in mice bearing human T-ALL (Supplementary Fig. 33a). In all three *NOTCH1*-mutated T-ALL PDX models, OxPhos-i/VXL produced a profound reduction of spleen size (Supplementary Fig. 33b), spleen weight ($p < 0.0001$; Supplementary Fig. 33c), and tumor burden reduction in PB (Supplementary Fig. 33d), spleen (Fig. 6e; Supplementary Fig. 33f), and BM (Supplementary Fig. 33e, f). Disease progression, as measured by circulating leukemia burden, was 2–3-fold slower with VXL/OxPhos-i (Fig. 6f), translating into longer overall survival in the VXL/OxPhos-i mice cohort ($p < 0.0001$; Fig. 6g). Taken together, our data indicate that OxPhos-i combined with standard-of-care chemotherapy containing ʟ-asparaginase, with its GLS-inhibitory activity, effectively induces metabolic and transcriptomic cellular catastrophe that impedes leukemia progression and leads to the extension of survival.

## Discussion

Our study characterized the metabolic features of *NOTCH1*-mutated and *NOTCH*-wt T-ALL and elucidated the link between *NOTCH1* and mitochondrial activation. Our findings suggest that targeting OxPhos in T-ALL is an effective way to control disease burden in vivo and that compensatory metabolic mechanisms upon OxPhos-i can be overcome by inhibiting reductive metabolism of the glutamine pathway.

OxPhos is a well-studied target in malignancies, with upregulated OxPhos activity being observed in AML and other hematologic and solid tumors. IACS-010759, a potent small molecule

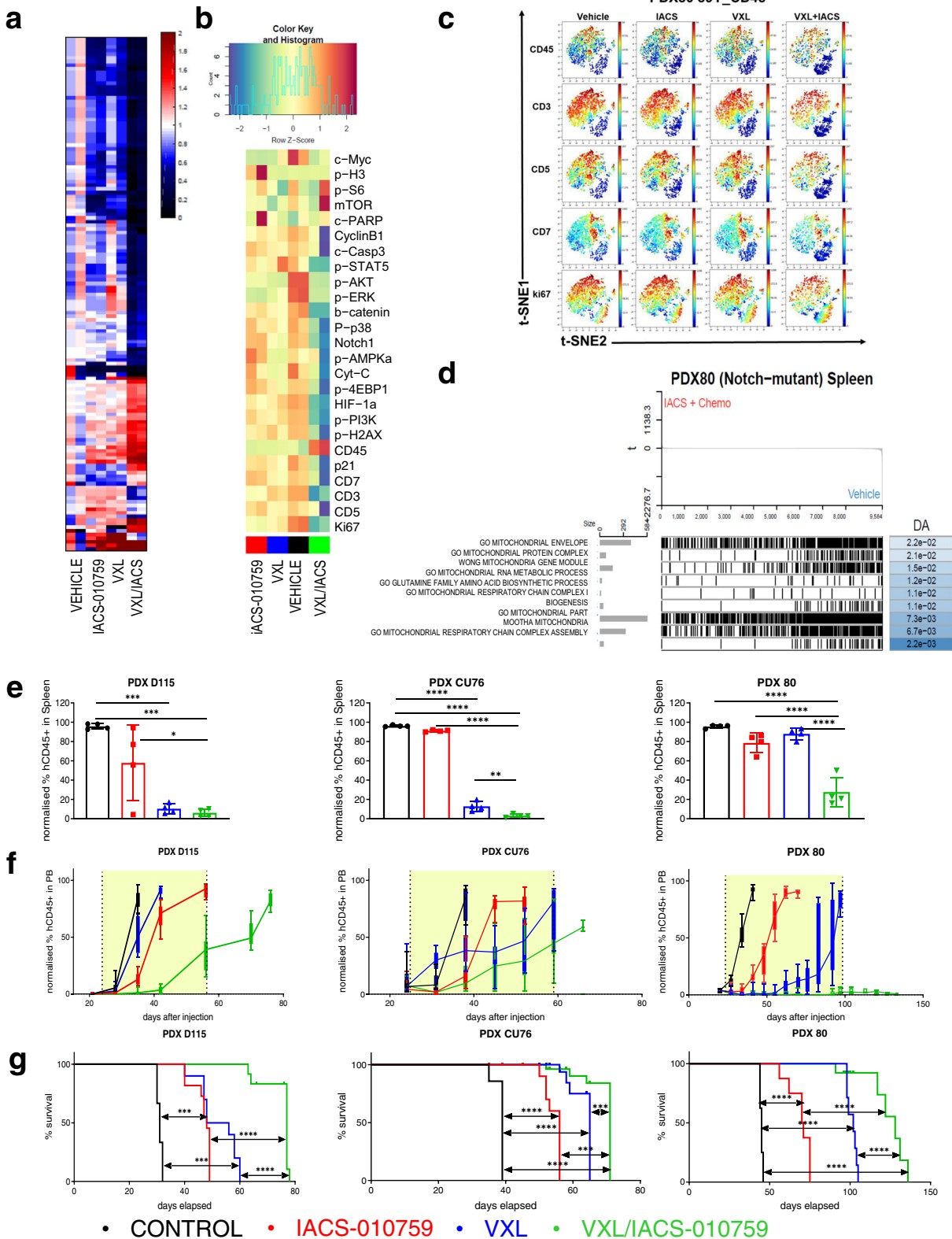

with selective inhibitory effects on OxPhos, has been studied in these indications[20,21,28,46].

We previously showed that T-pre-LSCs are more resistant to mainstay chemotherapy agents than the bulk of leukemic cells[47]. We now provide direct evidence that pre-LSCs in a T-ALL murine model are sensitive to OxPhos-i by IACS-010759 in the presence of induced or constitutive NOTCH1 signaling. We,

therefore, propose that the synergy of OxPhos-i and VXL chemotherapy will effectively target both drug-resistant pre-LSCs and bulk *NOTCH1*-dependent leukemic cells. Supporting the link between mutation of the *NOTCH1* oncogene and OxPhos activity, gene expression analysis from two independent patient cohorts demonstrated enrichment of OxPhos related genes in the transcriptome of *NOTCH1*-mutated T-ALL. Further, DepMap

**Fig. 6 VXL and IACS-010759 induce profound metabolic damage in vivo and leads to improvement of tumor burden reduction and overall survival in PDX models in vivo. a** MS Heatmap of metabolites in BM of mice transplanted with NOTCH1-mutated PDX80 after 12 h of treatment with vehicle, IACS-010759, VXL, or the combination of IACS-010759 and VXL (n = 2). **b** CyTOF heatmap of spleen samples from PDX80 model treated with vehicle, IACS-010759, VXL, or the combination of IACS-010759 and VXL (n = 2). **c** CyTOF UMAP for CD45, CD3, CD5, CD7 and Ki67 markers in spleen samples from PDX80 model treated with vehicle, IACS-010759, VXL, or the combination of IACS-010759 and VXL (n = 2). **d** Gene expression analysis of spleen samples from PDX80 model after treatment with the combination of IACS-010759 and VXL compared to vehicle treatment. **e** Leukemic engraftment (% of human CD45+ leukemia cells) in spleen at day 5 of treatment with vehicle, VXL, IACS-010759, or the combination of IACS-010759 and VXL, as measured by FL in 3 PDX models (from left to right: PDX D115, PDX CU76, PDX 80) (mean ± SD, n = 4 individual mice per treatment arm); one-way ANOVA; *p = 0.01; **p = 0.0016; ***p = 0.0002, ****p < 0.0001. **f** Tumor burden development in 3 PDX models (from left to right: PDX D115, PDX CU76, PDX 80) measured as weekly detection of circulating human CD45+ cells in peripheral blood (PB) and expressed as percent of normalized human to sum of human and murine CD45+ cells in mice undergoing treatment with vehicle, IACS-010759 (5 mg/kg once daily, 5 days on/2 days off), VXL (once weekly), or the combination of IACS-010759 and VXL (mean ± SD, n = 10 mice per treatment arm for PDX D115 and PDX CU76 and n = 8 mice per treatment arm for PDX 80). **g** Kaplan–Meier survival curves of mice transplanted with PDX models (from left to right: PDX D115, PDX CU76, PDX 80) and treated with vehicle, IACS-010759, VXL, or the combination of IACS-010759 and VXL (n = 8). Treatment was initiated once the level of circulating leukemia cells in PB reached 0.5–1%; the timeframe of treatment is labeled by the yellow windows on the graphs. log-rank test: ***p < 0.0006 ****p < 0.0001.

dependency analysis and metabolic analysis by Seahorse technology indicated stronger OxPhos dependency and higher oxygen consumption rates in NOTCH1-mutated cells compared to wt counterparts. This evidence is in line with data that demonstrated a dependency on ETC upon phenformin and mitochondria-targeted antioxidant MVE treatment in T-ALL Jurkat cells[25,40].

Targeting mitochondria in T-ALL is supported by mechanistic studies with a few agents (avicin, ABT-737, mitoTEMPO and oligomycin), but the search for compounds eligible for translational studies is still ongoing. We provide evidence that OxPhos dependency in NOTCH1-mutated T-ALL impacts ATP production and impairs proliferation. As previously demonstrated for AML, OxPhos-i with IACS-010759 blocked mitochondrial activity in T-ALL and exhibited potent antileukemic activity as monotherapy in vitro.

It should be noted that IACS-010759 also impacted viability and reduced OCR of NOTCH-wt cells compared with healthy lymphocytes. Although not studied here, alternative mechanisms, in addition to NOTCH1 gene mutation, are known to activate NOTCH signaling in T-ALL[10–12]; and while metabolic phenotypes described in this study are most prominent in NOTCH-mutated cells, OxPhos clearly contributes to metabolic fitness of NOTCH1 unmutated cells. In line with data reported for IACS-010759 in AML and for phenformin and MB1-47[48] in T-ALL, comprehensive metabolomic studies revealed that IACS-010759 caused an energetic crisis by depletion of ATP and NAD/NADH ratio, confirming on-target effects, with blockade of RNA and DNA synthesis through depletion of other NTPs. At the proteomic and transcriptomic levels, IACS-010759 induced AMPK, Akt, and DNA damage pathways, with further activation of glycolytic enzymes in vitro and activation of AMPK and DNA damage in vivo. We also demonstrated ROS induction, redox imbalance and DNA damage upon complex I inhibition in T-ALL cells, yet these did not translate into apoptosis, indicating the ability of T-ALL cells to cope with the mitochondrial blockade. Nevertheless, IACS-010759 demonstrated antileukemic activity in vivo delaying a progression and significantly extending survival in a murine model of Notch1-mutated T-ALL and human NOTCH1-mutated PDX models.

To dissect mechanisms of metabolic compensation, we examined the glycolytic capacity of T-ALL samples. Of note, we observed that while all T-ALL cells upregulate glycolysis, the level of upregulation is strongly dependent on NOTCH1-status, whereby NOTCH1-mutated cells exhibited a lower capacity to utilize glycolysis upon OxPhos blockade. Instead, IACS-010759 caused a strong reduction of TCA intermediates α-KG, succinate, oxaloacetate, and citrate, and a moderate reduction of fumarate and malate. In parallel, we found a profound change

in the α-KG/citrate ratio, indicating compensatory reprogramming through recruitment of glutamine reductive metabolism. This was supported by nutrient depletion assays, together with Seahorse and SIRM, which indicated that leukemic cells utilize glutaminolysis as a source of energy to fuel the TCA cycle and recruit reductive metabolism of glutamine to cope with metabolic stress under OxPhos-i. We demonstrated that activation of reductive glutamine metabolism reduced the efficacy of IACS-010759 as a single agent, allowing cells to escape the complex I blockade. These findings are consistent with the previously reported results demonstrating that glutamine sustains the TCA cycle in T-ALL[29,49], and resembles the reported reliance on Gln metabolism observed following FLT3 inhibitor treatment in AML[50–53]. This finding was further supported by transcriptomic analyses demonstrating enrichment of glutamine-related signatures in NOTCH1-mutated T-ALL. Notably, Herranz et al. identified T-ALL as a glutaminolysis-dependent disease showing that the blockade of glutamine metabolism in T-ALL led to autophagy induction[29], while in the study of Nguyen et al. glutamine depletion triggered apoptosis in T-ALL cells[49]. Altogether, our work provides strong evidence that T-ALL cells have a unique vulnerability related to both OxPhos and glutaminolysis.

To exploit this metabolic reprogramming therapeutically, we examined dual inhibition of OxPhos and glutaminolysis. We demonstrated that the genetic silencing of GLS in combination with complex I inhibition can eradicate NOTCH1-mutated T-ALL, producing a similar extension of survival as reported direct co-targeting Notch by γ-secretase inhibitor[29]. Given potential challenges of the combinatorial use of two novel small molecules in clinical trials, we utilized amino-acid depletion with L-asparaginase, known to have GLS-inhibitory activity. In aggressive NOTCH1-mutated human PDX-T-ALL models, the L-asparaginase-containing VXL regimen, in combination with IACS-010759, led to the metabolic and transcriptional collapse of T-ALL cells resulting in a significant reduction in tumor burden and prolonged survival. A recent report demonstrated enhanced combinatorial anti-tumor activity of ETC inhibition with dietary asparagine restriction with asparaginase, through depletion of exogenous asparagine that communicates active respiration to ATF4 and mTORC1[54]. While it remains to be seen if similar mechanisms are operational in leukemia, the synergy observed in our study could be broadly applicable to tumors beyond NOTCH1-mutated ALL.

In summary, our findings provide a comprehensive understanding of the role of oxidative phosphorylation and metabolic reprogramming in T-ALLs and offer a feasible translational strategy of OxPhos inhibition in clinical trials.

### Table 1 Panel of cell lines used in the study.

| Cell line | NOTCH1 status | Glucocorticoid sensitivity | Gamma secretase inhibitor sensitivity |
|---|---|---|---|
| JURKAT | Mutation | Resistant | Resistant |
| PF-382 | Mutation | Resistant | Resistant |
| 1301 | Mutation | Sensitive | Resistant |
| TALL-1 | Wild type | Resistant | Sensitive |
| LOUCY | Wild type | Resistant | Resistant |
| P12-ICHIKAWA | Mutation | Sensitive | Resistant |
| MOLT-3 | Mutation | Resistant | Resistant |
| MOLT-4 | Mutation | Resistant | Resistant |
| CCRF-CEM | Mutation | Sensitive | Resistant |
| KOPT-K1 | Mutation | Resistant | Sensitive |
| SUP-T1 | Wild type | Resistant | Sensitive |
| MOLT-16 | Wild type | Sensitive | Resistant |
| DND-41 | Mutation | Sensitive | Sensitive |
| ALL-SIL | Mutation | Sensitive | Sensitive |
| HPB-ALL | Mutation | Resistant | Sensitive |

### Table 2 Patient samples used in the study.

| UPIN | WBCs | % Blasts | NOTCH1 status | Other mutations |
|---|---|---|---|---|
| 6661738 | 4.6 | 50 | Mutation | SUZ12, FBXWT, KRAS, WT1 |
| 6666190 | 29.5 | 95 | Mutation | CEBPA, NRAS |
| 6506670 | 24.2 | 69 | Mutation | PTEN, SUZ12, TET2, U2AF1, WT1 |
| 6759914 | 102 | 53 | Mutation | JAK3, FBXW7, PHF6 |
| 6241832 | 2.3 | 60 | Mutation | EZH2, WT1, NF1, IL7R |
| 6818814 | 76.8 | 87 | Mutation | IL7R |
| 6818752 | 31.5 | 73 | Wild type | |
| 6070472 | 1.7 | 40 | Mutation | NRAS |
| 6033966 | 23 | 88 | Wild type | KRAS |
| 4081030 | 31 | 93 | Mutation | TP53 |
| 4186482 | 81 | 39 | Mutation | TP53 |
| 6126932 | 100 | 92.7 | Wild type | EZH, KRAS |

UPIN unique physician identification number, WBC white blood cell.

### Table 3 PDX samples used in the in vitro and in vivo studies.

| PDX ID | NOTCH1 status |
|---|---|
| PDX D115 | Mutation |
| PDX 12780 | Mutation |
| PDX CU178 | Mutation |
| MDX | Mutation |
| PDX CU76 | Mutation |
| PDX 80 | Mutation |
| PDX 22330 | Wild type |
| PDX 4911 | Wild type |
| PDX 1921 | Wild type |

## Methods

**Cell lines, primary T-ALL samples, and healthy T-lymphocytes**. The T-ALL cell lines: JURKAT, PF-382, 1301, TALL-1, LOUCY, P12-ICHIKAWA, MOLT-3, MOLT-4, CCRF-CEM, SUPT1 and KOPT-K1 were obtained initially from ATCC were maintained at an internal core facility of Institute of Science in Cancer MDACC and authenticated by short tandem repeat DNA fingerprinting in February 2016. MOLT-16 (ACC 29), DND-41 (ACC 525), ALL-SIL (ACC 511), and HPB-ALL (ACC 483), were purchased from DSMZ in March 2019. MS5 and MS5-DL4 stroma cell lines: Mouse stromal MS5 cells were kindly provided by K. Mori (Nagata University, Japan); VSV-G–producing cells transfected with the human DLL4 cDNA (MHS4426-98361330, GE Healthcare) were used for gene transduction in MS5 cells (48 h). All used leukemia cell lines are summarized in Table 1. Cell lines were maintained if not otherwise specified in RPMI-1640 medium with 2 mM Glutamine (SIGMA) containing 10–15% fetal calf serum and 1% penicillin-streptomycin (Life Technologies Laboratories).

Buffy coats were obtained from healthy blood donors provided by MD Anderson Blood Bank. Healthy bone marrow (BM) samples, BM and peripheral blood (PB) samples from T-ALL patients were collected during routine diagnostic procedures. Samples were collected after written informed consent was obtained from all patients and donors and was performed under the research protocols PA13-1025 (Targeting metabolic pathways in leukemia, for ALL), LAB02-295 and LAB04-0249 for healthy bone marrow and buffy coats from healthy blood donors, in accordance with the University of Texas MD Anderson Cancer Center Institutional Review Board regulations under the Declaration of Helsinki principles. The research protocols were approved by the respective Institutional Ethics Committees.

Mononuclear cells were separated by Ficoll-Hypaque (Sigma-Aldrich) density gradient centrifugation. $CD3^+$ T-lymphocytes were derived from buffy coats by $CD3^+$ magnetic beads isolation (Stem Cell Technology). Primary samples were maintained in QBSF medium containing 20% fetal calf serum (Gemini Bio-Products) and 1% penicillin-streptomycin (Life Technologies Laboratories). Patient characteristics are shown in Table 2. All cells were grown at 37 °C in a humidified atmosphere containing 5% carbon dioxide. D115 T-ALL PDX cells were kindly provided by Dr. Patrick Zweidler McKey. PDX CU76, PDX 12780, PDX CU178, PDX 2230. and murine T-ALL model (MDX) were kindly provided by Dr Adolfo Ferrando. PDX 4911 and 1921 were kindly provided by Dr. Daniel Lacorazza. PDX80 was generated in our lab from cells derived from a patient sample with the patient's written consent. The characteristics of the PDXs are shown in Table 3.

**Cell viability assay**. ALL-SIL, CCRF-CEM, DND-41, HPB-ALL, JURKAT, LOUCY, MOLT-3, MOLT-4, SUP-T1, PF-382, and TALL-1 cells, $CD3^+$ T-lymphocytes derived from healthy donors or bone marrow cells derived from healthy donors were plated at a density of 10,000 cells/well in complete RPMI-1640 (SIGMA) supplemented with 10% fetal calf serum and subjected to treatment with IACS-010759 (IACS, MD Anderson Cancer Center) at doses ranging from 0 to 123 nM and CB-839 at doses ranging from 0 to 30 μM, creating a 10 × 10 matrix of treatment. $CD3^+$ T-lymphocytes were additionally tested on DLL4 coated plates described in detail in DLL4 assay section and treated with IACS-010759. Each experiment was performed independently 3–5 times. Viable cell numbers were measured by quantifying ATP using a CellTiter-Glo Luminescent Cell Viability Assay (Promega). Dose-response curves were analyzed using a curve fitting routine based on nonlinear regression to compute the $IC_{50}$ value. The same

range of 10 drug concentrations was tested for each cell line to compute a curve-free area under the dose–response curve (AUC) based on linear interpolation of the 10 data points using a baseline of 0 (=100% inhibition) and a maximum of 100[55]. The AUCs generated for each cell line were further summarized based on NOTCH1 status.

To evaluate the impact of glutamine, glucose and pyruvate on ATP production, the cells were growing in RPMI1640 media without glutamine or glucose purchased from MP Biomedicals (no nutrients), that were supplemented with either 2 mM Gln, Glc at 5 mM or Pyr only at 1 mM; or with combination of these nutrients. Data were normalized to values obtained from triple supplemented media (Glc+Gln+Pyr). The results of five independent experiments were analyzed and presented as average ±SD.

To evaluate the synergistic effects of IACS-010759 with CB-839, vincristine, dexamethasone and L-asparaginase, cells were incubated in RPMI1640 (SIGMA) media supplemented with 2 mM glutamine. The Matrix of 10×10 was created for each combination and evaluated by CTG after 96 h of incubation. For synergy evaluation in combination with CB-839, IACS-010759 was used in the range of 0–123 nM, vincristine (MP Biomedicals) was used in the range of 0–50 ng/mL, dexamethasone (Sigma) in the range of 0–370 nM, and L-asparaginase from E. Coli (Sigma; Cat. Nr A3809) was used in the range of 0–5 IU/ml. For synergy evaluation in combination with VXL, IACS-010759 was used in the range of 0–50 nM, vincristine was used in the range of 0–5 ng/mL, dexamethasone in the range of 0–50 nM, and L-asparaginase in the range of 0–0.5 IU/ml. The results of five independent experiments were collected, processed, and visualized for synergy evaluation using COMBENEFIT software[56].

**Evaluation of cell numbers, apoptosis, and ROS**. The T-All cell lines ALL-SIL, CCRF-CEM, DND-41, HPB-ALL, JURKAT, LOUCY, MOLT-3, MOLT-4, SUP-T1, PF-382, and TALL-1 and $CD3^+$ T-lymphocytes derived from healthy donors, or bone marrow cells derived from healthy donors were plated at a density of $0.2 \times 10^6$ cells/mL in complete RPMI-1640 supplemented with 10% fetal calf serum and then treated with DMSO (control); IACS-010759 at concentrations of 1 nM, 5 nM, 10 nM, 30 nM, or 100 nM for 96 h; or CB-839 at 1 μM or VXL for 96 h. The effect of IACS-010759 treatment on cell number and apoptosis was evaluated by Annexin V (APC)-DAPI assay, after 96 h of exposure of the cells to the compound. The effect of IACS-010759 on ROS levels in viable cells was evaluated by staining cells with H2DCFDA. Briefly, after 96 h of drug treatment, cells were collected and washed twice in Annexin V binding buffer and resuspended in 100 μL of Annexin V binding buffer with Annexin V-APC antibody (BD Bioscience) at a

concentration of 1 μL/100 μL and H2DCFDA at a 1:1000 dilution and incubated in the dark at room temperature (RT) for 30 min. Cells were then washed with Annexin V binding buffer and resuspended after spinning in 300 μL Annexin V binding buffer containing DAPI and counting beads (Invitrogen) at 5 μL/300 μL. All conditions were run in triplicate and evaluated with Kaluza flow cytometry (FL) software or FlowJo software. The percentage of Annexin V-positive events was normalized to that in DMSO-treated controls and presented as the percentage of specific apoptosis, calculated as: (% Annexin V-positive cells in the sample−% Annexin V-positive cells in the control)/(100−% Annexin V-positive cells in the control) × 100. The number of living cells (DAPI-negative cells) was normalized to that in DMSO-treated controls. The mean fluorescence intensity of H2DCFDA, indicating ROS level, was evaluated in DAPI-negative, Annexin V-negative cells and normalized to that of DMSO-treated cells.

**Niche-based assay for pre-LSCs**. The niche-based assay was designed to sustain optimal pre-LSC survival ex vivo by mimicking physiological NOTCH1 signal strength and by adding KIT and FLT3 ligands and interleukin (IL)-7. Mouse MS5 stromal cells expressing or not expressing optimal levels of human DL4 were seeded in 96-well plates in the complete T-cell medium containing FLT3 ligand (5 ng/mL), KIT ligand (20 ng/mL), and IL-7 (5 ng/mL) (R&D Systems). The following day, purified CD4−CD8− thymocytes from 5–6-week-old preleukemic transgenic mice $LMO1^{tg}$ or $SCL^{tg}LMO1^{tg}$ were co-cultured on MS5 or MS5-DL4 monolayers for another 24 h prior to the addition of IACS-010759 at the indicated concentrations. Pre-LSC viability was determined after 72 h of drug treatment using multiparameter flow cytometry (Thy1+CD4−CD8−CD25+CD44−) with the VivaFix cell viability assay (BioRad) and the LSRII flow cytometer equipped with a high-throughput-screening module (BD Biosciences)[57]. Data acquisition was performed with a FACS Celesta flow cytometer equipped with a high-throughput-screening module (BD Biosciences). (Thy1+CD4−CD8−CD25+CD44−) Pre-LSC viability was normalized to that of control cultures containing the solvent alone (DMSO)[47].

**DLL4 assay**. Human DLL4 Fc Chimera Recombinant Protein (Thermo Fisher), was diluted in chilled phosphate-buffered saline (PBS) at 10 μg/mL as recommended[58], and 50 μL/well was coated in standard tissue-culture 96-well plates overnight at 4 °C. Wells were washed once with PBS before seeding cells to remove any unbound ligand from the wells. Healthy donor T-lymphocytes isolated from PBMCs as described in "Methods" were then suspended at the density of 30,000/well and seeded in the DLL4 coated plate or just washed with sterile PBS plates and further submitted to treatment with IACS-010759 at the concentration range 0-123 nM.

The same coating approach scaled up to 24 well format was used further for viable cell number/apoptosis evaluation by FL and for seahorse assay. Here shortly T-lymphocytes were seeded in complete RPMI1640 either on DLL4 coated wells or once PBS washed wells and subjected to treatment with IACS-010759 at the concentration of 10 nM in triplicated (for FL). The viability changes were then evaluated after 72 hr of incubation by FL. The impact of combined DLL4 and IACS-010759 on cell metabolism was evaluated after 24 h incubation by seahorse assay as described in the "Method" section for Seahorse assay.

**Colony-forming unit assay**. Healthy donor-derived fresh bone marrow cells were subjected to treatment with: DMSO, 10 nM IACS-010759, 1 uM CB839, VXL or a combination of IACS-010759/CB839 or IACS-010759/VXL for 24 h. Cells were then collected and counted, followed by preparing the suspension of $1\times10^5$ cells in 400 μl of complete RPMI-1640 (SIGMA), supplemented with 10% FBS in 15 ml Falcon tubes. 4 ml of MethoCult™ GF M3434 (Stem Cell Technologies; cat. 3434) was transferred to each Falcon tube using a 5 ml syringe with 18g1½ gauge needle and suspended in the methylcellulose by vortexing for 1 min. After 15 min of rest, 3 ml of methylcellulose were distributed between 3 bottom-grinded culture plates with aimed density of $2.5 \times 10^4$ cells/1 ml Methylcellulose/well) using a 3 ml syringe with 18g1½ gauge needle. Cells were allowed to grow for 12–14 days before colonies were manually counted.

**Analysis of OxPhos activity, glycolysis activity, and ATP using Seahorse technology**. To characterize the metabolic phenotype of T-ALL cells, T-lymphocytes or bone marrow cells derived from healthy donors, and to evaluate the impact of inhibition of complex I on mitochondrial metabolism, we measured OCR and ECAR in sets of T-ALL cell lines, patient samples, healthy T-lymphocytes, hBM cells and PDX samples ex vivo using the Seahorse Mito Stress Test, Glycolysis Stress Test, and ATP Rate Assay (Agilent Technologies). Each of the following cell lines: ALL-SIL, CCRF-CEM, DND-41, HPB-ALL, JURKAT, LOUCY, MOLT-3, MOLT-4, SUP-T1, PF-382, and TALL-1, as well as CD3+ T-lymphocytes and treated with IACS-010759 at doses of 0–123 nM for 4 h in complete RPMI-1640 media, followed by the appropriate Seahorse test for the specific scientific question.

In brief, cells were washed 2 times with phosphate-buffered saline (PBS) and resuspended in prewarmed (37 °C) Seahorse Basal Medium supplemented with 1 mM pyruvate, 2 mM glutamine, and 5 mM glucose, pH 7.4 (for Mito Stress Test); in prewarmed Seahorse Basal Medium supplemented with 2 mM glutamine, pH 7.4

(for Glycolysis Stress Test); or in buffered RPMI-1640 Seahorse Basal Medium supplemented with 1 mM pyruvate, 2 mM glutamine, and 5 mM glucose (for ATP Real-Time Rate Assay). Next, 175 μL of cells suspended at a density of 2 million/mL were plated in at least 4 replicates on 96-well Seahorse Cell Culture plates. To allow cells to attach to the bottom of the plates and create cell monolayers, the plates were precoated with Cell-Tak (Corning) according to the manufacturer's instructions. Once plated, cells were subjected to gentle centrifugation at RT for 4 min, then at 1500 rpm without a break, and then the centrifugation was repeated in the opposite direction.

OCR and ECAR were determined using a Seahorse XFe96 analyzer according to the manufacturer's instructions. The OCR and ECAR values were obtained at baseline (3 initial measurements) and after the injections of Seahorse XF Mito Stress Test Kit reagents: oligomycin (final concentration 1.5 μM), FCCP (final concentration 1.0 mM), and antimycin/rotenone (final concentration 0.5 μM); Glycolysis Stress Test Kit reagents: glucose (final concentration 10 mM), oligomycin (final concentration 1.5 μM), and 2-deoxy-D-glucose (final concentration 50 mM); or ATP Real-Time Rate Assay Kit reagents: oligomycin (final concentration 1.5 μM) and rotenone/antimycin (final concentration 0.5 μM). All measurements were carried out by the Mito Stress Test Generator, Glycolysis Stress Test Generator, or ATP Real-Time Rate Assay Generator (Agilent Technologies), as appropriate, and normalized to cell number after the assays. The data collected for basal and maximal OCR and ECAR after exposure to the range of drug concentrations were plotted to calculate the $IC_{50}$ values. AUC values were generated for each examined cell line and T-lymphocytes by $NOTCH1$-status. Further examination of OxPhos modulation by CB-839 and VXL was carried out by Mito Stress Test assay.

**Analysis of genome-wide chromatin occupancy by NOTCH1 in pre-LSCs**. Genome-wide chromatin occupancy by NOTCH1 was computed from chromatin immunoprecipitation-DNA sequencing data reported by Wang et al. for the murine T-ALL cell lines G4A2 and T6E[35]. The gene list was further restricted to those in which NOTCH1-bound peaks were present within 2 kb of their transcription start sites as previously described[36]. The expression of NOTCH1-bound genes within the OxPhos pathway during normal thymocyte differentiation was analyzed using ultra-low input RNA-seq data from the Immunological Genome Project IMMGEN (http://www.immgen.org/)[59]. $Notch1$-induced gene expression in DP T-cells was obtained by analysis of microarray data from purified murine CD4+CD8+GFP+ cells expressing or not expressing the $Notch1$ oncogene (ICN, GSE12948) at the preleukemic (2 weeks) and leukemic (6 weeks) stages (https://www.ncbi.nlm.nih.gov/geo/geo2r)[60]. Analysis of differentially expressed genes for GSEA and heatmaps (http://bioinformatics.sdstate.edu/idep/, https://www.gseamsigdb.org/gsea/index.jsp) was performed as described previously[36].

**RNA variant calling pipeline**. To detect short variants in RNA-seq, we ran the GATK RNA-seq variant calling pipeline (https://gatk.broadinstitute.org/hc/en-us/articles/360035531192-RNAseq-short-variant-discovery-SNPs-Indels-). We obtained Fastq files of RNA-seq data from the public T-ALL dataset (PRJNA572580; GEO Accession: GSE137768)[33]. Quality control and data pre-processing were done with fastp (version 0.20.0)[61]. We used the STAR aligner (version 2.6.0b) in two-pass mode aligned to the reference genome b37 (Homo_sapiens_assembly19.fasta)[62]. Duplicate reads were marked and removed by Picard MarkDuplicates (version 2.9.0). GATK (version 3.7) SplitNCigarReads was used to split and trim reads to remove any overhangs into introns. Base quality recalibration was performed with GATK (version 4.1.0.0), BaseRecalibrator, and APPlyBQSR. To correctly handle the splice junctions, the GATK Haplotypecaller was used to perform variant calling with a minimum phred-scaled confidence threshold of 20.0. We removed known single-nucleotide polymorphisms using dbSNP (version 138). We filtered variant call quality based on Fisher strand values greater than 30.0 and Qual by Depth values below 2.0 with GATK Variant Filtration. To annotate gene variants, we used ANNOVAR and combined each output table.

**DepMAP Crispr analysis**. Genome-scale CRISPR-Cas9 screen result was obtained at the DepMap Portal (https://depmap.org/portal/achilles) with all data (raw read counts to processed gene effect scores) available in the 19q4 DepMap dataset (FigShare: https://figshare.com/articles/dataset/DepMap_19Q4_Public/11384241/3)[63–65]. To determine the genetic dependencies that were enriched in T-ALL cell lines (HSB-2, PF-382, and SUP-T1), we utilized linear-model analyses from the limma R package[66] to perform a two-tailed t-test for the difference in the distribution of gene effect (dependency) scores in T-ALL cell lines compared to other cell lines screened. Statistical significance was calculated as a q-value derived from the p-value corrected for multiple hypothesis testing using the Benjamini & Hochberg method[67].

**RNA-Seq**. PF-382 and SUP-T1 cells were collected for RNA-seq when cells were harvested for mass spectrometry (MS) analysis. Cells from PDX80 were collected following short treatment studies with IACS-010759 and VXL as described. Total RNA was extracted from $4 \times 10^6$ cells using the RNeasy Mini Kit (Qiagen; cat. 74104). RNA concentration was determined by a NanoDrop

2000 spectrophotometer (Thermo Fisher Scientific), and RNA quality was assessed with an Agilent Bioanalyzer. Approximately 100 million reads per sample were run on an Illumina Nextseq 500 system using $75 \times 75$ paired end (PE) reads for stranded whole-transcriptome sequencing. Raw sequencing reads were pseudo aligned using Kallisto v0.44.059 with 30 bootstrap samples to a transcriptome index based on the human GRCh38.92 release (Ensembl). Abundance data were further analyzed with Sleuth v0.30.060 using models with covariates for condition. Gene-level abundance estimates were calculated as the sum of transcripts per million (TPM) mapped to a given gene. The data have been deposited in the NCBI's Gene Expression Omnibus (GEO) database (GSE167305) (https://www.ncbi.nlm.nih.gov/geo/query/acc.cgi?acc=GSE167305).

**NetBID analysis.** TPM-normalized count matrices from 3 cell lines (PF-382, SUP-T11, and PDX80), were log2 transformed and filtered for genes that were not detected in all samples (i.e., row sums > 0). To reduce the noise introduced by genes with low counts, we used an interquartile range method to rank the genes and selected the top 50% of the highly variable genes to run NetBID analysis on each cell line independently. This resulted in ~18 K genes for downstream analysis across 39 samples from the 2 cell lines PF382 and SUPT1 and 3 single drug-treated and 2 combination-treated cell lines (Vehicle, IACS-010759, CB-839, VXL, IACS-010759 + CB-839, and IACS-010759 + VXL) and 2 organs (spleen and BM) of PDX model T-ALL PDX80 treated with (Vehicle, IACS-010759, VXL, and IACS-010759 + VXL). Next, we merged all the gene sets from the Molecular Signatures Database (MsigDB) (http://www.gsea-msigdb.org/gsea/msigdb/) and calculated pathway activity in each sample using the mean expression level of genes in the gene sets; we then performed differential activity analysis at the pathway level. Significant pathways were selected at $p < 0.05$ or $p < 0.1$. We also applied the NetBID algorithm (https://github.com/jyyulab/NetBID) to the T-ALL RNA-seq data from the TARGET project and the COG AALL1231 trial to compare *NOTCH1*-mutant to -wild type samples.

**Mass spectrometry analysis of cell lines and PDX derived cells.** The *NOTCH1*-mutated T-ALL cell line PF-382 and the *NOTCH1*-wt T-ALL cell line SUP-T1 were cultured in T75 flasks at a density of 1 million cells/mL and then treated with DMSO, IACS-010759 (10 nM), CB-839 (1 μM), IACS-010759 + CB-839, VXL (1 ng/ml of vincristine, 10 nM dexamethasone, 0.1 IU/ml of L-asparaginase), or IACS-010759 + VXL, in RPMI-1640 media (MP Biomedicals) supplemented with 10% dialyzed fetal bovine serum (FBS; Life Technologies) with Gln and Glc; RPMI-1640 media supplemented with 10% dialyzed FBS with $^{13}C_5{}^{15}N_2$-glutamine (Cambridge Laboratories) and unlabeled Glc; or RPMI-1640 media supplemented with 10% dialyzed FBS with unlabeled Gln and $^{13}C_6$-glucose (Cambridge Laboratories). The *NOTCH1*-mutated cell line PF-382 was collected after 12 h of treatment, and the wild-type *NOTCH1* cell line SUP-T1 was collected after 24 h. Each condition was repeated in triplicate. After treatment, cells were centrifuged in 50 mL Falcon tubes, and cell pellets were washed twice in cold PBS. Cells were recounted, and cell pellets were flash-frozen in liquid nitrogen for further analysis.

For ex vivo metabolomic analyses, spleen and BM cells collected after organ harvesting were first subjected to engraftment analysis by FL, followed by counting and flash freezing in liquid nitrogen. Cell pellets were then subjected to mass spectrometry (MS) analysis. The metabolites were extracted using a modified Bligh-Dyer procedure by adding water, methanol, and chloroform in equal volumes (1:1:1). The resulting solution was vortexed vigorously and stirred at $1000g$ for 10 min, then centrifuged at $2119g$ for 20 min at 4 °C. The polar fraction was collected in Eppendorf tubes, evaporated to dryness in a Centri Vap refrigerated vacuum concentrator (Labconco) at 4 °C, and resuspended in 180 μL ultrapure water and 20 μL of internal standard solution, as previously described[68,69]. The 200-μL solution was filtered through a Nanosep 3 K ultracentrifugal device (Pall Co.) at 6010 rpm for 20 min at 4 °C[70]. The resultant filtrate was poured into liquid chromatography (LC) vial and stored at −20 °C until LC-MS analysis. Metabolomic analysis of polar fractions was performed on a Hybrid quadrupole-Orbitrap mass spectrometer (Q Exactive, Thermo Scientific) with a Thermo Scientific Accela 1250 ultra-high-performance LC system with an electrospray ionization source, simultaneously operating in positive/negative polarity-switching ionization mode. Metabolites were chromatographically separated using a Kinetex C18 150 × 2.1 mm (2.6 μm, 100 Å) column (Phenomenex) with gradient elution of 0.2% formic acid in water (A) and methanol (B) at a flow rate of 150 μL min$^{-1}$ within 30 min. The gradient elution was programmed as follows: 0–4 min, 2% B; 4–14 min, 2–80% B; 14–15 min, 80–98% B; 15–20 min, 98–98% B; 20–25 min, 98–2% B, equilibration time 5 min. Detection of metabolites was performed in full MS mode under the following conditions: spray voltage, 4.0 kV; capillary temperature, 300 °C; sheath gas, 50 (arbitrary units); auxiliary gas, 10 (arbitrary units); microscans, 1; AGC target, 3e6; maximum injection time, 200 ms; mass resolution, 70,000 full-width half-maximum; $m/z$ range, 50–750. To ensure mass accuracy below 5 ppm, the MS detector was calibrated prior to analysis using commercial calibration solutions. The LC-MS platform was controlled by XCalibur 2.2 software (Thermo Scientific). A quality control sample, representing the equivalent concentration of all samples, was prepared to control possible instrumental error (drift) in data acquisition and was run once every five samples[71]. All raw MS datasets were processed using Sieve 2.2 (Thermo Fisher Scientific), and features with a coefficient of variation lower

than 25% in the quality control samples were considered for further analysis. Peaks were scaled according to probabilistic quotient normalization, and features were then mined against an in-house database of accurate masses and retention times generated in our laboratory using the IROA 300 MS Metabolite Library of Standards (IROA Technologies). In addition, databases of accurate masses taken from the Kyoto Encyclopedia of Genes and Genomes and the Human Metabolome database were also mined[72,73]. Raw data are deposited in Zenodo open access repository (https://doi.org/10.5281/zenodo.6442444).

**Mass spectrometry analysis of peripheral blood.** A 10-μL whole blood sample was collected through the retroorbital sinus of the mice and immediately combined with 100 μL extraction solvent (ice-cold methanol:water = 80:20 with 10 μM d4-citrate internal standard), vortex-mixed for 2 min, and centrifuged at $20,000g$ for 10 min at 4 °C. The supernatant was transferred into 1.5-mL snap-cap tubes, dried using a Speedvac concentrator, and stored until further analysis. Upon analysis, the dried pellets were reconstituted in 100 μL ultra-pure water and centrifuged at $20,000g$ for 10 min at 4 °C. From the supernatants, 90 μL was transferred to polypropylene autosampler vials. A 10-μL volume of extract was injected into the ion chromatography (IC)-MS system without dilution. IC gradient and MS settings were as previously described[74]. The IC-MS platform included a Thermo Scientific Dionex ICS-5000+ capillary IC system with a Thermo IonPac AS11 250 × 2 mm 4 μm column. Data were acquired using a Thermo Orbitrap Fusion Tribrid Mass Spectrometer under electrospray ionization negative ionization mode. The raw files were then imported into Thermo Trace Finder software to extract chromatographic peak areas for targets of interest. Peak areas were normalized using internal standard peak areas. Raw data are deposited in Zenodo open access repository (https://doi.org/10.5281/zenodo.6442444).

**Ingenuity pathway analysis.** The values measured for each metabolite were calculated as expression log ratios of the following: GLS−/GLS+, GLS+IACS-010759/GLS+, and GLS−IACS-010759/GLS+ and uploaded to Ingenuity Pathway Analysis software (QIAGEN). The comparisons selected the most significantly affected canonical pathways.

**siRNA transfection.** *NOTCH1*-mutated T-ALL cell lines PF-382 and JURKAT ($1 \times 10^5$ cells per well) were seeded overnight. Negative control (NT-siRNA), KAG or GAC GLS-siRNA was transfected with jetPRIME transfection reagent, according to the manufacturer's instructions (Polyplus). The following plasmids were used: pHUSH puro D3; pHUSH puro KGA sh5; and pHUSH GAC sh3 (kindly provided by Dr. Georgia Hatzivassiliou, Genentech). For further studies puromycine selected cells were used. To induce knockdown of GLS, doxycycline at the concentration of 2 ug/ml was used. The detection of knockdown was evaluated at day 5 after initiation of treatment with doxycycline. The cells with confirmed knockdown were subjected to evaluate viability (CTG), cell growth, Seahorse analysis of metabolic changes with or without treatment with OxPhos inhibitor at the concentration of 10 nM, and to immunoblotting.

**Immunoblotting.** Cells ($1–2 \times 10^6$ per well) were seeded overnight in 6-well plates. Following treatment for the specified duration, total protein was extracted in Laemmli buffer, fractionated by sodium dodecyl sulfate-polyacrylamide gel electrophoresis and transferred to polyvinylidene fluoride membranes (Millipore Sigma). After blocking with Li-COR Odyssey blocking buffer, the membranes were incubated with primary antibodies overnight at 4 °C, washed, incubated for 2 h with infrared fluorochrome-conjugated secondary antibodies: Odyssey Irdye 680 RD anti-mouse and Odyssey Irdye 800 CW goat anti-rabbit, as appropriate. The protein signal was visualized using a LI-COR Odyssey imaging system. The panel of antibodies used for Immunoblotting is summarized in Supplementary Table 1.

**CYTOF.** Human T-ALL PDX 80 samples were collected from spleens. Samples were initially barcoded and stained with metal-conjugated antibodies according to Fluidigm's protocols as described below. Briefly, all antibodies were labeled with heavy metals using Maxpar-X8 labeling reagent kits (Fluidigm DVS Sciences) according to the manufacturer's instructions and titrated to determine the optimal concentration. For each mouse evaluated, $3 \times 10^6$ cells were aliquoted into separate FACS tubes and washed twice with Maxpar PBS buffer (Fluidigm; cat. 201058). For live/dead cell discrimination, cells were resuspended in 200 μL of 5 μM cisplatin (Fluidigm; cat. 201064) for 1 min on an orbital shaker, followed immediately by 3 washes in Maxpar cell staining buffer (CSB; Fluidigm; cat. 201068). Cells were then subjected for 10 min to fixation in 1 mL of 1× Fix I buffer (Fluidigm; cat. 201065), followed by 2 washes with 1× barcode perm buffer (Fluidigm; cat. 201057). Each unique palladium barcode (Fluidigm; Cell-ID 20-Plex Pd Barcoding Kit, cat. 201060) was suspended in 100 μL of barcode perm buffer and immediately transferred to cells in 800 μL barcode perm buffer. Cells were incubated with barcodes for 30 min, then washed twice with 2 mL CSB. Barcoded cells were then combined into a single tube for CyTOF staining. The staining factor was calculated as the total number of barcoded cells/3 × 10⁶ cells. Cells were blocked with 5 μL× staining factor of anti-human Fc receptor binding inhibitor (eBioscience Invitrogen) in 45 μL × staining factor of CSB for 15 min at RT. An appropriate amount of

**Table 4 Panel of antibodies used for CyTOF staining.**

| Isotope/Metal | Marker | Clone | Source | Catalog No. | Intracellular |
|---|---|---|---|---|---|
| $^{127}$I | S-phase | | Sigma | I7125-5G | NO |
| $^{139}$La | CD7 | CD7-6B7 | BioLegend | 343102 | NO |
| $^{142}$Nd | Caspase 3, cleaved | D3E9 | DVS-Sunnyvale | 3142004A | YES |
| $^{143}$Nd | PARP, cleaved | F21-852 | DVS-Fluidigm | 3143011A | YES |
| $^{144}$Nd | p-AKT | M89-61 | BD | 560397 | YES |
| $^{147}$Sm | p-STAT5(Y694) | 47 | DVS-Fluidigm | 3147012A | YES |
| $^{148}$Nd | CD34 | 581 | BD | 555820 | NO |
| $^{149}$Sm | p-4EBP1 | 236B4 | DVS-Fluidigm | 3149005A | YES |
| $^{150}$Nd | β-catenin, active | 8E7 | EMD | 05-665 | YES |
| $^{152}$Sm | p-H2AX, p-γ-H2AX | N1-431 | BD | 560443 | YES |
| $^{153}$Eu | Notch-1 | MHN1-519 | BioLegend | 352102 | YES |
| $^{154}$Sm | p21, WAF1/Cip1 | CP74 | Sigma | P1484 | YES |
| $^{156}$Gd | p-p38 (180/182) | D3F9 | DVS-Fluidigm | 3156002A | YES |
| $^{159}$Tb | p-Histone H3 | HTA28 | BioLegend | 641002 | YES |
| $^{160}$Gd | p-PI3K (p85/p55) | Polyclonal | CST | 4228BF | YES |
| $^{161}$Dy | Cytochrome C | 7H8.2C12 | Abcam | ab13575 | YES |
| $^{162}$Dy | mTOR | Polyclonal | GenScript | A01154 | YES |
| $^{163}$Dy | c-Myc | D84C12 | CST | 5605BF | YES |
| $^{164}$Dy | p-AMPKα (T172) | 40H9 | CST | 2535BF | YES |
| $^{165}$Ho | HIF-1α | Polyclonal | Novus | NB100-479 | YES |
| $^{167}$Er | p-ERK1/2 | D13.14.4E | DVS-Sunnyvale | 3167005A | YES |
| $^{168}$Er | Ki67 | (Ki67) X8 | DVS-Fluidigm | 3168001B | YES |
| $^{170}$Er | CD3 | UCHT1 | BioLegend | 300443 | NO |
| $^{174}$Yb | CD5 | L17F12 | BioLegend | 364002 | NO |
| $^{175}$Lu | p-S6 | N7-548 | DVS-Fluidigm | 3175009A | YES |
| $^{176}$Yb | Cyclin B1 | GNS-1 | BD | 554177 | YES |
| $^{195}$Pt | Cisplatin | | Sigma | P4394 | YES |
| $^{89}$Y | CD45 89Y | HI30 | DVS-Fluidigm | 3089003B | NO |

surface antibody master mix (Table 4) was added directly to the tube and incubated for 1 h at RT before 2 washes with CSB.

For intracellular staining, cells were dissociated in 1 mL of fixation/permeabilization buffer and incubated overnight at 4 °C, then washed twice with 2 mL of 1× permeabilization buffer. The supernatant from the final wash was discarded, and appropriate amounts of intracellular antibody master mix were added to the residual permeabilization buffer and incubated for 1 h in the dark at RT. For cell discrimination, a metallointercalator working solution (2 mL Permeabilization Buffer, 125 nM iridium metallointercalator [Fluidigm; cat. 201103 A]) was added to the sample and incubated overnight at 4 °C. Cells were washed with a minimum of 3 mL CSB, then again with 3 mL ddH₂O with 0.1% bovine serum albumin. Finally, cells were filtered through a 35-μm filter and resuspended in 100 μL ddH₂O with 0.1% bovine serum albumin. The MD Anderson Flow Cytometry and Cellular Imaging Core Facility prepared the antibody cocktails and acquired data on a Helios CyTOF machine (DVS Sciences).

Data were first demultiplexed by Fluidigm Debarcoder software. Individual mass cytometry data files (.fcs) were then filtered using FlowJo to remove the normalization beads, debris, doublets, and dead cells. The remaining analysis was performed in R (version 3.6.1, The R Foundation for Statistical Computing) using the R packages 'cytofkit' and 'flowcore'[75]. Processed data were subjected to negative value pruned inverse hyperbolic sine transformation and clustered based on the PhenoGraph algorithm ($k = 22$) using all cell surface markers[76]. Dimensionality reduction was performed using the uniform manifold approximation and projection (UMAP) method[77].

**Animal studies.** For the studies of pre-LSCs, all mouse lines were backcrossed onto a C57BL/6J background for more than 12 generations to produce *pSTIL-SCL* (*SCL*tg), *Lck-LMO1* (*LMO1*tg), and *Lck-NotchIC9* (*Notch1*tg) mice (NIAID/Taconic Repository). Mice (male and female) were maintained separately in pathogen-free conditions according to institutional animal care and use guidelines set by the Canadian Council on Animal Care. At least 3 mice per genotype were used for experimental work. For the studies of T-ALL, all experimental animal procedures were approved by MD Anderson Cancer Center's Institutional Animal Care and Use Committee (IACUC). The study was compliant with all relevant ethical regulations regarding animal research. Animal studies were conducted at MD Anderson's animal facilities in accordance with the IACUC guidelines. Mice were maintained in a pathogen-free environment with free access to food. 8-week-old female C57BL/6 mice were purchased from The Jackson Laboratory and allowed to acclimatize for 1 week before being irradiated with a sublethal dose of 400 cGy on the day before T-ALL cell injection. To examine the impact of complex I inhibition in a genetic model of glutaminase knockout, we used *NOTCH1*-induced T-ALL

murine cells with tamoxifen-inducible conditional *Gls* knockout (*Rosa26*$^{Cre-ERT2/+}$*Gls*$^{f/f}$), which were generated and kindly provided by Drs. Daniel Herranz and Adolfo Ferrando (Columbia University). A total of $1 \times 10^6$ cells suspended in 100 μL sterile PBS were injected into the tail veins of the female C57BL/6 mice[29]. Disease progression was monitored weekly by retroorbital blood draws, and measurements of GFP⁺ leukemic cells in the PB were normalized to the total amount of murine CD45⁺ cells, after exclusion of DAPI⁺ (dead) cells. Once the leukemia cells were detectable in PB (day 7, Figure S22A, B), mice were randomized into 2 treatment groups ($n = 10$ mice per group). One group underwent conditional knockout of GLS by administration of 1 mg tamoxifen over 5 consecutive days (GLS⁻), and the other was treated with corn oil (GLS⁺). To measure the additive or synergistic effects of complex I inhibition, GLS⁻ and GLS⁺ mice were treated with either vehicle or IACS-010759 (7.5 mg/kg) ($n = 5$ mice per arm) by oral gavage once daily on a schedule of 5 days on/2 days off starting at Day 7 up to Day 68. Disease progression and response to treatment were measured by weekly engraftment checks. Mice were euthanized when visible signs of clinical illness were observed or weight loss of >20% was recorded. To measure the short-term impact of complex I inhibition and glutaminase knockout, mice ($n = 3$ per treatment arm) engrafted at a level of 5% of circulating leukemia cells in PB were subjected to the same treatment regimen for 5 days, followed by mice euthanasia by cervical dislocation, organ harvesting and measurement of circulating leukemia cells in PB and the level of leukemic cell engraftment in the spleen, BM, and liver. Tissues collected after the treatment were subjected to histopathological examination, and blood collected after treatment was further subjected to MS analysis.

To examine the impact of complex I inhibition in combination with standard-of-care chemotherapy, 8- to 10-week-old female NOD.Cg-PrkdcscidIl2rgtm1Wjl/SzJ (NSG) mice were purchased from The Jackson Laboratory (cat # 005557) and allowed to acclimatize for 1 week. After irradiation on the day prior to injection, each mouse was injected via the tail vein with $1.0 \times 10^6$ T-ALL PDX cells suspended in 100 μL sterile PBS; the PDX models used were CU76, PDX D115, and PDX80. Disease progression was monitored weekly by retroorbital blood draws, and FL was used to monitor the percentage of human CD45 + cells normalized to that of murine CD45+ cells after exclusion of DAPI+ (dead) cells. Treatment was initiated at detection of 1% to 5% circulating leukemia cells in the blood ($n = 10$ mice per arm). IACS-010759 was administered via oral gavage once daily at 5 mg/kg (5 days on/2 days off). Standard-of-care chemotherapy (VXL) was administered intraperitoneal once per week, 8 h after administration of IACS-010759, for 4 weeks. VXL was used at a reduced dose of: dexamethasone (5 mg/kg), vincristine (0.15 mg/kg); and L-asparagine (1000 IU/kg, approx. 25 IU/mouse). E. coli-derived L-asparaginase was purchased from Elspar, Lundbeck, Deerfield, IL, USA.

To measure the short-term impact of complex I inhibition on leukemia metabolism and VXL sensitivity, mice engrafted at a level of 30–50% circulating leukemia cells in PB were subjected to the treatment regimen above for 5 days

($n = 4$ mice per PD treatment arm). 12 h after the last dose of IACS-010759, mice were euthanized by cervical dislocation, followed by harvest of PB, spleen, BM in order to measure circulating leukemia cells in PB and the level of leukemic cells in spleen and BM. Tissues of spleen, BM and liver collected after treatment were subjected to histopathological examination, blood collected after treatment was subjected additionally to MS analysis, and spleen cells collected after treatment were flash-frozen in liquid nitrogen for MS analysis and for RNA extraction followed by RNA-seq, or frozen in 10% DMSO/90% fetal calf serum for further CyTOF analysis.

**Histopathology evaluation**. Five-micron-thick sections of tissues were cut using a Leica Microsystems cryostat, transferred onto Superfrost-Plus slides (Thermo Fisher Scientific), and stained with hematoxylin and eosin (H&E). BM, spleen, and liver tissue samples were further evaluated by pathologist, blinded to the treatment groups, for determining the leukemic infiltration of tissues and morphologic changes of hematopoiesis. Stained tissue slides were examined microscopically using Olympus BX 41 microscope and by scanning the slides with Aperio AT2 scanner and viewing the images with Aperio Image Scope software.

**Statistical analysis**. Unless otherwise indicated, all calculations and statistical analyses were carried out using GraphPad Prism software v. 7 or 8. Figure legends indicate specific statistical analyses used and define error bars. Statistically significant differences between 2 groups were assessed by unpaired Student's $t$ tests. Ordinary one-way analysis of variance (ANOVA) was used to analyze more than 2 groups. Two-way ANOVA was used to analyze cell proliferation at multiple time points. Results are expressed as mean ± SD (as noted in the figure legends) of at least 3 independent experiments, and $p$ values less than 0.05 were considered statistically significant. Survival analysis for in vivo studies was done using Mantel–Cox and Gehan–Breslow–Wilcoxon tests.

**Reporting summary**. Further information on research design is available in the Nature Research Reporting Summary linked to this article.

## Data availability

All data supporting this study are provided in the manuscript or made publicly available.

ChIP-seq data published previously by Wang et al. were used for computation of genome-wide chromatin occupancy by NOTCH1 and are available under accession code GSE29600.

Ultra-low input RNA-seq data from the Immunological Genome Project IMMGEN are available under accession code GSE100738 (http://www.immgen.org/).

Microarray data from purified murine CD4+CD8+GFP+ cells expressing or not expressing the Notch1 oncogene at the preleukemic (2 weeks) and leukemic (6 weeks) stages are available under accession code GSE12948 (https://www.ncbi.nlm.nih.gov/geo/geo2r).

Genome-scale CRISPR-Cas9 screen result was obtained at the DepMap Portal (33T33T https://depmap.org/portal/achilles33T) with all data (raw read counts to processed gene effect scores) available in the 19q4 DepMap dataset (FigShare: https://figshare.com/articles/dataset/DepMap_19Q4_Public/11384241/3).

RNA-seq data generated in this study have been deposited in the NCBI's Gene Expression Omnibus (GEO) database (GSE167305).

The gene sets from the Molecular Signatures Database (MsigDB) (http://www.gsea-msigdb.org/gsea/msigdb/) were merged and used for pathway activity calculations using NetBID in each sample using the mean expression level of genes in the gene sets.

Data may be accessed through the TARGET website at https://ocg.cancer.gov/programs/target. The sequencing BAM and FASTQ files from TARGET RNA-seq are accessible through the database of genotypes and phenotypes (dbGaP; http://www.ncbi.nlm.nih.gov/gap) under accession number phs000218 (TARGET) and substudy specific accession phs000464 (TARGET ALL Expansion Phase 2).

Fastq files of RNA-seq data from the public T-ALL dataset from the frontline Children's Oncology Group (COG) T-ALL clinical trial AALL1231 are deposited under accession code (PRJNA572580; GEO Accession: GSE137768)

Metabolomics data are deposited in Zenodo open access repository (https://doi.org/10.5281/zenodo.6442444)

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

## Acknowledgements

Supported in part by grants from the Cancer Prevention and Research Institute of Texas (CPRIT) and the National Institutes of Health (NIH) R01CA231364, Leukemia SPORE P50 CA100632 and CPRIT RP180309 to M.K. and S.T., R01GM134382 to J.Y. MD Anderson Cancer Center's Cancer Center Support Grant (P30CA016672) from the NIH/ National Cancer Institute supports the Flow Cytometry and Cellular Imaging Facility, the RNA Sequencing Core Facility, and the Metabolomics Core Facility. The Metabolomics Core Facility is also supported by CPRIT grant RP130397 and NIH grant S10OD012304-01. This work was also supported by research grants from the Canadian Cancer Society (T.H., Grant # 704867), the Oncopole-Cancer Research Society-Fonds de recherche du Québec (T.H. # 265879), the Canadian Institute for Health Research (T.H., PJT–148943), Genome Canada and Génome Québec (G.S.), and a fellowship from the Cole Foundation (D.T.V.). This work benefited from data assembled by the IMMGEN consortium. We thank Amy Ninetto, Scientific Editor, Research Medical Library, MD Anderson Cancer Center, for editing the manuscript. We thank Craig Smith and Jay Dunn from Agilent for technical support with Seahorse experiments.

## Author contributions

N.B., T.H., and Ma.Ko conceived and designed the study; N.B., A.L., Y.D., M.C., S.S., R.P., S.T., Me.K., S.S.S., A.S., Sh.P., M.T., V.M.K., A.C., K.H., N.F., J.G., M.E.D.F., J.R.M., M.O.W., D.D., P.L.L., R.E.D., M.D., L.M.G., G.A.-A., K.R., Su.P., H.M., V.R., S.R.-S., A.H., D.T.V., Y.G., A.M., G.S., T.H., D.H., A.F., M.G. performed experiments; N.B., A.L., S.S., R.P., M.C., S.T., S.H., Y.D., J.Y., D.T.T., T.M.H., F.W.H., S.K., K.R., K.F., Ko.T., N.F., J.G., M.E.D.F., J.R.M., W.Y., J.J.Y., E.J.J., S.R.-S., A.H., D.T.V., Y.G., A.M., G.S., P.L.L., T.H., M.G., Ka.T., Ma.Ka., M.A. analyzed and interpreted data; N.B., T.H., and Ma.Ko wrote the manuscript; N.B., A.L., S.T., R.E.D., T.H., D.H., A.F., J.R.M., Ma.Ko. revised the manuscript, Ma.Ko., S.T., T.H. supervised work, directed the study, provided advice on experiments, all authors commented on the manuscript.

## Competing interests

The authors declare no competing interests.

## Additional information

[1]Department of Leukemia, The University of Texas MD Anderson Cancer Center, Houston, TX, USA. [2]Department of Nutritional Sciences, Dell Pediatric Research Institute, Dell Medical School, The University of Texas at Austin, Austin, TX, USA. [3]St. Jude Graduate School of Biomedical Sciences, St. Jude Children's Research Hospital, Memphis, TN, USA. [4]Institute for Research in Immunology and Cancer, The University of Montreal, Montréal, QC, Canada. [5]Department of Stem Cell Transplantation and Cellular Therapy, The University of Texas MD Anderson Cancer Center, Houston, TX, USA. [6]Department of Bioinformatics and Computational Biology, The University of Texas MD Anderson Cancer Center, Houston, TX, USA. [7]The Jackson Laboratory for Genomic Medicine, Farmington, CT, USA. [8]Department of Oral and Maxillofacial Surgery, Hirosaki University Graduate School of Medicine, Hirosaki, Aomori, Japan. [9]Department of Pharmaceutical Sciences, St. Jude Children's Research Hospital, Memphis, TN, USA. [10]Department of Cancer Systems Imaging, The University of Texas MD Anderson Cancer Center, Houston, TX, USA. [11]TRACTION Platform, Therapeutics Discovery Division, University of Texas M. D. Anderson Cancer Center, Houston, USA. [12]Department of Immunology, St. Jude Children's Research Hospital, Memphis, TN, USA. [13]Department of Translational Molecular Pathology, The University of Texas MD Anderson Cancer Center, Houston, TX, USA. [14]Department of Lymphoma and Myeloma, The University of Texas MD Anderson Cancer Center, Houston, TX, USA. [15]Rutgers Robert Wood Johnson Medical School, Cancer Institute of New Jersey, New Brunswick, NJ, USA. [16]Irving Cancer Research Center, Columbia University Irving Medical Center, New York, NY, USA. [17]Institute for Applied Cancer Science, The University of Texas MD Anderson Cancer Center, Houston, TX, USA. [18]Perelman School of Medicine, The University of Pennsylvania, Philadelphia, PA, USA. [19]Texas Children's Cancer Center, Baylor College of Medicine, Houston, TX, USA. [20]Department of Veterinary Medicine and Surgery, The University of Texas MD Anderson Cancer Center, Houston, TX, USA. [21]Department of Computational Biology, St. Jude Children's Research Hospital, Memphis, TN, USA. [22]Department of Pharmacology and Physiology, Faculty of Medicine, University of Montreal, Montreal, QC, Canada. ✉email: mkonople@mdanderson.org

