## [Peer Review File · Nature Communications]

Inhibition of mitochondrial complex I reverses NOTCH1-driven metabolic reprogramming in T-cell acute lymphoblastic leukemiaREVIEWER COMMENTS

Reviewer #1 (Remarks to the Author):

In this manuscript, Baran and colleagues investigate the role and therapeutic targeting of oxphos in T-ALL. This manuscript has several parts of data that are new and interesting, but also a number of problems that need to be addressed. Specific critiques:

1. In most of Fig 1, the authors use gene expression analyses to argue that NOTCH1 regulates “OxPhos function”, by which they mean expression of genes encoding oxphos components. They then use DepMap data (Fig 1H, I) to argue that some oxphos genes are selectively depleted in T-ALL, however there are even more oxphos genes that have the inverse association in both comparisons – they are selectively depleted in all other cancers more than T-ALL. If all oxphos genes in DepMap are considered together, is there any significant association with T-ALL? At this point in the paper the case that oxphos is selectively linked to T-ALL is quite weak. I recommend several changes to address this: i) the suggestion that gene expression is a measure of “OxPhos function” (2nd line of 2nd paragraph of p. 6) should be rephrased, ii) the conclusion that these data implicate oxphos function in T-ALL should be weakened (“suggest” would be OK), and iii) I recommend removing the DepMap data entirely unless a global analysis of all oxphos genes shows a significant association with T-ALL, or the authors can somehow reconcile why so many oxphos genes have the inverse association.

2. Fig 2A: These data are very interesting, but I would like to be able to tell whether the drug kills MS5-DL4 treated cells, or whether DL4 treatment increases cell growth/proliferation and the drug reduces their proliferation back to baseline. I can think of several ways this could be addressed: by showing absolute cell counts over time (do cell counts in MD5-DL4 cells drop below initial starting cell counts, or did the drug just stop proliferation), or cell cycle state and apoptosis markers (if cell death is apoptotic) could be assessed to distinguish these.

3. Fig 2B: These data are plotted in a way that is very difficult to interpret. What precisely is plotted in the X axis, is this relative change in AUC? I cannot tell from relative changes in AUC how drug-sensitive the cells are at clinically relevant concentrations. Are these data taken from curves such as in Fig 2A? If so I would much prefer to see the curves themselves, these are much more informative.

4. A second point related to Fig 2B is that most mouse T-ALLs induced by Scl acquire NOTCH1 mutations during oncogenic transformation (<https://pubmed.ncbi.nlm.nih.gov/16166587/>), this may be the case in the other models as well. However, the authors suggest that T-ALLs lacking a NOTCH1 transgene are NOTCH1 wild-type (the authors refer to “the genetic status of NOTCH1” on p. 6) which seems questionable. It may be that the NOTCH1 transgene drives stronger NOTCH1 activation than the acquired NOTCH1 mutations in the other cases or maybe they truly are NOTCH1 wild-type, either way I would like to see Notch1 activity measured directly. I think the simplest way to address this would be by comparing ICN1 levels by Western blot in the various leukemias, cultured on MS5 vs MS5-DL4. This will allow the authors to test a prediction from their model that leukemias lacking a NOTCH1 transgene should have much lower ICN1 levels than the NOTCH1 transgenic leukemias, but only when cultured on MS5 cells and not MS5-DL4.

5. Fig 3A: I believe the “no nutrients” is only glutamine, glucose and pyruvate removed but all other nutrients in the basal media (RPMI-1640) of these cells present, this should be clarified. The product number of the specific RPMI-1640 medium used should also be listed in the methods, there are different versions with glutamine concentrations that vary widely.

6. On p. 10 the authors argue that the “elevated aKG/citrate ratio indicates utilization of reductive metabolism”, but this conclusion i) needs supporting references, and ii) seems too strong, I suggest rephrasing to an elevated ratio “has been associated with” a switch to reductive metabolism (unless there are references I am missing).

7. Fig 4B: I simply cannot understand the conclusions the authors draw in the text based on Fig 4B. On p. 11 the authors state that “the relative sensitivity of NOTCH1-mutant cells to IACS was increased when glutamine was provided”, however the cells are actually LESS drug sensitive when glutamine is added compared to no nutrients (compare blue vs pink curve). They then say that this sensitivity “further improved with Pyr addition”, however Pyr has biphasic effects – Pyr + Gln (orange curves) cells are more sensitive than Gln (blue) cells at low drug doses, but less sensitive at high doses, which is a difficult-to-interpret result that is ignored here. It is also not clear what the authors mean by “improved”, do they mean more or less drug sensitive? This part of the paper comes perilously close to being a fatal flaw for the entire paper, it needs to be rewritten entirely to clarify how the authors are interpreting these results, and to avoid stating conclusions that are not supported by the data.

8. The authors then move on to test genetic inhibition of glutamine synthase in combination with oxphos inhibition, which looks promising but there is a major caveat of the approach which should be stated in the text - GLS is genetically inhibited in the leukemic cells and not in the rest of the mouse. I would also like for the authors to test whether there is a therapeutic index for GLS inhibition + IACS – is this combination any less toxic to normal hematopoietic cells? Likewise, is VXL + OxPhos inhibition less toxic to normal hematopoietic cells than leukemic cells?

9. One rationale the authors give for using asparaginase is that it has glutaminase activity, however this is quite modest for the most common clinical asparaginase from *E. coli* (<https://pubmed.ncbi.nlm.nih.gov/17011787/>). Which form of asparaginase did the authors use, and does this lead to depletion of glutamine levels when treating leukemic cells? This can be assayed using the clinical assay for serum amino acid quantification in the supernatant of treated cells. If glutamine is not depleted, the rationale for using asparaginase will need rethinking.

I also recommend additional changes to the text:

1. This sentence on p.4 of the introduction is problematic: “Unlike solid tumors, which often depend on aerobic glycolysis for their metabolic needs, leukemia cells depend on oxidative phosphorylation (OxPhos) to meet their energetic and biosynthetic demands”

The sentence suggests that dependence on glycolysis vs oxphos are mutually exclusive, but most cancers are dependent on both at least to some degree. Simply starting with “Leukemia cells depend on oxphos...” would avoid this problem.

2. What “VXL” chemotherapy is needs to be defined in the text

Reviewer #2 (Remarks to the Author):

The authors identify OxPhos as a critical pathway for T-ALL cell survival and demonstrate a direct relationship between NOTCH1, OxPhos activity and acquired chemoresistance in pre-leukemic and leukemic models. The link with NOTCH1 is not clearly shown.

The authors inhibit OxPhos with IACS-010759, an inhibitor of mitochondrial complex I, which caused growth inhibition in NOTCH1-mutated T-ALL cells. The authors study the mechanism of metabolic reprogramming in detail. In addition it is shown that IACS-010759 combined with L-asparaginase reduced glutaminolysis and acted in synergy to inhibit leukemia cell growth.

General comments:

T-cell acute lymphoblastic leukemia (T-ALL) is not commonly associated with relapse and chemoresistance as described in the first sentence of the abstract. T-ALL is indeed associated with a higher risk than many other B-ALL subtypes, but relapse is not very common in T-ALL.

The authors focus on NOTCH1 mutant T-ALL and seem to suggest that the findings are unique for

NOTCH1 mutant T-ALL, but do not give evidence for this by comparing the findings in NOTCH1 mutant versus NOTCH1 wild type T-ALL.

Are the findings specific for NOTCH1 mutant T-ALL or is it more general ?

The text is very dense and the figure legends are rather poor, so that it is not easy to understand what the authors do on each figure (which samples used, which technique used, what is compared)

There are references lacking in the text, for example: "glutaminolysis is another metabolic feature of T-ALL and AML that is crucial for anaplerosis...", page 4, "part of the efficacy of L-Asp against T-ALL relies on its GLS-I activity", page 5...

The representation of the data is not consistent: the authors analyze the data in a different ways on each figure / figure panel (sometimes IC50, sometimes AUC, others normalized AUC ...). Normally AUC is used when the number of doses are limited, and that is not the case here, so it seems better to use always IC50.

The authors indicate that current NOTCH1 inhibitors can not be used due to their toxicity, but they don't check the toxicity of the drugs that they use.

There are some overstatements in the text (for example the use of "directly linked" without clear confirmation). Overall, the text would benefit from some adjustments to correct for this.

Specific points:

Figure legends 1B and 1C are switched

Figure 1 H and I: they used previous datasets of Crispr Cas9 screen but it is not clear what the authors are comparing here.

Figure 2B: not a clear way of showing the data, seems the authors are picking the statistics to show the best differences.

Figure 2 C-K: It is not clear if there is any specificity for NOTCH1 mutated T-ALL

page 7, at the end: "IC50 was not reached"? IC50 is always reached (if the concentration range is high enough), in fact the numbers are shown in the graph.

Figure 3A: viability measured by ATPlite. Since the authors show differences in metabolism, they could validate these effects on viability by viability dye or other method not related to mitochondria.

Figure 4B: The synergy between IACS and Gln or Glc/Gln or no-nutrients is not clear. The curves are parallel or even the inhibitor shows less effect than control medium. The major effect seems to be the nutrients, not the drug. Only the yellow curve shows this increased sensitivity to IACS.

Figure 5: the clear difference is in survival, I don't see the value of the measurements at day 5 (no differences in combi)

Figure 5 I: the graph is quite dense, the errors are bigger than the curves.

Figure 6: I don't see the added value of single cell sequencing here. Unless the authors compared NOTCH1 mutant and NOTCH1 wild type cells.

Figure 6 E-G: not clear in the figure/legends what is shown here in each column (write PDX names, mutations...).

Figure 6: the authors picked L-Asp for its GLS activity to check synergy with oxphos inhibitor, but they don't show any proof of the mechanism of the synergy being mediated by this activity. Or show there is no synergy with other chemo drugs without GLS activity?

Reviewer #3 (Remarks to the Author):

In this manuscript authors aim to identify and target metabolic vulnerabilities in T-cell acute lymphoblastic leukemia (T-ALL), driven by NOTCH1 mutation. The research area is current and potentially of clinical importance and is of interest to the reader of Nature Communication. Initially the authors use bioinformatic approaches to demonstrate a link between NOTCH1 status and increased mitochondrial metabolism/oxidative phosphorylation (OXPHOS). The authors then apply clinically relevant and potent Complex 1 inhibitor (IACS-010759) and show that T-ALL cells are sensitive to OXPHOS inhibition. Using nutrient deprivation assays, coupled with metabolic assays, the authors next show that T-ALL cells induce glutamine metabolism following IACS-010759 treatment. Finally, the authors show that the combination of OXPHOS inhibition (IACS-010759) and inhibition of glutamine metabolism (GLS deletion or L-asparaginase treatment) is effective against T-ALL in vitro and in vivo.

The main strengths of the paper are the application of different relevant models to address the research questions, coupled with challenging functional assays (i.e., Seahorse), transcriptomics and isotope-assisted metabolomics. In most part, the data is convincing and the effect of IACS-010759, either as a single agent, or in combination with glutamine metabolism inhibition is impressive. The main weaknesses of the manuscript are that it is very data heavy (30 supplementary figures), with some of the assays poorly explained (i.e., complex nutrient tracing experiments are poorly explained) reducing the clarity/focus of the manuscript. It lacks novel mechanistic insight; IACS-010759 has already been shown to be very potent OXPHOS inhibitor and it is not unexpected that glutamine metabolism is induced following Complex 1 inhibition. Additionally, the concerns regarding the potential toxicity when OXPHOS and glutamine metabolism is combined is not addressed sufficiently.

Below I have listed some of the specific concerns/comments I have.

1. Page 5: Provide reference for "Also, part of the efficacy of L-asparaginase against T-ALL relies on its GLS-i activity". L-asparaginase does not, as far as I know, have GLS-inhibitor activity, it degrades glutamine on top of asparagine.
2. Figure 2: some of the differences between wt and mutant are not hugely convincing.
3. Figure 2: Does DL4 stimulation increase OXPHOS transcriptionally (or OCR by Seahorse)?
4. Figure 2c-d: It would be informative to present the seahorse profiles.
5. Figure 2k: Does ROS have a functional role (do ROS scavengers block some of the effects)?
6. Figure 3a-c: Is pyr-gluc and pyr-gln mixed up (latter looks more effective)? Also, the interpretation of some of these experiments are not clear. "The addition of Pyr with Gln further improved ATP generation to 80%" is that correct? "while NOTCH1-mutants being dependent on Gln to activate OxPhos and generate ATP" is that correct?
7. Figure 3b: Gln addition does not significant increase OCR – explain/discuss.
8. Is 11mM glucose, 2mM glutamine and 1mM pyruvate used throughout (which is much higher than blood, ca 5.5mM, 550-650µM, 50-100µM)? This needs to be discussed.
9. Figure 4b, have IC50's been calculated and are they backing up claims made in paper (I don't see a clear shift)?
10. Figure 4h: It would be better to have untreated single arms and combo for these experiments. Some decreases were also seen in figure 3k (IACS only).
11. For animal experiments, as well as for ex-vivo LCMS: was there a difference in viability of cells between arms? If there was a lot of death, then LCMS would be affected.
12. Figure 6: Here it would be informative to see what effect media without asparagine, glutamine or both would be (Vxl will deplete both and it would be good to see if one or both contribute to observed phenotypes).
13. Note, the Seahorse measured OCR, not ATP levels, so better to refer to ATP-linked OCR.
14. "The exposure to Glc in the presence of Gln switched NOTCH1-mutated cells from OxPhos

towards glycolysis" I guess this is expected?

15. Page 11: "These results suggest that the maintenance of OxPhos function depends on Gln in NOTCH1-mutated cells and that Gln is crucial for the sensitivity to OxPhos-i (P=0.0001, Figures 4C; S11A-B)." – So what was the rationale for inhibiting GLS?

16. Discuss results in relation to Nguyen et al, Downregulation of Glutamine Synthetase, not glutaminolysis, is responsible for glutamine addiction in Notch1-driven acute lymphoblastic leukemia, Mol Oncol, 2020.

17. Page 13: "Consistent with in vitro observations, metabolomic analysis of PB from mice" clarify if this is leukaemic or normal cells?

18. Provide reference for "Also, part of the efficacy of L-asparaginase against T-ALL relies on its GLS-i activity."

19. Fig S11A-B: Do JURKAT and PF not have Notch mutation?

Reviewer #4 (Remarks to the Author):

In this paper, the authors investigate the molecular mechanisms by which activating mutations in NOTCH1 associates with T-ALL outcome and therapy. The article is well-written and provides new knowledge that is important from both a scientific and clinical viewpoint. The claims that are being made are generally well-supported by data and I only have minor points for improvements:

- In the abstract, the following sentence should be adapted as now it appears the treatment combinations results in lethality in the mice (and not in the T-ALL cells): "Thus, pharmacological blockade of OxPhos combined with inducible knockdown of the key glutamine enzyme glutaminase confers synthetic lethality in mice harboring NOTCH1-mutated T-ALL".

- IACS-010759 was used to inhibit complex I but I missed the evidence that this indeed happened as expected. Showing complex I inhibition (e.g. be PMP seahorse) should be done and the authors should also take into account that this compounds can have off-target effects. This could be experimentally ruled out and/or discussed.

-C57VL6 -> C57BL6?

-rephrase 'deeper reduction'

REVIEWER COMMENTS

Reviewer #1 (Remarks to the Author):

1. In most of Fig 1, the authors use gene expression analyses to argue that NOTCH1 regulates “OxPhos function”, by which they mean expression of genes encoding oxphos components. They then use DepMap data (Fig 1H, I) to argue that some oxphos genes are selectively depleted in T-ALL, however there are even more oxphos genes that have the inverse association in both comparisons – they are selectively depleted in all other cancers more than T-ALL. If all oxphos genes in DepMap are considered together, is there any significant association with T-ALL? At this point in the paper the case that oxphos is selectively linked to T-ALL is quite weak. I recommend several changes to address this:

i) the suggestion that gene expression is a measure of “OxPhos function” (2nd line of 2nd paragraph of p. 6) should be rephrased,

ANSWER: We thank the reviewer for this valuable suggestion and changed “OxPhos function” to “OxPhos activity”

ii) the conclusion that these data implicate oxphos function in T-ALL should be weakened (“suggest” would be OK), and

ANSWER: We appreciate this suggestion and weakened our statement to: “possibly linking”

iii) I recommend removing the DepMap data entirely unless a global analysis of all oxphos genes shows a significant association with T-ALL, or the authors can somehow reconcile why so many oxphos genes have the inverse association.

ANSWER: We thank reviewer for this advice, we agree that the data set of T-ALL in the DepMap cell line inventory is not as compelling. Nevertheless, even with 3 representative cell lines we can observe statistically relevant changes between OxPhos signature in T-ALL and other hematological malignancies or other cancers, which supports our hypothesis. These data together with new graphical comparison of gene sets are now moved to Supplement and are shown in new Supplementary Figure S1G.

Fig.1SG: Venn diagrams enriched by over-representation analysis of differentially expressed genes, and Volcano plots showing cancer dependencies associated with OxPhos-related genes, graphed as P value ($-\log_{10}$, y-axis) against effect size (x-axis), comparing T-ALL versus other hematological malignancies (left) and T-ALL versus other cancers (right).

2. Fig 2A: These data are very interesting, but I would like to be able to tell whether the drug kills MS5-DL4 treated cells, or whether DL4 treatment increases cell growth/proliferation and the drug reduces their proliferation back to baseline. I can think of several ways this could be addressed: by showing absolute cell counts over time (do cell counts in MD5-DL4 cells drop below initial starting cell counts, or did the drug just stop proliferation), or cell cycle state and apoptosis markers (if cell death is apoptotic) could be assessed to distinguish these.

ANSWER: We thank the reviewer for this important suggestion. To address this concern, we have now documented the time course of drug response and illustrated the percentage of apoptotic cells as assessed by Annexin V and DAPI staining, including early and late apoptotic cells together with the percentage of necrosis, in addition to the percentage of remaining viable cells. Data are shown for *LMO1*^{tg} and *SCL*^{tg}*LMO1*^{tg} pre-LSCs co-cultured on MS5-DL4 stromal cells in a new supplemental Figure S2C, S2D for pre-LSCs. Moreover, we show that the viability of MS5-DL4 stromal cells was not affected by IACS-010759 at any of the tested concentrations (Figure S2E) together with an absence of visual morphological change, illustrated here at 133 nM of IACS-010759 (Figure S2F).

Fig. S2 C-F.

(C) Cell death analysis of *LMO1*^{tg} pre-LSCs co-cultured on MS5-DL4 stromal cells following drug treatment. Shown is the percentage of apoptotic cells as assessed by Annexin V and DAPI staining at the indicated time.

(D) Cell death analysis of *SCL^{tg}LMO1^{tg}* pre-LSCs co-cultured on MS5-DL4 stromal cells following drug treatment. Shown is the percentage of apoptotic cells as assessed by Annexin V and DAPI staining at the indicated time.

(E) Viability analysis of MS5-DL4 stromal cells following drug treatment. MS5-DL4 stromal cells were grown in presence of different concentrations of IACS-010759. Shown is the percentage of living MS5-DL4 stromal cells.

(F) DIC image of cell cultures after 48h treatment with the indicated dose of IACS-010759.

3. Fig 2B: These data are plotted in a way that is very difficult to interpret. What precisely is plotted in the X axis, is this relative change in AUC? I cannot tell from relative changes in AUC how drug-sensitive the cells are at clinically relevant concentrations. Are these data taken from curves such as in Fig 2A? If so I would much prefer to see the curves themselves, these are much more informative.

ANSWER: We thank the reviewer for this valuable comment. As suggested, we have now illustrated full dose-response curves for all five genotypes maintained on MS5-DL4 (**Figure 2C**) or MS5 (**Figure 2D**). Absolute cell counts per culture treated with 133 nM of IACS-010759 or control DMSO are shown in **Figure 2E** (MS5-DL4) and **Figure 2F** (MS5). Finally, these data are captured in **Figure 2G** with more precise labelling, illustrating the areas under the curves for pre-LSCs from all five genotypes maintained on MS5-DL4 or MS5, treated with the doses of IACS-010759 shown in **Figures 2C** and **2D**, compared to DMSO controls. The remaining important data are shown in **Supplemental Figure S2 F**.

Figure 2

(A) The Notch1 gene mutational status differs between *SCL^{tg}LMO1^{tg}* preleukemic and leukemic thymocytes.

(B) Overexpression of NOTCH1 protein as assessed by flow cytometry immunofluorescence. Thymocytes from Wild type and *NOTCH1^{tg}* mice were stained with an antibody against NOTCH1 or an isotype control.

(C) Dose-response of pre-LSCs to IACS-010759 treatment. Primary pre-LSCs from the indicated genotype taken during the preleukemic phase were purified and seeded on MS5-DL4 stromal cells followed by 48h drug treatment at the indicated doses. Data shown are representative of independent experiments performed in triplicates.

(D) Dose-response of pre-LSCs to IACS-010759 treatment. Primary pre-LSCs from the indicated genotype taken during the preleukemic phase were purified and seeded on MS5 stromal cells followed by 48h drug treatment at the indicated doses. Data shown are representative of independent experiments performed in triplicates.

(E) Time course analysis of IACS-010759 effects on pre-LSCs viability. Shown are the numbers of viable primary pre-LSCs recovered after treatment with IACS-010759 (133 nM) or DMSO control for the indicated time on MS5-DL4 stromal cells (mean \pm SD, n=3).

(F) Time course analysis of IACS-010759 effects on pre-LSCs viability. Shown are the numbers of viable primary pre-LSCs recovered after treatment with IACS-010759 (133 nM) or DMSO control for the indicated time on MS5 stromal cells (mean \pm SD, n=3).

(G) Pre-LSC sensitivity analysis to IACS-010759 for the indicated genotypes. The Area under the curves (AUC) were computed from dose-response data illustrated in (C). Shown is the AUC difference obtained between DMSO (black ring) and drug-treated cells ($P < 0.001$, multiple t-tests, $n=2$ or 3 mice per genotype). Notice the difference in sensitivity to IACS-010759 with (green) and without DL4 (blue) stimulation, as well as in pre-LSCs bearing or not the NOTCH1 oncogene in the absence of DL4.

4. A second point related to Fig 2B is that most mouse T-ALLs induced by Scl acquire NOTCH1 mutations during oncogenic transformation (<https://pubmed.ncbi.nlm.nih.gov/16166587/>), this may be the case in the other models as well. However, the authors suggest that T-ALLs lacking a NOTCH1 transgene are NOTCH1 wild-type (the authors refer to “the genetic status of NOTCH1” on p. 6) which seems questionable. It may be that the NOTCH1 transgene drives stronger NOTCH1 activation than the acquired NOTCH1 mutations in the other cases or maybe they truly are NOTCH1 wild-type, either way I would like to see Notch1 activity measured directly. I think the simplest way to address this would be by comparing ICN1 levels by Western blot in the various leukemias, cultured on MS5 vs MS5-DL4. This will allow the authors to test a prediction from their model that leukemias lacking a NOTCH1 transgene should have much lower ICN1 levels than the NOTCH1 transgenic leukemias, but only when cultured on MS5 cells and not MS5-DL4.

ANSWER: We thank the reviewer for this opportunity to clarify our experimental model. As stated by the reviewer, T-ALL that develop in transgenic mouse models of the human disease frequently harbor *Notch1* gain of function mutations that occur within the conserved dimerization domain or the PEST domain, exactly mimicking mutations found in the human disease. Indeed, we and others described the acquisition of *Notch1* mutations in the *SCL*^{tg} mouse model starting at 8 weeks of age, to be followed by a rapid onset of detectable T-ALL at 12 weeks of age. A major advantage of the mouse model is the possibility of mechanistic analysis of the pre-leukemic phase, to distinguish initiating events from later events that drive progression to acute leukemia. This has allowed us to demonstrate that the human *SCL* and *LMO1* oncogenes drive stemness gene expression in DN3 thymocytes and convert these cells into aberrantly self-renewing pre-leukemic stem cells, that later acquire *Notch1* mutations to become leukemia initiating cells. This distinction between the pre-leukemic stage, that we carefully took before 5 weeks to avoid any possible mutation, and the leukemic stage is now explicitly illustrated in **Figure 2A**, showing precise *Notch1* mutations in **Figure S2A**. In contrast to these endogenous mutations, the *Notch1/IC* transgene was designed to be a much stronger hyperactive allele for two reasons, this synthetic transgene harbors mutations causing ligand independent dimerization, as well as truncation of the PEST domain resulting in a highly stable NOTCH1 protein. These double mutations have not been found in the human disease, nor in mouse models. We illustrate the major difference in NOTCH1 protein levels between wild type thymocytes and *NOTCH1*^{tg} thymocytes by flow cytometry analysis (**Figure 2B**).

5. Fig 3A: I believe the “no nutrients” is only glutamine, glucose and pyruvate removed but all other nutrients in the basal media (RPMI-1640) of these cells present, this should be

clarified. The product number of the specific RPMI-1640 medium used should also be listed in the methods, there are different versions with glutamine concentrations that vary widely.

ANSWER: We thank the reviewers for pointing out the need to specify the condition of the experiment performed and depicted in **Figures 3 A-C**. For specific starvation condition we used the RPMI1640 media with 2 g/L sodium bicarbonate, w/o L-glutamine & glucose and pyruvate, provided by MPBio (Catalog Number: 1646849). This allowed us to examine the growth condition in the absence of glucose, glutamine and pyruvate (no nutrient conditions), or to examine in a targeted way the advantage of particular nutrients separately, glutamine only (labelled as Gln), Glucose only (Glc) or Pyruvate only (Pyr), compared to the advantage of two-by-two combinations (Gln+Pyr; Gln+Glc; or Glc+Pyr) and finally all three nutrients together. The above-mentioned media was also used for isotope labelling experiments, allowing us to precisely supplement the media with desired concentration of normal or heavy glucose and glutamine. The proper catalog number of the RPMI1640 media w/o glutamine and glucose was added in the method section under Cell Viability assay. For all other experiments the regular RPMI1640 media purchased from SIGMA already supplemented with 2 mM Glutamine was used.

6. On p. 10 the authors argue that the “elevated aKG/citrate ratio indicates utilization of reductive metabolism”, but this conclusion

i) needs supporting references, and

ANSWER: The supporting references to the publications authored by Fendt et al. Reductive glutamine metabolism is a function of the α -ketoglutarate to citrate ratio in cells, Nat. Communication 2013; 4: 2236 and Metallo et al. Reductive glutamine metabolism by IDH1 mediates lipogenesis under hypoxia, Nature 2012; **481**, 380–384 were added.

ii) seems too strong, I suggest rephrasing to an elevated ratio “has been associated with” a switch to reductive metabolism (unless there are references I am missing).

ANSWER: We feel that above mentioned references justified sufficiently our choice to use “ indicates” instead “has been associated with” .

7. Fig 4B: I simply cannot understand the conclusions the authors draw in the text based on Fig 4B. On p. 11 the authors state that “the relative sensitivity of NOTCH1-mutant cells to IACS-010759 was increased when glutamine was provided”, however the cells are actually LESS drug sensitive when glutamine is added compared to no nutrients (compare blue vs pink curve). They then say that this sensitivity “further improved with Pyr addition”, however Pyr has biphasic effects – Pyr + Gln (orange curves) cells are more sensitive than Gln (blue) cells at low drug doses, but less sensitive at high doses, which is a difficult-to-interpret result that is ignored here. It is also not clear what the authors mean by "improved", do they mean more or less drug sensitive? This part of the paper comes perilously close to be a fatal flaw for the entire paper, it needs to be rewritten entirely to clarify how the authors are interpreting these results, and to avoid stating conclusions that are not supported by the data.

ANSWER: We thank the reviewer for giving us the opportunity to clarify this point and present our data in more lucid way. To address this concern, we presented every IACS-010759 response in particular condition as a value normalized to the response in DMSO

control. As presented first in **Fig. 3A** we first show that the supplementation with glutamine only increased the viable cell number when comparing with cells grown in media without supplementation, without OxPhos-i treatment.

In the new **Fig. 4B** we now present how the cells in each of the examined conditions respond to IACS-010759. To visualize this better, every response was normalized to cells treated with DMSO in particular condition. This helped to present that the two greatest contributors to IACS-010759 response are glutamine and glutamine+ pyruvate.

The response is however flattening /inferior after combining glutamine or glutamine+pyruvate with glucose due to the switch to glycolysis. To avoid confusion regarding “improved” we rephrased the sentence as following:

“NOTCH1-mutant cell lines showed higher sensitivity to IACS-010759 in presence of Gln (blue curve) and glutamine+pyruvate (orange curve), with average IC₅₀s of 0.677 nM and 0.552 nM, ($p<0.05$ and $p<0.0005$ respectively, **Figures 4B-C; S12A-C**).”

8. The authors then move on to test genetic inhibition of glutamine synthase in combination with oxphos inhibition, which looks promising but there is a major caveat of the approach which should be stated in the text - GLS is genetically inhibited in the leukemic cells and not in the rest of the mouse. I would also like for the authors to test whether there is a therapeutic index for GLS inhibition + IACS-010759– is this combination any less toxic to normal hematopoietic cells? Likewise, is VXL + OxPhos inhibition less toxic to normal hematopoietic cells than leukemic cells?

ANSWER: We thank reviewers for rising this concern and suggestion.

To clarify the experiment, we changed the sentence to “Mice received vehicle or IACS-010759 therapy, alone or with tamoxifen to induce *GLS* knockout selectively in leukemic cells.”

To complete the cytotoxicity study and to examine if the selected doses of all drugs might be toxic for healthy hematopoietic cells, we performed CTG assay, seahorse assay and colony forming unit assay on freshly harvested bone marrow cells from healthy donors that we pretreated with IACS-010759, CB839, IACS-010759/CB839, VXL and VXL/IACS-010759 at the doses selected for T-ALLs.

As presented in new included supplemental figure S16: we observed that these agents alone or in combination reduced OCR and moderately reduced viable cell numbers (**Fig S16 C, D**), but had minimal effect on colony growth of healthy BM progenitor cells (**Fig S16 E, F**).

The results are included in **Supplemental Figure S16 C-F** for effects of CB839 and CB839/IACS-010759 combination.

C

D

E

F

(C) Oxygen consumption rate (OCR) (left) and extracellular acidification rate (ECAR) (right) determined by Mito Stress Test in healthy donor derived bone marrow cells treated with 4 hr treatment with vehicle (control), 10 nM of IACS-010759, 12 hr treatment with 1 μ M of CB-839; a shown for 2 independent healthy donor bone marrow samples;

(D) Viability analysis of human bone marrow cells derived from healthy donor following treatment with 1 μ M CB-839; 10 nM IACS-010759 or combination of both for 96 hr; normalised to DMSO-treated controls, from 3 independent bone marrow donors, as measured by CTG assay;

(E) Comparison of results for colony forming unit assay. Healthy bone marrow cells were obtained fresh from healthy bone marrow donors and seeded at the density of 1 million/ml for 24 hr with DMSO (CONTROL), 1 μ M CB-839; 10 nM IACS-010759 or combination of both, followed by collecting the cells and counting and seeding in Methylcellulose at the density of 25 k/ml for 12-14 days; colonies were counted manually from 3 independent dishes per consistency.

(F) Presentation of colony units according the lineage and comparison across the treatments as described in (E).

The effects of VXL and VXL/IACS-010759 combination on healthy bone marrow cells were summarized in a new **Supplemental Figure S25 C-E.**

(C) Oxygen consumption rate (OCR) (left) and extracellular acidification rate (ECAR) (right) determined by Mito Stress Test in healthy donor derived bone marrow cells treated with 24 h treatment with vehicle (control), 10 nM of IACS-010759, VXL or its combination; a shown for 2 independent healthy donor bone marrow samples;

(D) Viability analysis of human bone marrow cells derived from healthy donor following treatment with VXL; 10 nM IACS-010759 or combination of both for 96 hr; normalized to DMSO-treated controls, from 3 independent bone marrow donors, as measured by CTG assay;

(E) Comparison of results for the total number of colonies from colony forming unit assay (left) and presentation of colony units according to the lineage and comparison across the treatments (right); Healthy bone marrow cells were obtained fresh from healthy bone marrow donors and seeded at the density of 1 million/ml for 24 hr with DMSO (CONTROL), 10 nM IACS-010759, VXL or combination of both, following by collecting the cells and counting and seeding in Methylcellulose at the density of 25 k/ml for 12-14 days; colonies were counted

manually from 3 independent dishes per consistency. All graphs show mean values, and error bars represent standard deviations, n=3. P-values: *<0.05; **<0.005; ***<0.001; and ****<0.0001, two-tailed Student t-test.

9. One rationale the authors give for using asparaginase is that it has glutaminase activity, however this is quite modest for the most common clinical asparaginase from E. coli (<https://pubmed.ncbi.nlm.nih.gov/17011787/>). Which form of asparaginase did the authors use, and does this lead to depletion of glutamine levels when treating leukemic cells? This can be assayed using the clinical assay for serum amino acid quantification in the supernatant of treated cells. If glutamine is not depleted, the rationale for using asparaginase will need rethinking.

ANSWER: We thank reviewer for this comment.

In our assays we used L-Asparaginase from E. Coli purchased from Sigma (Cat. Nr A3809) for in vitro studies (we included this information under *Methods* section).

For in vivo work E. coli-derived L-asparaginase was purchased from Elspar, Lundbeck, Deerfield, IL, USA.

We performed additional experiments to determine if the L-Asparaginase preparation used in our study has glutaminase activity. As we could demonstrate in our mass spectrometry analysis performed on cell culture media after treatment with VXL, both asparagine and glutamine levels were significantly depleted in VXL or VXL+IACS-010759 treated cells, whereas media collected from control cells or IACS-010759 treated cells did not show such changes. These results were included in a new supplemental **Fig. S26C and S27C**.

C

S26

S26 C

C) Comparison of selected metabolites: glutamine, glutamate, Asparagine and aspartate measured in culture media collected from NOTCH1-mutated cell line PF382 cell culture after treatment with vehicle, IACS-010759, VXL or VXL/IACS-010759 combination.

C

S27

C) Comparison of selected metabolites: glutamine, glutamate, Asparagine and aspartate measured in culture media collected from NOTCH1-wild type cell line SUPT1 cell culture after treatment with vehicle, IACS-010759, VXL or VXL/IACS-010759 combination.

I also recommend additional changes to the text:

1. This sentence on p.4 of the introduction is problematic: “Unlike solid tumors, which often depend on aerobic glycolysis for their metabolic needs, leukemia cells depend on oxidative phosphorylation (OxPhos) to meet their energetic and biosynthetic demands” The sentence suggests that dependence on glycolysis vs oxphos are mutually exclusive, but most cancers are dependent on both at least to some degree. Simply starting with “Leukemia cells depend on oxphos...” would avoid this problem.

ANSWER: We thank the reviewers for this suggestion. The sentence was rephrased as suggested.

2. What “VXL” chemotherapy is needs to be defined in the text

ANSWER: Thank you for rising this question. The explanation what VXL was included on page 3:

“The backbone of standard of care chemotherapy for T-ALL includes vincristine, dexamethasone, and L-asparaginase (VXL)” and is cited appropriately.

Reviewer #2 (Remarks to the Author):

The authors identify OxPhos as a critical pathway for T-ALL cell survival and demonstrate a direct relationship between NOTCH1, OxPhos activity and acquired chemoresistance in pre-leukemic and leukemic models. The link with NOTCH1 is not clearly shown. The authors inhibit OxPhos with IACS-010759, an inhibitor of mitochondrial complex I, which caused growth inhibition in NOTCH1-mutated T-ALL cells. The authors study the mechanism of metabolic reprogramming in detail. In addition it is shown that IACS-010759 combined with L-asparaginase reduced glutaminolysis and acted in synergy to inhibit leukemia cell growth.

General comments:

T-cell acute lymphoblastic leukemia (T-ALL) is not commonly associated with relapse and chemoresistance as described in the first sentence of the abstract. T-ALL is indeed associated with a higher risk than many other B-ALL subtypes, but relapse is not very common in T-ALL.

ANSWER: We thank reviewer for this valuable comment. Our abstract was rephrased accordingly to:

“T-cell acute lymphoblastic leukemia (T-ALL) is commonly driven by activating mutations in NOTCH1 that facilitate glutamine oxidation. Here we identify oxidative phosphorylation (OxPhos) as a critical pathway for leukemia cell survival and demonstrate a direct relationship between

NOTCH1, elevated OxPhos gene expression, and acquired chemoresistance in pre-leukemic and leukemic models”.

The authors focus on NOTCH1 mutant T-ALL and seem to suggest that the findings are unique for NOTCH1 mutant T-ALL, but do not give evidence for this by comparing the findings in NOTCH1 mutant versus NOTCH1 wild type T-ALL.

ANSWER: We thank reviewers for this comment, however we think that across the manuscript we presented multiple examples of difference and distinct metabolic phenotype and response to OxPhos inhibitors between NOTCH mutant and NOTCH wild type.

First we showed the different signature in gene expression on the level of pre leukemic cells with data presented on **Fig. 1**, we also showed different expression level in 2 different patients' cohorts with distinction between NOTCH1 mutant and NOTCH1 wild type patients, providing the evidence of differently driven metabolism with prevalence of OxPhos in Notch1 mutant patients.

We further show that the leukemic cells that harbor NOTCH1 mutation show significantly higher sensitivity to OxPhos inhibitors (**Fig. 2**). We also show that pattern of the response between leukemic cells with NOTCH 1 mutation is different comparing with NOTCH-wild type leukemia, as only mutants accumulate reactive Oxygen species.

We further show that the differences in response to OxPhos are related to higher dependency on utilization of glutamine by NOTCH1 mutants and dependency on glucose in NOTCH1 wild types (Fig. 3 E, F, G, H , I).

Are the findings specific for NOTCH1 mutant T-ALL or is it more general ?

ANSWER: We believe that throughout the manuscript we could convince the reviewers and will convince the readers that OxPhos dependency is indeed the hallmark of leukemia, however the NOTCH1 mutation makes the cells much more OxPhos-dependent and hence more vulnerable to OxPhos inhibition therapy.

The text is very dense and the figure legends are rather poor, so that it is not easy to understand what the authors do on each figure (which samples used, which technique used, what is compared)

ANSWER: We thank reviewer for this comment and we did our best to improve the figure legends.

There are references lacking in the text, for example: “glutaminolysis is another metabolic feature of T-ALL and AML that is crucial for anaplerosis...”, page 4, “part of the efficacy of L-Asp against T-ALL relies on its GLS-I activity”, page 5...

ANSWER: We thank for this valuable suggestion.

To the statement on page 4, we cited following reports:

Herranz, D. *et al.* Metabolic reprogramming induces resistance to anti-NOTCH1 therapies in T cell acute lymphoblastic leukemia. *Nat Med* **21**, 1182-1189, (2015).

Jacque, N. *et al.* Targeting glutaminolysis has antileukemic activity in acute myeloid leukemia and synergizes with BCL-2 inhibition. *Blood* **126**, 1346-1356, (2015).

To the statement on page 5: “, “part of the efficacy of L-Asp against T-ALL relies on its GLS-I activity” the following reference was added:

Chan,W-K. et al. Glutaminase Activity of L-Asparaginase Contributes to Durable Preclinical Activity against Acute Lymphoblastic Leukemia. *Mol Cancer Ther* **18**, 1587-1592 (2019)

The representation of the data is not consistent: the authors analyze the data in a different way on each figure / figure panel (sometimes IC50, sometimes AUC, others normalized AUC ...). Normally AUC is used when the number of doses is limited, and that is not the case here, so it seems better to use always IC50.

ANSWER: We thank reviewers for this suggestion. However, both agents used in the study, IACS-010759 and CB-839, have rather cytostatic and not cytotoxic effects, and as a single compound they reach in many cases a plateau effect without further decreasing the viability with increasing dose. In our opinion the IC50 measurement is additionally supported by measurements of AUC that reflect both drug sensitivity and maximum inhibition, to get a more comprehensive appreciation of drug effects on cell growth/ATP production or OCR.

We therefore consistently include both measurements, IC50 and AUC, and consistently use the same way to normalize and present data.

The authors indicate that current NOTCH1 inhibitors cannot be used due to their toxicity, but they don't check the toxicity of the drugs that they use.

ANSWER: We thank the reviewers for this valuable comment. To address this gap on knowledge, we performed treatment on healthy Bone Marrows with all used in the manuscript drug combinations including:

Vehicle, IACS-010759, CB839, CB839+IACS-010759, VXL and VXL+IACS-010759 and we performed after 24 hr pretreatment the colony forming unit assay to evaluate if the doses used to target leukemia might have a negative impact on healthy cells.

The results generated on 3 independent freshly harvested bone marrows from healthy donors, indicate that the used doses either have a minimal (not statistically significant effect on CFU formation) or rather support colony growth.

Those results were included in new supplemental figure **S16** for CB839 and IACS-010759 and new figure **S25** for VXL and IACS-010759 combinations.

There are some overstatements in the text (for example the use of “directly linked” without clear confirmation). Overall, the text would benefit from some adjustments to correct for this.

ANSWER: We thank reviewer for pointing this out and have rephased the statement as “linked to”.

Specific points:

Figure legends 1B and 1C are switched

ANSWER: We thank reviewers for pointing this out. The labelling of the figures was corrected.

Figure 1 H and I: they used previous datasets of Crispr Cas9 screen but it is not clear what the authors are comparing here.

ANSWER: We thank reviewer for this comment. Following the suggestions expressed by Reviewer 1, we agree that the data set of T-ALL in the DepMap cell line inventory is rather preliminary, nevertheless even with 3 representative cell lines we can observe statistically relevant changes between OxPhos signature in T-ALL and other hem malignancies or other cancers, which supports our hypothesis.

As the statistical data show rather modest significance, we decided to move this data together with graphical comparison of gene sets to **Supplementary Figure S1G**.

Therefore we rephrased the addressed paragraph in manuscript as follow:

Furthermore, the comparison of dependence scores from DepMap unbiased genome-scale CRISPR-Cas9 screening data³⁹ between three T-cell lines (PF382, SUPT1, HSB2) to other hematologic cell lines (N=73) or other cancer cell lines (N=686) showed significantly higher number of OxPhos genes in T-ALL. In addition, OxPhos genes were significantly over-represented among T-ALL dependent genes, as compared to those for hematologic malignancies or other cancer cell lines ($p=0.03$ and $p=0.048$ respectively; **Figure S1G**) suggesting the importance of OxPhos genes for T-ALL survival³⁹.

Fig.1SG: Venn diagrams enriched by over representation analysis of differentially expressed genes, and Volcano plots showing cancer dependencies associated with OxPhos-related genes, graphed as P value ($-\log_{10}$, y-axis) against effect size (x-axis), comparing T-ALL versus other hematological malignancies (left) and T-ALL versus other cancers (right).

Figure 2B: not a clear way of showing the data, seems the authors are picking the statistics to show the best differences.

ANSWER: We thank reviewer for this comment, and we understand that this way of presenting very complex data could raise questions. To address this, we now exchanged the **Fig 2A** and **2B** with graphs showing full dose-response curves for each genotype separately, which we hope will lead to better understanding of the data presented in former **Fig 2B**. We also separated co-cultures of pre-LSCs on MS5-DL4 (**2C**) and on MS5 (**2D**) stromal cells. Raw cell counts are shown in **Figure 2E** and **F**. Doses-response curves were analyzed by non-linear curve fitting, and IC50 as well as maximum inhibition are shown in new **Figure S2A**.

We now include the revised new figures **2A-F** (below) and moved the previous **Fig. 2B** to the supplemental **Figure S3**.

Figure 2

(A) The Notch1 gene mutational status differs between *SCL^{tg}LMO1^{tg}* preleukemic and leukemic thymocytes.

(B) Overexpression of NOTCH1 protein as assessed by flow cytometry immunofluorescence. Thymocytes from Wild type and *NOTCH1^{tg}* mice were stained with an antibody against NOTCH1 or an isotype control.

(C) Dose-response of pre-LSCs to IACS-010759 treatment. Primary pre-LSCs from the indicated genotype taken during the preleukemic phase were purified and seeded on MS5-DL4 stromal cells followed by 48h drug treatment at the indicated doses. Data shown are representative of independent experiments performed in triplicates.

(D) Dose-response of pre-LSCs to IACS-010759 treatment. Primary pre-LSCs from the indicated genotype taken during the preleukemic phase were purified and seeded on MS5 stromal cells followed by 48h drug treatment at the indicated doses. Data shown are representative of independent experiments performed in triplicates.

(E) Time course analysis of IACS-010759 effects on pre-LSCs viability. Shown are the numbers of viable primary pre-LSCs recovered after treatment with IACS-010759 (133 nM) or DMSO control for the indicated time on MS5-DL4 stromal cells (mean ± SD, n=3).

(F) Time course analysis of IACS-010759 effects on pre-LSCs viability. Shown are the numbers of viable primary pre-LSCs recovered after treatment with IACS-010759 (133 nM) or DMSO control for the indicated time on MS5 stromal cells (mean ± SD, n=3).

(G) Pre-LSC sensitivity analysis to IACS-010759 for the indicated genotypes. The Area under the curves (AUC) were computed from dose-response data illustrated in (C). Shown is the AUC difference obtained between DMSO (black ring) and drug-treated cells ($P < 0.001$, multiple t-tests, $n=2$ or 3 mice per genotype). Notice the difference in sensitivity to IACS-010759 with (green) and without DL4 (blue) stimulation, as well as in pre-LSCs bearing or not the NOTCH1 oncogene in the absence of DL4.

Figure 2 C-K: It is not clear if there is any specificity for NOTCH1 mutated T-ALL

ANSWER: We thank reviewer for this comment. To emphasize the specificity for NOTCH1 mutated T-ALL we now illustrate full dose-response curves for all five genotypes maintained on MS5-DL4 (**Figure 2C**) or MS5 (**Figure 2D**). Absolute cell counts per culture treated with 133 nM of IACS-010759 or control DMSO are shown in **Figure 2E** (MS5-DL4) and **Figure 2F** (MS5). As can be seen in Figure 2D and 2F, when pre-LSCs were maintained in the absence of DL4, expression of the *IC9-Notch1* transgene confers sensitivity to IACS-010759 treatment whereas pre-LSCs lacking the *Notch1* transgene remain viable. We have also clarified more explicitly that pre-LSCs were taken at the preleukemic stage, i.e. before 5 weeks, to absolutely rule out the possibility of *Notch1* gain of function mutations in the *SCL^{tg}LMO1^{tg}* or the *LMO1^{tg}* model that occur at 8 weeks of age or later (Figure 2A). Sequencing data for the endogenous *Notch1* gene are shown in Figure S2A. Therefore, the genetic mouse models indicate that in the absence of DL4 stimulation, sensitivity to IACS-010759 is conferred by the *IC9-Notch1* transgene. In contrast, in the presence of DL4 stimulation, all five genotypes are sensitive to IACS-010759 inhibition. These data are also captured in **Figure 2G** with more precise labelling, illustrating the areas under the curves for pre-LSCs from all five genotypes maintained on MS5-DL4 or MS5, treated with the doses of IACS-010759 shown in **Figures 2C** and **2D**, compared to DMSO controls. The remaining important data are shown in **Supplemental Figure S2 F**.

We also included new Fig. 2 H, where we show that oxygen consumption rate is significantly increased in NOTCH1 mutant cell lines but not in wild type cell lines when compared to T-lymphocytes. We show in Fig, 2 I, that the IACS-010759 inhibits OCR basal in NOTCH1 mutant cell lines more significantly than in NOTCH1 wild type. In NOTCH1 mutant cell lines; as shown in Fig. 2 J, IC₅₀ is the lowest one across all examined cell lines. In New Fig. 2 K we added 3 new T-lymphocyte samples and show that these cells decrease only slightly the viability at 10 nM of IACS-010759, whereas NOTCH1 wild type and mutant decrease viability. The previous Fig. 2 F, H, I, we now moved to Supplementary Figures S3 C, D, F, G, H respectively.

page 7, at the end: “IC50 was not reached”? IC50 is always reached (if the concentration range is high enough), in fact the numbers are shown in the graph.

ANSWER: We thank reviewers for pointing this.

To clarify this observation, we rephrased the sentence accordingly:

“Consistent with this, IACS-010759 produced striking dose-dependent inhibition of ATP production in T-ALL lines (**Figures S3F**), with the lowest IC₅₀ in *NOTCH1*-mutant lines

($p=0.0008$, Figures 2J), while only moderate ATP reduction and viability reduction of less than 20% at the highest used dose was observed in healthy T-lymphocytes (Figures S3F-H).

Figure 3A: viability measured by ATPlite. Since the authors show differences in metabolism, they could validate these effects on viability by viability dye or other method not related to mitochondria.

ANSWER: We thank the reviewers for this comment, and we exchanged the old figure that shows the ATP measurements with the new one reflecting the summary of viable cell number obtained after culturing the cells in different conditions.

Here we can much clearer demonstrate that of all 3 nutrients, the addition of only glutamine contributes significantly to cell number increase up to 50%, comparing to no nutrients added, and was stronger than pyruvate alone. On the contrary, glucose reduced the growth of cells.

We also showed that glutamine in addition of pyruvate further supported cell growth, whereas glucose in combination of glutamine rather decreased the cell growth.

This observation indicates that the T-ALL cells preferentially use glutamine for cell proliferation and that this effect can be additionally increased by pyruvate addition, those two nutrients have synergistic effect on cell growth, whereas glucose rather reduces cell growth in T-ALL context.

Figure 4B: The synergy between IACS-010759 and Gln or Glc/Gln or no-nutrients is not clear. The curves are parallel or even the inhibitor shows less effect than control medium. The major effect seems to be the nutrients, not the drug. Only the yellow curve shows this increased sensitivity to IACS-010759.

ANSWER: We thank for pointing this out.

In revised **Figure 3A** we showed now much clearly the advantage of every nutrient alone and their combination. We therefore normalized the responses to drug for each media condition in reference to cells treated with DMSO in these conditions. This helped to clarify the complex picture and show better the absolute response to IACS-010759 and show the distinct pattern between conditions containing glutamine and conditions predominantly containing glucose. We now convincingly demonstrate in the new **Fig. 4B**, that the curves obtained for glutamine and glutamine+pyruvate showed the highest sensitivity to IACS-010759.

B

(B) Summary of the viability of NOTCH1-mutated T-ALL cell lines treated for 96 hr with 0-100 nM of IACS-010759 under the following conditions: no nutrients, pyruvate (Pyr), glutamine (Gln), glutamine+pyruvate (Glut+Pyr), glucose+pyruvate (Glc+Pyr), glucose+glutamine+pyruvate (Glc+Glut+Pyr), normalized to DMSO treated cells in each condition, as measured by CTG;

Figure 5: the clear difference is in survival, I don't see the value of the measurements at day 5 (no differences in combi)

ANSWER: We thank the reviewer for this comment, however we think that showing the results on day 5 indicate two things: that the model is very aggressive and that the approach is very effective after only 5 days of inducing knockout in leukemic cells and treatment administration with IACS-010759. Therefore we would prefer to keep the data as presented.

Figure 5 I: the graph is quite dense, the errors are bigger than the curves.

ANSWER: We thank reviewer for this comment. To address this comment, we decided to present the data of each mouse of the study to justify the source of mentioned error deviation.

Figure 6: I don't see the added value of single cell sequencing here. Unless the authors compared NOTCH1 mutant and NOTCH1 wild type cells.

ANSWER: The aim of using single-cell mass cytometry was to obtain broad insights into signaling pathways occurring in the leukemic cells. Since the sample size was limited, the classical way of analyzing the cell signaling by western blotting would not provide us the complexity of the signals obtained by CYTOF.

Figure 6 E-G: not clear in the figure/legends what is shown here in each column (write PDX names, mutations...).

ANSWER: We thank reviewer for pointing this and we now added to every Figure 6 E-G a name of PDX used. Because of general figure density, all additional information about the NOTCH mutation status of the PDX is included in the Table 3.

Figure 6: the authors picked L-Asp for its GLS activity to check synergy with Oxphos inhibitor, but they don't show any proof of the mechanism of the synergy being mediated by this activity. Or show there is no synergy with other chemo drugs without GLS activity?

ANSWER: We thank reviewers for this comment. In our in vitro studies we could demonstrate that L-Asparaginase at low concentrations and short period of time, similar to Dexamethasone and Vincristine, increase OxPhos (revised **Fig. S23 A**). The same effect, but with lower concentrations of all 3 compounds was observed in combination treatment (revised **Fig. S23 E**).

Figure S23

The chemotherapy-induced OxPhos upregulation would make the cells more sensitive to IACS-010759 inhibition, and overtime more sensitive to enzymatic asparagine and glutamine depletion by L-Asp (revised Fig. S23 and revised Fig. S24 and revised Fig. S26 and S27).

In the figure S26 and S27 we demonstrated the glutamine and aspartate depletion that mimic the mechanism occurring under CB839, blocking access of glutamine-> glutamate and further incorporation of alpha-KG to TCA cycle.

We also show in new figure **S28** a mass spectrometry analysis providing evidence of the advantage of the VXL combinations with IACS-010759 over monotherapy with IACS-010759 or VXL by strong reduction of TCA cycle intermediates together with glutamine and aspartate.

Reviewer #3 (Remarks to the Author):

In this manuscript authors aim to identify and target metabolic vulnerabilities in T-cell acute lymphoblastic leukemia (T-ALL), driven by NOTCH1 mutation. The research area is current and potentially of clinical importance and is of interest to the reader of Nature Communication.

Initially the authors use bioinformatic approaches to demonstrate a link between NOTCH1 status and increased mitochondrial metabolism/oxidative phosphorylation (OXPHOS). The authors then apply clinically relevant and potent Complex 1 inhibitor (IACS-010759) and show that T-ALL cells are sensitive to OXPHOS inhibition. Using nutrient deprivation assays, coupled with metabolic assays, the authors next show that T-ALL cells induce glutamine metabolism following IACS-010759 treatment. Finally, the authors show that the combination of OXPHOS inhibition (IACS-010759) and inhibition of glutamine metabolism (GLS deletion or L-asparaginase treatment) is effective against T-ALL in vitro and in vivo. The main strengths of the paper are the application of different relevant models to address the research questions, coupled with challenging functional assays (i.e., Seahorse), transcriptomics and isotope-assisted metabolomics. In most part, the data is convincing and the effect of IACS-010759, either as a single agent, or in combination with glutamine metabolism inhibition is impressive.

The main weaknesses of the manuscript are that it is very data heavy (30 supplementary figures), with some of the assays poorly explained (i.e., complex nutrient tracing experiments are poorly explained) reducing the clarity/focus of the manuscript. It lacks novel mechanistic insight; IACS-010759 has already been shown to be very potent OXPHOS inhibitor and it is not unexpected that glutamine metabolism is induced following Complex 1 inhibition. Additionally, the concerns regarding the potential toxicity when OXPHOS and glutamine metabolism is combined is not addressed sufficiently. Below I have listed some of the specific concerns/comments I have.

1. Page 5: Provide reference for “Also, part of the efficacy of L-asparaginase against T-ALL relies on its GLS-i activity”. L-asparaginase does not, as far as I know, have GLS-inhibitor activity, it degrades glutamine on top of asparagine.

ANSWER: We thank the reviewer for this comment. We added the reference to the following publication published in MCT: “Glutaminase Activity of L-Asparaginase Contributes to Durable Preclinical Activity against Acute Lymphoblastic Leukemia, By Wai-Kin Chan et al. *Mol Cancer Ther* **18**, 1587-1592 (2019).

2. Figure 2: some of the differences between wt and mutant are not hugely convincing.

ANSWER: The reason for not hugely different results between wt and mutant might be associated with different sample size. While we have in our laboratory 8 Notch-mutant cell lines, the number of available Notch-wild type cell lines is limited to 3. While this could contribute to less clear difference, we are convinced that together with our pre-leukemic model

data we could show and provide proof that the differences in Notch status play a crucial role in metabolism maintenance and with that to cellular response to OxPhos inhibitor.

3. Figure 2: Does DL4 stimulation increase OXPPOS transcriptionally (or OCR by Seahorse)?

ANSWER: We thank the reviewer for this question. The DL4 stimulation of Notch and therefore stimulation of OxPhos has the most crucial role in pre-leukemic stage of ALL development as presented in **Fig. 1 B and C**.

At the leukemic stage, the DL4 does not play anymore the role in increasing OxPhos activity.

To examine this we tested the viability response to IACS-010759 after culturing the healthy T-lymphocytes derived from healthy donors with or without DL4 ligand by measuring viable cell number, apoptosis, relative fluorescence by ATP assay and Oxygen Consumption rate by seahorse.

We could not observe any differences in terms of ATP decrease after IACS-010759 treatment. The viable cell number did not change after DL4 stimulation. Also Mito Stress Test Profile did not show any signs of OxPhos increase. Those results we include as new Supplementary figures S3 I-K and S5 C-F.

4. Figure 2c-d: It would be informative to present the seahorse profiles.

ANSWER: The exemplary seahorse profiles for healthy lymphocytes are now included in the new Supplementary Fig. S5 C-D and Seahorse profiles for leukemia cell lines are presented for each line in Supplementary Fig. S4.

And Figure S5

C

D

E

F

5. Figure 2k: Does ROS have a functional role (do ROS scavengers block some of the effects)?

ANSWER: We thank reviewers for raising this question. To address this, we performed Viability measurement by CTG assay and viability evaluation with Apoptosis induction analysis and measurement of ROS in the cells either pretreated or not with 2 mM N-Acetylcysteine (NAC) by flow cytometry. To make sure that the effects of NAC are not artificially exacerbated due to acidity in culture, both control and NAC treated wells were

additionally buffered with 20 mmol/l HEPES. Our results were in line with data previously reported on NAC in T-ALL in publication: "Targeting oncogenic interleukin-7 receptor signalling with N-acetylcysteine in T cell acute lymphoblastic leukaemia" by Marc R. Mansour in *BJH* Volume 168, Issue 2, (2015), Pages 230-238, showing that T-ALL in particular those with IL7R mutation showed a very high sensitivity to NAC, with IC50 below 300 μ M.

In our first experiment we pretreated cells with 0.5 mM or 2 mM of NAC which is a relatively low dose that is commonly used for quenching ROS in solid tumors.

Jurkat and PF382 cell line showed slight reduction of viability under NAC in control when treated with 2 mM NAC but increased strongly apoptosis rate and decreased significantly viable cell number after IACS-010759 treatment and IACS-010759/CB839 or IACS-010759/VXL treatment under NAC supplementation.

Also, the addition of NAC did enhance the accumulation of ROS under the blockade of Complex I and combination with CB839 and VXL.

The quenching properties of NAC were only observed moderately in conditions containing CB839 alone or VXL alone (Data included in Figure **S21** and Fig. **S29** respectively), indicating that NAC supplementation is not able to quench the ROS accumulation and glutathione depletion upon OxPhos-i/GLS-i treatment, making those changes irreversible.

Figure S21

Figure S21.

(A) Viability analysis of NOTCH1-mutated T-ALL cell lines JURKAT and PF382, following treatment with 1 μ M CB-839; 10 nM IACS-010759 or combination of both for 72 hr; in presence of N-Acetylcysteine (NAC) at the concentration of 0.5 or 2.0 mM respectively, normalized to DMSO-treated controls, from 3 independent experiments (by flow cytometry);

(B) Apoptosis evaluation by Annexin V-assay from the experiment described in (A).

(C) ROS evaluation as expressed by H2DCFDA mean fluorescence intensity (MFI) measured by flow cytometry normalized to CONTROL (cells treated with DMSO) as described in (A). All graphs show mean values, and error bars represent standard deviations, n=3. P-values: *<0.05; **<0.005; ***<0.001; and ****<0.0001, two-tailed Student t-test.

Figure S29

Figure S29.

(A) Viability analysis of NOTCH1-mutated T-ALL cell lines JURKAT and PF382, following treatment with 10 nM IACS-010759, VXL or combination of both for 72 hrs; in presence of N-Acetylcysteine (NAC) at the concentration of 0.5 or 2.0 mM respectively, normalized to DMSO-treated controls, from 3 independent experiments (by flow cytometry);

(B) Apoptosis evaluation by Annexin V-assay from the experiment described in (A).

(C) ROS evaluation as expressed by H2DCFDA mean fluorescence intensity (MFI) measured by flow cytometry normalized to CONTROL (cells treated with DMSO) as described in (A). All graphs show mean values, and error bars represent standard deviations, n=3. P-values: *<0.05; **<0.005; ***<0.001; and ****<0.0001, two-tailed Student t-test.

6. Figure 3a-c: Is pyr-gluc and pyr-gln mixed up (latter looks more effective)? Also, the interpretation of some of these experiments are not clear. “The addition of Pyr with Gln further improved ATP generation to 80%” is that correct? “while NOTCH1-mutants being dependent on Gln to activate OxPhos and generate ATP” is that correct?

ANSWER: We thank reviewer for this suggestion. To present the data in more clear way we now include as **Fig. 3A** the summary of viable cell number normalized to media supplemented with glutamine+ pyruvate+ glucose. We could show that only glutamine supplementation has the strongest effect to support as a single nutrient the cell growth.

7. Figure 3b: Gln addition does not significant increase OCR – explain/discuss.

ANSWER: Whereas OCR was not significantly increased by glutamine, much stronger contrast was observed in the context of glucose supplementation. To visualize that accordingly, we plotted the obtained OCR and ECAR values for each condition as a ratio of OCR/ECAR known as metabolic phenotype and included it as a new **Fig. 3C**. This helped to show that no nutrient, pyruvate and glutamine and glutamine+ pyruvate show significantly different metabolic profile (aerobic) as expressed by high OCR/ECAR ratio than conditions with glucose, which drastically decreased the ratio towards energetic phenotype. The

conditions containing glucose clustered altogether, whereas conditions with pyruvate and or glutamine build up a separate cluster. This cluster separation reflected the differences in sensitivity to IACS-010759.

Fig. 3 C Summary of the metabolic phenotype expressed as ratio of basal OCR/ basal ECAR for the 4 NOTCH1-mutated T-ALL cell lines cultured as described in (B), measured by Seahorse;

8. Is 11mM glucose, 2mM glutamine and 1mM pyruvate used throughout (which is much higher than blood, ca 5.5mM, 550-650µM, 50-100µM)? This needs to be discussed.

ANSWER: We thank reviewer for raising these concerns. In our in vitro experiments we followed the recommendations of the Agilent company and supplemented the Basal media according to the manufacturer instructions. While in vitro work serves as proof of principle, the further validation was conducted in the in vivo setting.

Indeed the levels of glucose and glutamine in human serum is much lower, therefore we were verifying our observations in three in vivo PDX models and one murine leukemia model to make sure that our properly observations reflect the in vivo changes.

9. Figure 4b, have IC50's been calculated and are they backing up claims made in paper (I don't see a clear shift)?

ANSWER: We thank reviewer for this comment. We normalized the dataset separately for each nutrient condition according to DMSO treated cells in this condition and presented the curves as % of control for each condition. We can now in the new Fig. 4B better present that the cells in conditions containing only glutamine or glutamine with pyruvate respond to IACS-010759 with higher sensitivity than in other examined conditions.

B
Fig. 4 (B) Summary of the viability of NOTCH1-mutated T-ALL cell lines treated for 96 hr with 0-100 nM of IACS-01759 under the following conditions: no nutrients, pyruvate (Pyr), glutamine (Gln), glutamine+pyruvate (Gln+Pyr), glucose+pyruvate (Glc+Pyr), glucose+glutamine+pyruvate (Glc+Gln+Pyr), normalized to DMSO treated cells in each condition, as measured by CTG;

10. Figure 4h: It would be better to have untreated single arms and combo for these experiments. Some decreases were also seen in figure 3k (IACS-010759 only).

ANSWER: We thank reviewer for those comment. For the purpose of clear presentation we were focusing for the final result of the treatment, however to provide the effects of the monotherapy treatments and to demonstrate the differences associated with the combination therapy, we are including the comparison of the levels for selected metabolites across all treatment arms.

We compared Vehicle versus IACS-010759 versus CB839 versus CB839+IACS-010759 in the new supplementary Figure **S19**:

Figure S19

A PF382

B SUPT1

We compared Vehicle versus IACS-010759 versus VXL versus VXL+IACS-010759 in a new supplementary Figure S28:

Figure S28

11. For animal experiments, as well as for ex-vivo LCMS: was there a difference in viability of cells between arms? If there was a lot of death, then LCMS would be in affected.

ANSWER: We thank reviewer for raising this concern. At the time of tissue collection for ex-vivo LCMS no changes of viability were observed, however a significant difference in yielded cell number was noticed.

12. Figure 6: Here it would be informative to see what effect media without asparagine, glutamine or both would be (Vxl will deplete both and it would be good to see if one or both contribute to observed phenotypes).

ANSWER: Thank you for sharing this thought and raising the question. To address this point, we analyzed media after coculturing the PF382 and SUPT1 cell lines with VXL or VXL/IACS-010759 combination and measured the impact of L-Asparaginase on the level of glutamine, glutamate, asparagine and aspartate.

L-Asparaginase at the low concentration of 0.1 IU used as a part of VXL combination did decrease significantly the level not only of asparagine but also of glutamine with $p < 0.001$ or lower and led to the accumulation of glutamate and aspartate, accordingly with $p < 0.0001$. This proves not only asparagine-, but also glutamine-depleting activity of L-Asparaginase and supports our data of glutamine depletion /glutaminase inhibition as an important approach for T-ALL treatment.

These results we added accordingly as a new Supplemental Figure S25 C and S26 C.

C

S26

C

S27

13. Note, the Seahorse measured OCR, not ATP levels, so better to refer to ATP-linked OCR.

ANSWER: We assume that this concern refers to the **Figure 3I** where we displayed %ATP production. For those calculations we used a specific kit and media provided by Agilent to

perform Seahorse XFp ATP Rate Assay. This assay and associated calculator allow to evaluate specifically mitochondrial ATP and Glycolytic ATP. To plot the changes in both we therefore labelled axis as % of ATP production referring to either glycolytic or mitochondrial ATP production.

https://www.agilent.com/cs/library/usermanuals/public/103591-400_Seahorse_XFp_ATP_Rate_Assay_Kit_User_Guide.pdf

In all other analysis where we performed Mito Stress Test and specifically the focus was on mitochondrial OCR, we were referring to changes in OCR (either basal or maximal) without ATP calculation, which indeed would be interpreted as OCR-linked ATP.

14. “The exposure to Glc in the presence of Gln switched NOTCH1-mutated cells from OxPhos towards glycolysis” I guess this is expected?

ANSWER: Yes, we agree with the reviewer’s observation. The switch towards glycolysis in the presence of Glc is expected. We intended to emphasize that the leukemic cells have a very strong ability to switch between OxPhos and Glycolysis, however Notch mutations predominantly force cells to OxPhos and glutamine utilization to maintain the growth and proliferation.

15. Page 11: “These results suggest that the maintenance of OxPhos function depends on Gln in NOTCH1-mutated cells and that Gln is crucial for the sensitivity to OxPhos-i (P=0.0001, Figures 4C; S11A-B).” – So what was the rationale for inhibiting GLS?

ANSWER: We thank the reviewer for this question. The intention and rationale for inhibition of GLS was the blockade of the reductive metabolism.

16. Discuss results in relation to Nguyen et al, Downregulation of Glutamine Synthetase, not glutaminolysis, is responsible for glutamine addiction in Notch1-driven acute lymphoblastic leukemia, Mol Oncol, 2020.

ANSWER: The part of the discussion was supplemented by our comments in reference to above mentioned work. Please see part of the discussion on page 20.

17. Page 13: “Consistent with in vitro observations, metabolomic analysis of PB from mice” clarify if this is leukemic or normal cells?

ANSWER: We thank for this suggestion. The indicated sentence was changes to: “Consistent with *in vitro* observations, metabolomic analysis of PB from leukemic mice subjected to dual intervention, revealed an accumulation of NMP, with decreased ATP and TCA intermediates, indicative of profound metabolic reprogramming that suppressed all critical amino acid and nucleotide biosynthesis pathways (Ingenuity Pathway Analysis, IPA)”.

18. Provide reference for “Also, part of the efficacy of L-asparaginase against T-ALL relies on its GLS-i activity.”

ANSWER: We thank the reviewer for providing us this suggestion. The appropriate reference MCT paper: "Glutaminase Activity of L-Asparaginase Contributes to Durable Preclinical Activity against Acute Lymphoblastic Leukemia by Wai-Kin Chan et al. Mol Cancer Ther 18, 1587-1592 (2019)

19. Fig S11A-B: Do JURKAT and PF not have Notch mutation?

ANSWER: Yes, both of the lines are harboring Notch mutation.

Reviewer #4 (Remarks to the Author):

In this paper, the authors investigate the molecular mechanisms by which activating mutations in NOTCH1 associates with T-ALL outcome and therapy. The article is well-written and provides new knowledge that is important from both a scientific and clinical viewpoint. The claims that are being made are generally well-supported by data and I only have minor points for improvements:

ANSWER: We thank the reviewer for such generous comments on our manuscript and appreciate the suggestions to improve our work.

- In the abstract, the following sentence should be adapted as now it appears the treatment combinations results in lethality in the mice (and not in the T-ALL cells): "Thus, pharmacological blockade of OxPhos combined with inducible knockdown of the key glutamine enzyme glutaminase confers synthetic lethality in mice harboring NOTCH1-mutated T-ALL".

ANSWER: We thank for this valuable advice. We rephrased the suggested sentence in following way: "Thus, pharmacological blockade of OxPhos combined with inducible knock-down of the key glutamine enzyme glutaminase confers synthetic lethality of *NOTCH1*-mutated T-ALL *in vivo*."

- IACS-010759 was used to inhibit complex I but I missed the evidence that this indeed happened as expected. Showing complex I inhibition (e.g. be PMP seahorse) should be done and the authors should also take into account that this compound can have off-target effects. This could be experimentally ruled out and/or discussed.

ANSWER: We thank for raising this concerns as many of complex I inhibitors did show a spectrum of off-target effects. However in case of IACS-010759 the drug development group at MDACC has already published detailed studies on this compound authored by Molina et al. in Nat Med 2018 Jul;24(7):1036-1046., demonstrating that IACS-010759 inhibits selectively Complex I. In this study they demonstrated that IACS-010759 is acting solely at complex I and the treatment of detergent-permeabilized cells with IACS-010759 in medium supplemented with pyruvate and malate (to generate NADH for use by complex I) resulted in strongly reduced oxygen consumption rate (OCR was decreased down to 20%), whereas the OCR was not affected by treatment with IACS-010759 when medium was supplemented with succinate to feed complex II, thus bypassing the requirement for complex I function. This mechanism was further supported by the finding that ectopic expression of *Saccharomyces cerevisiae* NDI1, the yeast complex I ortholog, completely restored cell viability and OCR to baseline levels in the presence of IACS-010759. In comparison to doses used for AML or pancreatic tumor in above mentioned paper, we selected the dose of 10-13 nM for majority of

presented in our manuscript experiments, which represent low nanomolar range of the compound with preserved strong mitochondria inhibitory activity.

-C57VL6 -> C57BL6?

ANSWER: Thank you for indicating the typo, the change was accordingly made in the manuscript.

-rephrase 'deeper reduction'

ANSWER: Thank you for this suggestion, instead of initial title “BLOCKADE OF GLUTAMINE METABOLISM WITH COMPLEX I INHIBITION PRODUCES DEEPER REDUCTION OF TUMOR BURDEN IN VIVO”

we rephrased our statement as “BLOCKADE OF GLUTAMINE METABOLISM WITH COMPLEX I INHIBITION IMPROVES DEPTH OF TUMOR BURDEN REDUCTION IN VIVO”

REVIEWER COMMENTS

Reviewer #1 (Remarks to the Author):

The authors have addressed most of my critiques, and I commend them for the huge amount of work in this manuscript. I do however have one remaining minor request:

1. The new data showing apoptosis (Fig S2C-D) is nice, but it is not clear whether the authors are claiming differences and whether these are statistically significant or not. Showing P values on this plot would clarify this. The authors could also remove the "live cells" bars from these plots, which would make room for brackets/P values without diminishing the ability to interpret the data shown.

Reviewer #2 (Remarks to the Author):

The authors have addressed many of the questions, mostly by changing the text or re-analyzing the data. There are still a number of weak points in the current manuscript.

Figure 2: Specificity of OxPhos inhibitors for Notch1 mutant vs Notch1 wild type in T-ALL is not convincing: the authors approach it, but they argue that "Notch1 mutated cells show significantly higher sensitivity to OxPhos inhibitors" based on Figure 2. To me it is not so clear since panels H I J K do not show a statistically significant difference (between wild type Notch and mutant Notch): the statistics are not shown for this comparison.

The authors write: "Oxygen consumption rate (OCR) measurements in T-lymphocytes and T-ALL cell lines separated by NOTCH1 mutation status showed a positive effect of NOTCH1 activation on the basal OCR level ($p=0.05$; Figure 2H)," but in the figure it becomes clear that this statistical difference ($p=0.05$) is between T-lymphocytes and mutant NOTCH1 T-ALL; and has nothing to do with difference between Notch1 mutant versus Notch1 wild type leukemia cells. I find this very misleading, the authors need to clearly state which groups they are analyzing in statistics and also state where there is no statistical difference.

The response is quite variable among groups, and especially in Notch1 WT the authors use $N=3$ cell lines, so the statistics are not very strong. It is not so clear that the difference in response comes from the presence of Notch mutations and not from any other mutations/factors.

The authors also write that "only mutants accumulate ROS" but it is not so clear in figure 2L-M: the majority of Notch1mut cell lines accumulate ROS, but some Notch1mut cell lines present similar levels as observed in the few Notch1 wild type cases, so it is not so clear that Notch1 mutation defines this ROS accumulation, although here overall based on the statistics there is a significant difference, this is acceptable.

Figure 3: The dependence on Gln by Notch1 mut compared to WT based on Figure 3 E-F is not at all clear. Notch1 mutant and Notch1 wild type seem to have very similar profile. The authors write: "With Gln only, NOTCH1-mutants relied only on OxPhos" but is that also not the same for Notch1 wild type cells? I do not see a difference between wild type and mutant in panels E-F.

Toxicity of OxPhos inhibitors: This is now tested in vitro using BM cells (supp S16). The authors nicely show no effects on CFU potential (S16 E), but the effect on cell viability seems important (S16 D).

Figure 5I: This now clearly shows the data for each mouse. The response to IACS-010759 + tamoxifen (GLS -) is very good as there are 3 of 5 mice that do not develop leukemia at all. This is in a way a result that seems surprising (too good) and would need a critical evaluation. Since there is transplantation of cells involved in this model, one easy explanation for 3 mice that do not develop disease at all is that these animals did not have engraftment of the injected cells (for example by a technical problem of injection or lack of sufficient irradiation). What is the evidence that there was engraftment in these 3 mice? Another way to test this is to repeat this experiment, as it seems to be

just one biological replicate at this stage (all 5 mice injected at same time with same cells).
Is there a possibility that tamoxifen has a direct effect on metabolism in this experiment? (that should be controlled by using GLS wild type cells (not floxed) and also treat these with tamoxifen.

Reviewer #3 (Remarks to the Author):

Overall, the authors have made a good effort to address our comments. Consequently, the manuscript has been strengthened in places.

I have no further comments.

Reviewer #4 (Remarks to the Author):

The authors addressed my minor comments. While reading the rebuttal letter, I noticed several important points raised by other reviewers that appear to be specialists in the field. They should further assess whether their distinct points were addressed before proceeding to publication.

REVIEWER COMMENTS

Reviewer #1 (Remarks to the Author):

The authors have addressed most of my critiques, and I commend them for the huge amount of work in this manuscript. I do however have one remaining minor request:

1. The new data showing apoptosis (Fig S2C-D) is nice, but it is not clear whether the authors are claiming differences and whether these are statistically significant or not. Showing P values on this plot would clarify this. The authors could also remove the “live cells” bars from these plots, which would make room for brackets/P values without diminishing the ability to interpret the data shown.

ANSWER: We are very thankful to the Reviewer for this kind comment. We modified the Fig. S2C-D following the above-mentioned suggestions. We removed the “live cells” bars and included *p* values, indicating statistical significance. However, all analyzed differences remain below 10% in LMO1-pre LSC (Fig. S2C) and below 20% of all examined cells for SCL LMO1-pre LSC models, respectively (Fig. S2D). Thus, there was no statistical significance between groups for comparison of total % of necrotic/apoptotic cells. The modified figure and the legend are below.

Legend:

(C) Cell death analysis of *LMO1^{tg}* pre-LSCs co-cultured on MS5-DL4 stromal cells following drug treatment. Shown is the percentage of early apoptotic, late apoptotic and necrotic cells as assessed by Annexin V and DAPI staining at the indicated time.

(D) Cell death analysis of *SCL^{tg}LMO1^{tg}* pre-LSCs co-cultured on MS5-DL4 stromal cells following drug treatment. Shown is the percentage of early apoptotic, late apoptotic and necrotic cells as assessed by Annexin V and DAPI staining at the indicated time.

All graphs show mean values, and error bars represent standard deviations, *n*=3. *p*-values: ns=no significance, *<0.05; **<0.005; ***<0.001; and ****<0.0001, two-tailed Student t-test.

Reviewer #2 (Remarks to the Author):

The authors have addressed many of the questions, mostly by changing the text or re-analyzing the data. There are still a number of weak points in the current manuscript.

Figure 2: Specificity of OxPhos inhibitors for Notch1 mutant vs Notch1 wild type in T-ALL is not convincing: the authors approach it, but they argue that “Notch1 mutated cells show significantly higher sensitivity to OxPhos inhibitors” based on Figure 2. To me it is not so clear since panels H I J K do not show a statistic significant difference (between wild type Notch and mutant Notch): the statistics are not shown for this comparison.

The authors write: “Oxygen consumption rate (OCR) measurements in T-lymphocytes and T-ALL cell lines separated by NOTCH1 mutation status showed a positive effect of NOTCH1 activation on the basal OCR level ($p=0.05$; Figure 2H),” but in the figure it becomes clear that this statistical difference ($p=0.05$) is between T-lymphocytes and mutant NOTCH1 T-ALL; and has nothing to do with difference between Notch1 mutant versus Notch1 wild type leukemia cells. I find this very misleading, the authors need to clearly state which groups they are analyzing in statistics and also state where there is no statistical difference.

ANSWER: We are very thankful for this comment and appreciate raising this pertinent concern. Following the Reviewer’s suggestions, we indicated where the statistical comparison did not reach significance and additionally indicated where the borderline p values were trending towards significance, for instance in Fig. 2H, 2I, 2J, 2K.

The response is quite variable among groups, and especially in Notch1 WT the authors use $N=3$ cell lines, so the statistics are not very strong. It is not so clear that the difference in response comes from the presence of Notch mutations and not from any other mutations/factors.

ANSWER: We are very thankful for this comment. Indeed, these observations were limited due the prevalence of NOTCH1-mutated and paucity of wt-NOTCH1 cell lines available to us. Among 7 examined T-ALL cell lines with NOTCH1 mutation, 3 cell lines CCRF-CEM, ALL SIL and MOLT3 showed the highest basal OCR, and 4 had higher OCR than 2 of 3 wild type T-ALL cell lines. Both, wild type and NOTCH1-mutant cell lines showed higher OCR than lymphocytes which indicates metabolic reprogramming toward OxPhos. Given the known limitations of cell lines, we would like to emphasize that in two independent large T-ALL clinical cohorts of total more than 300 patients we observed distinct differences in OxPhos gene activity which was enriched in NOTCH1-mutant subset (Fig. 1), supporting biologically meaningful trends observed in cell lines.

To acknowledge Reviewer's concerns, we have added to the Discussion (page 19) the following:

It should be noted that IACS-010759 also impacted viability and reduced OCR of *NOTCH*-wt cells compared with healthy lymphocytes. Although not studied here, alternative mechanisms, in addition to *NOTCH1* gene mutation, are known to activate *NOTCH* signaling in T-ALL¹⁻³; and while metabolic phenotypes described in this study are most prominent in *NOTCH*-mutated cells, OxPhos clearly contributes to metabolic fitness of unmutated cells.

The authors also write that “only mutants accumulate ROS” but it is not so clear in figure 2L-M: the majority of *Notch1*mut cell lines accumulate ROS, but some *Notch1*mut cell lines present similar levels as observed in the few *Notch1* wild type cases, so it is not so clear that *Notch1* mutation defines this ROS accumulation, although here overall based on the statistics there is a significant difference, this is acceptable.

ANSWER: We appreciate this comment. To address it, we removed Figures 2L and 2M and present new Fig. 2L, which in our opinion better visualizes changes in ROS between lymphocytes, wt *NOTCH* and mutated *NOTCH* in a dose-dependent manner. Here we show that none of groups is responding significantly to 1 nM of IACS-010759. However, at 10 nM there is a clear rise of ROS content in *NOTCH* mutants and slight increase in *NOTCH* wild type (not reaching significance between *NOTCH* wild type and lymphocytes, yet statistically significant between wild type and mutants). The power of statistical significance in ability to induce ROS is even more evident at 100 nM, where a very clear difference is observed in both: between lymphocytes versus *NOTCH* mutant and *NOTCH* wild type versus *NOTCH* mutant, not reaching significance in comparison of wild type *NOTCH* with lymphocytes.

Legend:

(H) Comparison of basal oxygen consumption rate (OCR) in T-lymphocytes (red), NOTCH1-wild type TALL-1, LOUCY, SUP-T1 (blue) and NOTCH1-mutant: JURKAT, PF-382, MOLT-4, CCRF-CEM, DND-41, ALL-SIL, HPB-ALL (black) cell lines by Mito Stress Test assay;

(I) Comparison of IC₅₀ values calculated from basal OCR curves in (Figure S4A) for T-lymphocytes (red) and NOTCH1-wild type (blue) and -mutated (black) T-ALL cell lines;

(J) Comparison of IC₅₀ values calculated from viability measurements for T-lymphocytes (n=5) and each cell line (n=10) after 96 h of treatment with 0-123 nM of IACS-010759;

(K) Comparison of viable cell number in T-lymphocytes (red), T-ALL cell lines with NOTCH1-wild type (blue) and -mutant (black) after 96 h of treatment with 1, 10 nM and 100 nM of IACS-010759; (mean±SD, n≥3)

(L) Comparison of mean fluorescence intensity (MFI) of reactive oxygen species (ROS) staining by H2DCFDA after treatment with IACS-010759 at 1, 10 and 100 nM in healthy T-lymphocytes (red) and T-ALL cell lines harboring wild-type (blue) and mutant (black) NOTCH1; (mean±SD, n≥3).

All graphs show mean values, and error bars represent standard deviations, $n \geq 3$. p -values: ns-no significance, * <0.05 ; ** <0.005 ; *** <0.001 ; and **** <0.0001 , two-tailed Student t-test.

Figure 3: The dependence on Gln by Notch1 mut compared to WT based on Figure 3 E-F is not at all clear. Notch1 mutant and Notch1 wild type seem to have very similar profile. The authors write: “With Gln only, NOTCH1-mutants relied only on OxPhos” but is that also not the same for Notch1 wild type cells? I do not see a difference between wild type and mutant in panels E-F.

ANSWER: We are very thankful for this comment and appreciate raising this important concern.

To visualize better differences between levels of OxPhos and glycolysis for NOTCH1 wild type and mutant respectively, we decided to present Fig. 3E in the context of response to nutrients, separating according to NOTCH status.

Data in Fig 3E clearly indicate that upon addition of glucose, the ECAR is increased significantly more in WT than in mutant NOTCH T-ALL.

We also agree with very important comment about dependence on OxPhos in the absence of glucose. Indeed, in Fig 3F both wild type and mutants are dependent on OxPhos with Gln only supplementation. However, upon glucose supplementation OxPhos activity remains stable in NOTCH-wt, but decreases in NOTCH-mutated cells, likely indicating switch towards glycolytic metabolism with reduced OxPhos utilization and therefore more tightly regulated nutrient utilization.

Legend:

(E) Comparison of basal ECAR level in cell lines with NOTCH1-wild type ($n=3$) and NOTCH1-mutation ($n=7$) as measured after incubation of cells in media containing only glutamine (blue) or in response to acute injection of glucose (black), measured by Seahorse;

(F) Comparison of basal OCR level in cell lines with NOTCH1-wild type (n=3) and NOTCH1-mutation (n=7) as measured after incubation of cells in media containing only glutamine (blue) or in response to acute injection of glucose (black), measured by Seahorse.

Toxicity of OxPhos inhibitors: This is now tested in vitro using BM cells (supp S16). The authors nicely show no effects on CFU potential (S16 E), but the effect on cell viability seems important (S16 D).

ANSWER: We are very thankful for this comment and appreciate raising this important concern.

The analysis in the Fig. S16D is based on CTG assay (ATP content), and in Fig S16E on the actual number of colonies. To avoid unnecessary confusion, we changed the Y-axis description to "ATP content normalized to control". Fig. S16D shows that after 5 days the ATP was moderately decreased by 30-35% in CB839/IACS-010759 combination group, yet this reduction did not impact colony unit formation in Fig S16E.

(D) ATP content analysis of human bone marrow cells derived from healthy donor following treatment with 1 μ M CB-839; 10 nM IACS01759 or combination of both for 96 hr; normalized to DMSO-treated controls, from 3 independent bone marrow donors, as measured by CTG assay;

To be consistent we changed the Y-axis in Fig. S25D:

(D) Analysis of ATP content of human bone marrow cells derived from healthy donor following treatment with VXL; 10 nM IACS01759 or combination of both for 96 hr; normalized to DMSO-treated controls, from 3 independent bone marrow donors, as measured by CTG assay.

Figure 5I: This now clearly shows the data for each mouse. The response to IACS-010759 + tamoxifen (GLS -) is very good as there are 3 of 5 mice that do not develop leukemia at all. This is in a way a result that seems surprising (too good) and would need a critical evaluation. Since there is transplantation of cell involved in this model, one easy explanation for 3 mice that do not develop disease at all is that these animals did not have engraftment of the injected cells (for example by a technical problem of injection or lack of sufficient irradiation). What is the evidence that there was engraftment in these 3 mice? Another way to test this is to repeat this experiment, as it seems to be just one biological replicate at this stage (all 5 mice injected at same time with same cells).

Is there a possibility that tamoxifen has a direct effect on metabolism in this experiment? (that should be controlled by using GLS wild type cells (not floxed) and also treat these with tamoxifen.

ANSWER: We are very thankful for this comment and appreciate raising this important concern.

To make sure we have equal injections across all mice, we use catheter technique which minimizes tail vein bleeding and reduces deviations in number of injected cells.

Prior to the experiment shown, we performed several pilot studies optimizing timing of engraftment detection and optimizing dose and time of starting induction of GLS KO with Tamoxifen, since this model originally developed by Dr Daniel Herranz (Nat. Med. 2015) is very aggressive.

We performed intravital microscopy of T-ALL prior to GLS ablation at day 14 when mice peripheral blood sampling showed only 1-3% circulating leukemia, however in organs as shown here below, leukemia was fully dominant.

Considering this very rapid disease development, we started monitoring peripheral blood leukemia burden in mice at day 7 after injections. By flow cytometry measurements after excluding dead cells by DAPI staining in UV channel, we observed bright signal of GFP (FITC channel) and excellent separation of murine CD45 signal (APC channel). This was observed in peripheral blood on day 7, summarized on a new graph included in Fig. S22 A. Flow cytometry contour plots for each mouse of cells with bright GFP+ signal (double positive GFP and mCD45+) and non-GFP (mCD45+ population only) were gated, and the percentage of pre-treatment circulating leukemia calculated as % of GFP+ cells/ % of mCD45+ cells *100%. The summary of engraftment and randomization for each group, prior to starting treatment with Tamoxifen or IACS-010759 is included in Fig. S22 B. This data indicate that all mice had established leukemia at the time of therapy start.

Figure S22

Figure S22.

(A) Flow cytometry analysis of murine blood on day 7 post transplantation. After gating out debris and dead cells (DAPI+), double positive: GFP+/mCD45+ leukemic cells were identified on a GFP (FITC) (y-axis)- vs mCD45 (APC) (x-axis)-gated contour plot.

(B) Column diagram presenting the average level of engraftment in BL6 mice harboring GLS fl/fl leukemia prior to treatment initiation: % of gated double positive cells were analyzed and divided by % of all mCD45+ cells and multiplied by 100% to obtain normalized leukemic engraftment, followed by mice randomization into 4 groups: GLS+, GLS+IACS-010759, GLS-, GLS-/IACS-010759, respectively.

Regarding the Tamoxifen effects: In our studies we used 1 mg/mouse tamoxifen over 5 consecutive days, instead of dose published by Herranz at al. Nat. Med. 2015, where just two days after the cell injection, mice were treated with tamoxifen at 5 mg/mouse by intraperitoneal injection to induce deletion of the Pten, Atg7 or Glis.

The study of Herranz at al⁴. already showed that tamoxifen alone at a dose 5 times higher and administered at earlier timepoint than in our studies, did not affect survival/tumor progression.

Also, long-term effects in our studies are very similar to those obtained in GLS KO in combination with DBZ. Below we present for the Reviewer the comparison side by side of the studies published by Herranz at al⁴ (left) and ours (right, with legend included). These results demonstrate very similar extension of survival upon conditional GLS KO (green line Left, blue Right panel) and that co-targeting OxPhos has similar efficacy as inhibition of a direct NOTCH1 target γ -secretase with diazepam (red line left, green line on the right).

Reviewer #3 (Remarks to the Author):

Overall, the authors have made a good effort to address our comments. Consequently, the manuscript has been strengthened in places.

I have no further comments.

ANSWER: We are very thankful for this comment and appreciate this positive feedback on our work.

Reviewer #4 (Remarks to the Author):

The authors addressed my minor comments. While reading the rebuttal letter, I noticed several important points raised by other reviewers that appear to be specialists in the field. They should further assess whether their distinct points were addressed before proceeding to publication.

ANSWER: We thank very much for this comment and appreciate this positive feedback on our work.

References:

- 1 King, B. *et al.* The ubiquitin ligase FBXW7 modulates leukemia-initiating cell activity by regulating MYC stability. *Cell* **153**, 1552-1566, doi:10.1016/j.cell.2013.05.041 (2013).
- 2 Larson Gedman, A. *et al.* The impact of NOTCH1, FBW7 and PTEN mutations on prognosis and downstream signaling in pediatric T-cell acute lymphoblastic leukemia: a report from the Children's Oncology Group. *Leukemia* **23**, 1417-1425, doi:10.1038/leu.2009.64 (2009).
- 3 Liu, Y. *et al.* The genomic landscape of pediatric and young adult T-lineage acute lymphoblastic leukemia. *Nat Genet* **49**, 1211-1218, doi:10.1038/ng.3909 (2017).
- 4 Herranz, D. *et al.* Metabolic reprogramming induces resistance to anti-NOTCH1 therapies in T cell acute lymphoblastic leukemia. *Nat Med* **21**, 1182-1189, doi:10.1038/nm.3955 (2015).

REVIEWER COMMENTS

Reviewer #2 (Remarks to the Author):

The authors have now addressed all my concerns, they provided clear answers and have included changes where needed.

I have no further comments.

REVIEWER COMMENTS

Reviewer #2 (Remarks to the Author):

The authors have now addressed all my concerns, they provided clear answers and have included changes where needed.

I have no further comments.

ANSWER: We are very thankful for this comment and appreciate this positive feedback on our work.